# A rationally designed miniature of soluble methane monooxygenase enables rapid and high-yield methanol production in *Escherichia coli*

Yeonhwa Yu [1], Yongfan Shi[2], Young Wan Kwon[3], Yoobin Choi[1], Yusik Kim[1], Jeong-Geol Na [2], June Huh [1] ✉ & Jeewon Lee [1] ✉

Soluble methane monooxygenase (sMMO) oxidizes a wide range of carbon feedstocks (C1 to C8) directly using intracellular NADH and is a useful means in developing green routes for industrial manufacturing of chemicals. However, the high-throughput biosynthesis of active recombinant sMMO and the ensuing catalytic oxidation have so far been unsuccessful due to the structural and functional complexity of sMMO, comprised of three functionally complementary components, which remains a major challenge for its industrial applications. Here we develop a catalytically active miniature of sMMO (mini-sMMO), with a turnover frequency of 0.32 s$^{-1}$, through an optimal reassembly of minimal and modified components of sMMO on catalytically inert and stable apoferritin scaffold. We characterise the molecular characteristics in detail through in silico and experimental analyses and verifications. Notably, in-situ methanol production in a high-cell-density culture of mini-sMMO-expressing recombinant *Escherichia coli* resulted in higher yield and productivity (~ 3.0 g/L and 0.11 g/L/h, respectively) compared to traditional methanotrophic production.

Methanotrophs consume ~30 million metric tons of methane every year[1,2] through the oxidation of aqueous methane using methane monooxygenase (MMO) incorporated with essential metal (copper or iron) cofactors. Methanotrophs and MMO have a great biotechnological potential, because they can convert methane and a wide range of other carbon feedstocks to methanol and various value-added chemicals under mild conditions[3–5]. However, despite a long-term, tremendous effort through engineering methanotrophs or recombinant expression of MMO[6–8], low yields and conversion efficiencies remain the major hurdle in developing economically viable biomanufacturing processes. Two different, complex forms of MMO have been identified in methanotrophs: particulate MMO (pMMO), found in nearly all methanotrophs under copper-rich conditions and soluble MMO (sMMO) found in some methanotrophs (including facultative *Methylocella* and *Methyloferula* species[9–11] that possess only sMMO) under copper-limited conditions[12,13]. pMMO is comprised of multiple transmembrane and periplasmic domains, while sMMO is comprised of functionally differentiated three components (hydroxylase, reductase, and regulatory protein). In particular, sMMO produces methanol through the recombination of methane-derived methyl radical and diiron-bound hydroxyl radical[14–16], which necessitates an exquisite, coupled process of electron transfer from native reductants, NADH or NADPH, to the catalytic site.

[1]Department of Chemical and Biological Engineering, Korea University, Anam-Dong 5-1, Seongbuk-Gu, Seoul 02841, Republic of Korea. [2]Department of Chemical and Biomolecular Engineering, Sogang University, Seoul 04107, Republic of Korea. [3]KU-KIST Graduate School of Converging Science and Technology, Korea University, Anam-Dong 5-1, Seongbuk-Gu, Seoul 02841, Republic of Korea. ✉e-mail: junehuh@korea.ac.kr; leejw@korea.ac.kr

As for pMMO, the understanding of electron transfer mechanism is still incomplete due to its complexity associated with the membrane-bound state and a number of possible catalytic sites in the periplasmic and/or transmembrane regions[17,18]. Moreover, pMMO activity depends on electron/proton transfer efficiency in the cytoplasmic membrane, requires additional protein components necessary for stabilization and copper loading[19–21], and cannot receive electrons directly from NADH, and thus recombinant pMMO does not oxidize methane in the intracellular region. Some efforts for in vitro application of pMMO-based, active recombinant enzymes envision the potential use of pMMO[22–24], but its industrial feasibility remains in question because synthetic reductants (e.g., duroquinol, plastoquinol, etc.) are required for the in vitro methane oxidation, thus causing significant commercial weakness. The catalytic mechanism of sMMO, on the other hand, is relatively better understood owing to some seminal studies on enzymological nature, including reaction kinetics[16,25] as well as crystallographic and NMR investigations[16,25], which unveil how the three different components comprising the complex sMMO system are involved in the catalytic cycle[26]. The three components are structurally different but functionally complementary to each other: (1) hydroxylase (MMOH) with a heterohexamer conformation ($\alpha_2\beta_2\gamma_2$) having a diiron active site in each $\alpha$ subunit (MMOH$\alpha$), (2) reductase (MMOR) possessing two different domains—the domain of flavin adenine dinucleotide (FAD) that receives electrons from intracellular NADH and the other domain of ferredoxin (iron-sulfur/$Fe_2S_2$) that provides an intermediate passage of electrons from the FAD to the diiron center of MMOH$\alpha$, and 3) a regulatory protein (MMOB) alternates between binding to MMOH and MMOR, the binding pair depending on the phase of the catalytic cycle of sMMO, and thus plays a critical role in regulating proton transfer and methane/$O_2$ gating[16,27,28]. Due to this structural and functional complexity, the heterologous expression of catalytically active recombinant sMMO in a genetically easily tractable host like *E. coli* remains still unsettled; however, it might be achieved the minimal, core catalytic elements of sMMO were identified and then optimally assembled to produce a catalytic construct.

Our overriding objective in this work is to construct a miniaturized form of native sMMO (denoted as mini-sMMO) through molecular editing based on the structural and functional features of sMMO, which involves trimming, inverting, and assembling/fabricating of modified and/or minimal versions of the three components (MMOH, MMOR, and MMOB). Specifically, the mini-sMMO construct is built based on the following requirements: (1) relocation of the FAD domain of MMOR at a different position so that the electron transfer can occur directly from FAD to the diiron center of MMOH$\alpha$ without passing through ferredoxin domain; (2) trimming of complex MMOH to such a smaller sub-structure that still retains hydroxylase activity; (3) assembly of the modified and/or minimal versions of MMOR, MMOH, and MMOB on a catalytically inert and stable scaffold to construct the mini-sMMO; and finally (4) rapid and high-yield biosynthesis of the mini-sMMO using *E. coli* as a host and concurrently rendering the recombinant *E. coli* to oxidize methane directly using intracellular NADH as an electron donor. It is worthy of note that we employ human apo-ferritin (human heavy chain ferritin/huHF) as the scaffold for the mini-sMMO construction—which is a flexibly guest-adaptable, highly ordered, and stable protein with the shape of a spherical nano-scale particle—and successfully tackle the demanding challenges above. We expect that this approach opens up a route for optimally miniaturizing other multicomponent, complex enzymes and thereby developing a variety of industrially viable biocatalysts and biomanufacturing processes.

## Results

### Important considerations for the design of mini-sMMOs
The electron transfer from NADH to the diiron active site of MMOH via MMOR is an essential precondition in designing the mini-sMMOs.

Since the exact MMOH-MMOR complex structure remains elusive due to the lack of crystallographic evidence, we referred to the simulated structure of MMOH-MMOR complex, previously obtained through the docking of ferredoxin domain of MMOR into the canyon region of MMOH[29], which shows that the diiron active site in MMOH$\alpha$ is 14 Å away from the iron-sulfur cluster of MMOR, i.e. close enough for electron shuttling[30]. Since the FAD domain is located at the top of the docked ferredoxin domain (Fig. 1a, b), the direct distance between the flavin group of FAD and the diiron center must be quite longer (at least 20 to 30 Å away), which is believed to be the reason why MMOR needs the ferredoxin domain to shuttle the FAD electrons to the diiron center. It seems that it could be possible to transfer the FAD electrons directly to the diiron center without passing through the ferredoxin domain if the FAD domain would be relocated to such a position that is sufficiently close to the diiron center through the design of a minimal model of sMMO.

Reportedly, MMOH alone can oxidize methane but very slowly[31] and is much more activated through interacting with MMOB[16]. A structural study of native MMOH-MMOB complex (PDB 4GAM) revealed that the N-terminal region (-35 a.a.) of MMOB plays a critical role in the interaction with reduced MMOH$\alpha$ and that the N- and C-terminus of MMOH$\alpha$ and MMOB, respectively, are facing the same direction (Supplementary Fig. 1a, b)[27], suggesting how to keep the native orientation and interaction of MMOH and MMOB in the minimal model design. Also, the crystallography data of native MMOH$\alpha$ shows that four helices ($\alpha$B, $\alpha$C, $\alpha$E, and $\alpha$F) comprise an inner shell around the diiron center, while the other four helices ($\alpha$A, $\alpha$D, $\alpha$G, and $\alpha$H) form a sort of second shell that stabilizes the inner shell conformation[32,33]. Thus, it seems that the eight helices ($\alpha$A to $\alpha$H) of MMOH$\alpha$, encoded by eight consecutive sequences, need to be involved in the minimal version of MMOH. The above structural features of the MMOH-MMOR- and MMOH-MMOB complexes provide useful guidelines for the design of mini-sMMOs.

### Construction of mini-sMMOs using huHF scaffold
Our first step in designing the mini-sMMO was to create a minimal form of MMOH-MMOR complex. The key strategy in creating the minimal model involved removing the ferredoxin domain (a.a.1-98) from MMOR (a.a.1–348) (Fig. 1a, b) and relocating the remaining FAD domain (a.a. 99–348, referred to as $R_{FAD}$) close to the diiron center of MMOH, enabling direct electron transfer from FAD to the active site. To achieve this, we performed a series of sequence editing trials, aided by structure stability analysis using various in silico approaches, including molecular dynamics (MD) and protein–protein docking (PPD) simulations. The relocation of $R_{FAD}$ was attempted by the following two steps. First, a minimal version of hydroxylase (denoted as $\Delta$H$\alpha$) was created from MMOH$\alpha$, retaining only 258 residues (a.a. 64–321) out of the full 526 residues, while removing other MMOH subunits, MMOH$\beta$ and MMOH$\gamma$, for the minimal model (Fig. 1c, d). The resulting trimmed sequence covers only the helices from $\alpha$A to $\alpha$H, without MMOH$\beta$ and MMOH$\gamma$, which exposes $\alpha$A, $\alpha$B, $\alpha$C, and $\alpha$D and provides more potential binding sites for the $R_{FAD}$ (Fig. 1e).

The second step involved identifying $R_{FAD}$ candidate that poses on the $\Delta$H$\alpha$ surface using PPD simulations. This revealed a bimodal distribution with $R_{FAD}$ binding to either the canyon or the region near $\alpha$A, $\alpha$B and $\alpha$C (referred to as N-terminal half (NTH) region) (Fig. 1e). High-scoring poses were predominantly found in the NTH region, which were further narrowed down by constraining the distance between the C-terminus of $R_{FAD}$ and the N-terminus of $\Delta$H$\alpha$ to within 10 Å. This constraint was introduced to consider only configurations of $R_{FAD}$ linked to $\Delta$H$\alpha$ by forming a single chain with the N-to-C orientation from $R_{FAD}$ to $\Delta$H$\alpha$ (denoted as $R_{FAD}$-$\Delta$H$\alpha$, where the C-terminus of $R_{FAD}$ is linked to the N-terminus $\Delta$H$\alpha$). The $R_{FAD}$-$\Delta$H$\alpha$ was then relaxed using an energy minimization process followed by MD equilibration in water medium at 300 K and 1 bar, demonstrating that the diiron center

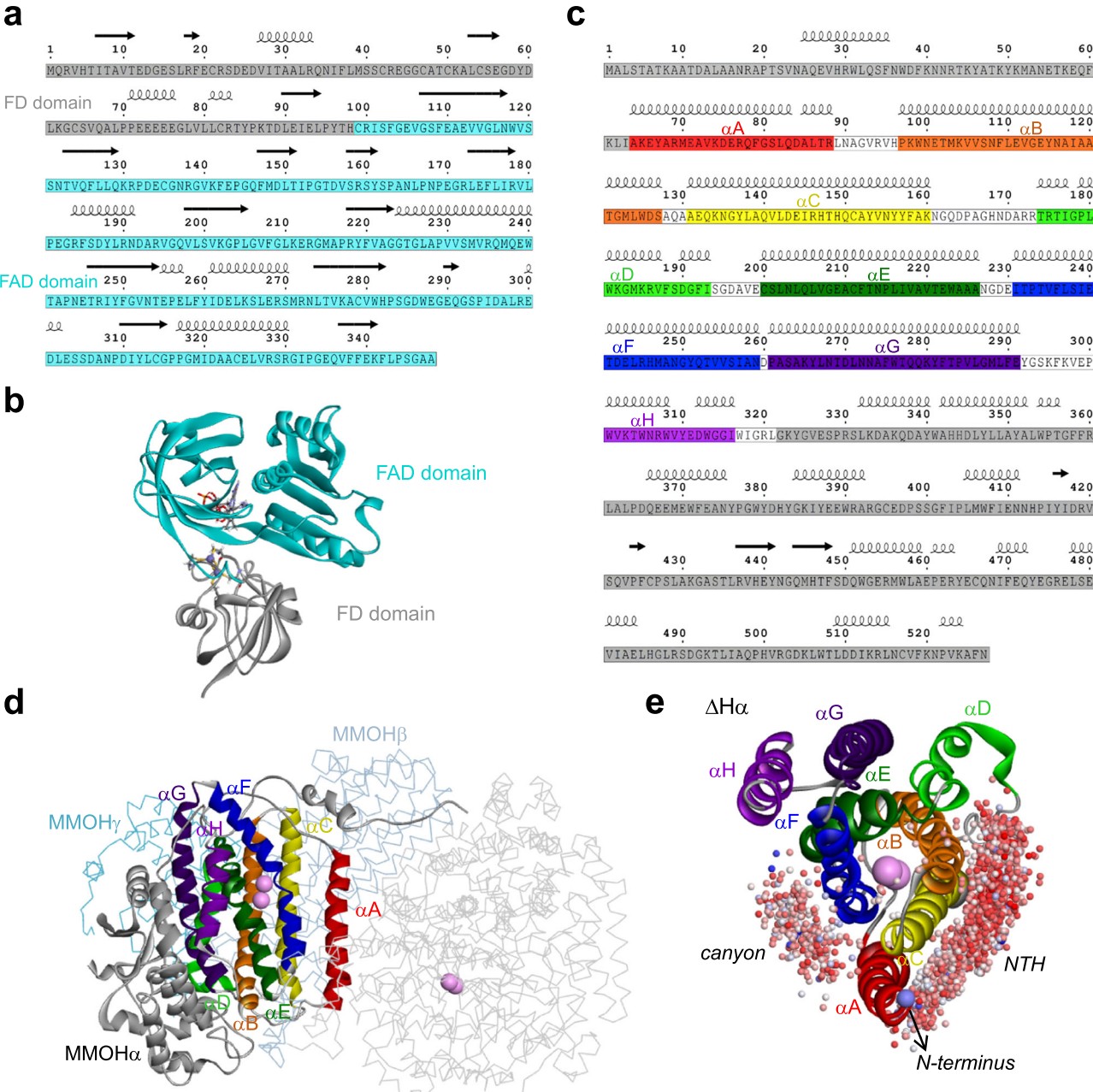

**Fig. 1 | Molecular editing of MMOR and MMOH for miniaturizing sMMO.**
**a**, **b** Primary (**a**) and tertiary structure (**b**) of MMOR. **c**, **d** Primary structure of MMOHα (**c**) and tertiary structure of MMOH (**d**). The gray-colored regions in (**a**–**d**) were removed from the native structure for constructing the sMMO miniatures (mini-sMMOs). The gray region in (**a**, **b**) indicates the ferredoxin domain, while the remaining cyan region represents the FAD domain. The rainbow-colored regions in (**c**, **d**) indicate the eight helices of MMOHα (αA to αH in rainbow order) surrounding the diiron center (depicted as pink spheres in (**d**)). In (**d**), the MMOHα subunit in the MMOH protomer is represented by ribbons, while the MMOHβ and MMOHγ subunits and the other protomers in the dimer are represented as wires. The FAD domain of MMOR (defined as $R_{FAD}$) in (**a**, **b**) and the eight helices and white-colored regions in MMOHα (defined as ΔHα) in (**c**, **d**) are included in the mini-sMMOs. **e** Structure of ΔHα with center-of-mass positions (colored spheres) of $R_{FAD}$ poses, obtained from PPD simulations, indicating the bimodal distribution of $R_{FAD}$ poses in the canyon and NTH (N-terminal half). The docking score of the poses is indicated by a color gradient that ranges from blue (representing a low score) to red (representing a high score). The N-terminus of ΔHα is indicated by an arrow.

of ΔHα and the FAD of $R_{FAD}$ are well secured in the $R_{FAD}$-ΔHα preserving the initial skeleton with RMSD of ~8 Å (Fig. 2a, b) and also that the distance between the diiron and N1- or N5-position of the FAD ($d$ = 13.5 Å on average, short enough to permit electron shuttling) is well-equilibrated[34] (Fig. 2c, d), suggesting that the $R_{FAD}$-ΔHα is a viable option in constructing the minimal model of sMMO.

From the native MMOH-MMOB complex structure (PDB 4GAM, *M. capsulatus* (Bath)) (Fig. 2e and Supplementary Fig. 1), it seems to be of crucial importance to mimic the native interaction between MMOHα

and the N-terminal region of MMOB. For this, we examined the molecular interaction between ΔHα and the reversed sequence of native MMOB (denoted as retroB) as well as between ΔHα and the normal sequence of native MMOB (denoted as B) through a series of MD simulations, resulting in the contact maps between the residues of ΔHα and B/retroB (Fig. 2e–g). In the contact maps, the residue indices 1 to 258 correspond to a.a. 64–321 of native MMOHα (i.e., the entire sequence of ΔHα), while the indices 259 to 390 correspond to a.a. 2–133 of the normal or reversed sequence of native MMOB. The

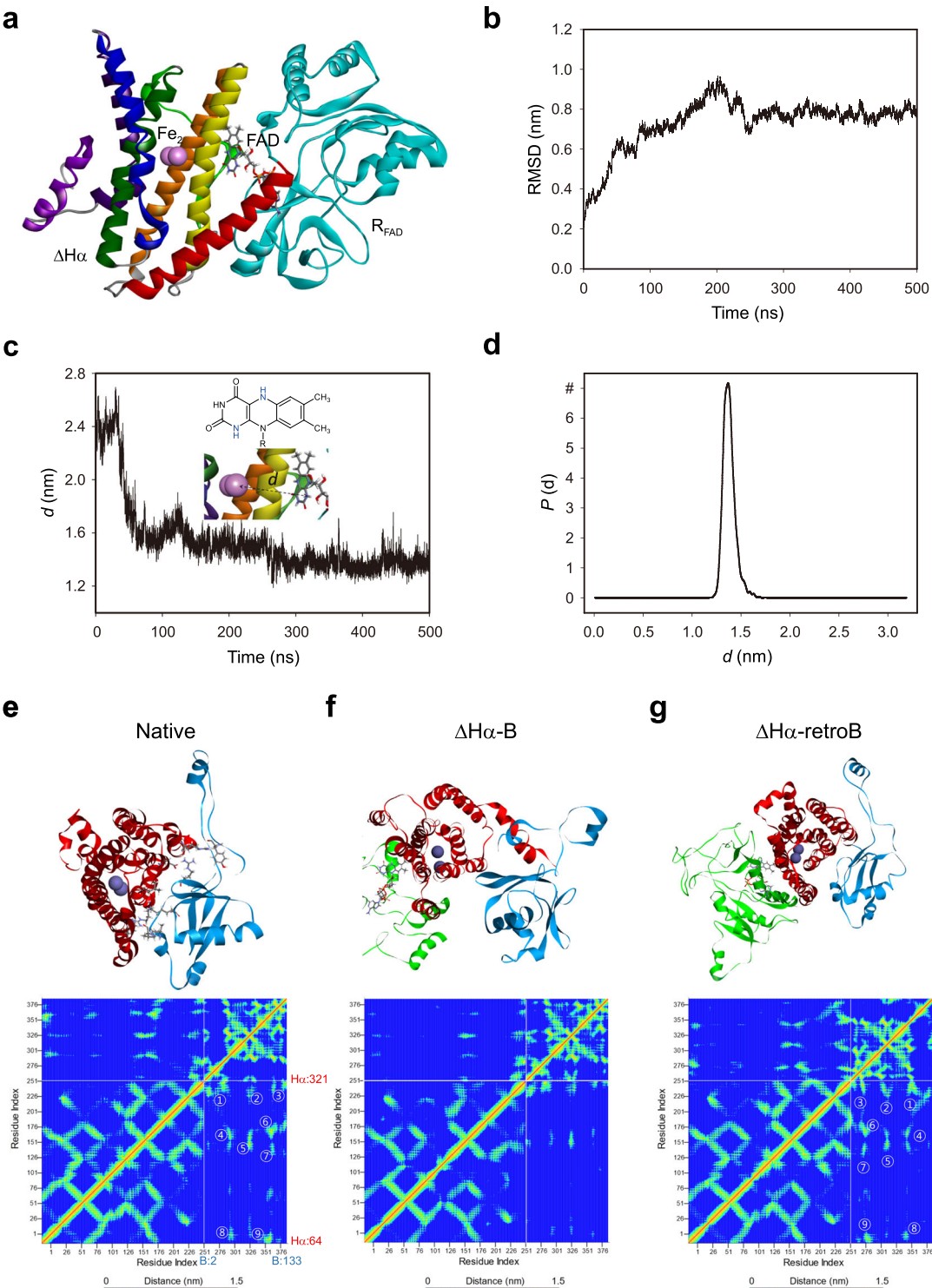

**Fig. 2 | MD simulations for R$_{FAD}$-ΔHα. a** 3D structure of equilibrated R$_{FAD}$-ΔHα. **b** Time-dependent RMSD of R$_{FAD}$-ΔHα in target to that in the native form. **c** The distance ($d$) between diiron and N1- or N5-position of FAD as a function of the simulation time. The inset figures are the chemical structure of FAD (top) and a part of the 3D structure of sMMO-m1, showing the distance ($d$) between diiron and FAD (bottom). **d** The probability density function $P(d)$ obtained from (**c**). All results were obtained from isobaric-isothermal MD simulations of R$_{FAD}$-ΔHα equilibrated in water medium at 300 K and 1 bar for 500 ns. **e**–**g** 3D structure and contact map between ΔHα and B in native sMMO (**e**), between ΔHα and B in mini-sMMO (**f**), and between ΔHα and retroB in mini-sMMO (**g**). In 3D structures, red, sky-blue, and green ribbons represent ΔHα, B/retroB, and R$_{FAD}$, respectively, and violet-colored spheres represent diiron. Source Data were provided with this paper.

contact map between native MMOHα and MMOB (Fig. 2e) reveals the nine significant interactive regions as follows: (1) Y8 (B)−R307 (ΔHα) & E34 (B)−R320 (ΔHα); (2) R73 (B)−T304 (ΔHα); (3) I113 (B)−I316 (ΔHα) & D133 (B)−L320 (ΔHα); (4) K43(B)−L237 (ΔHα); (5) R73 (B)−E222 (ΔHα); (6) L106 (B)−R245 (ΔHα); (7) S111 (B)− N214 (ΔHα); (8) E47 (B)−R77

(ΔHα); and (9) V107 (B)−K74 (ΔHα) (Fig. 2e), while in the contact map between ΔHα (in the R$_{FAD}$-ΔHα) and B, the nine interactive regions were notably weakened or disappeared (Fig. 2f). The interaction of Y8(B)-R307(ΔHα) in the region 1 of MD-simulated contact map−which was also experimentally investigated through a previous study[27]−

causes the reorientation of W308 of MMOHα through π-interaction and enhances the structural stability, and likewise, the interaction of S111(B)-N214(ΔHα) in the region seven induces the reorientation of T213 of MMOHα and thus facilitates proton transfer, as experimentally confirmed through the same study[27]. Surprisingly, nearly the same contact pattern as Fig. 2e was observed in the contact map between ΔHα (in the R$_{FAD}$-ΔHα) and retroB (Fig. 2g), indicating that if optimally assembled to reproduce the native interactions between MMOH and MMOB, the R$_{FAD}$-ΔHα and retroB—the minimal and modified components of native sMMO—could enable the construction of catalytically active mini-sMMO.

Based on the findings above, several mini-sMMOs were designed by assembling the R$_{FAD}$-ΔHα and retroB (or B as a control) on the apoferritin (huHF) scaffold. huHF is an iron-free, hollow protein particle (with a diameter of ~12 nm) consisting of 24 identical heavy chain subunits and can be synthesized in a large quantity even in *E. coli*[35,36]. The stable 24mer structure of huHF is formed through robust inter-helical docking via many hydrogen bonds and salt bridges[37–39]. Each heavy chain subunit is comprised of four antiparallel α-helices and a short C-terminal α-helix, and these helix-rich subunits are arranged in a unique octahedral 432 symmetry, resulting in the formation of two- to fourfold axes within the huHF particle structure. In particular, it is worthy of note that each fourfold axis, where four C-termini are gathered together on a narrow surface area (~2 nm$^2$), has a highly flexible conformation. That is, the four C-termini of the fourfold axis are buried inside the cage of native huHF but can be flipped outward when are genetically linked with foreign protein, depending on the foreign protein size. Accordingly, the six fourfold axes per huHF particle are notably favorable sites for linking the R$_{FAD}$-ΔHα and B/retroB because the linked components can interact very closely with each other on the exterior surface of huHF, which makes huHF a very attractive scaffold to construct the mini-sMMOs.

We constructed a series of mini-sMMOs (sm1, sm2, and sm3) by genetically linking the R$_{FAD}$-ΔHα and B/retroB to each fourfold axis of huHF (Fig. 3), the objective being to make the mini-sMMOs preserve both the stable 24mer structure of huHF and the catalytic function of sMMO. Actually, the sm1 to sm3 are hetero-multimers (α$_x$β$_y$ where $x + y = 24$) comprised of two different subunits (α and β), meaning that each of the R$_{FAD}$-ΔHα and B/retroB are genetically linked to the different C-terminus in the fourfold axis of huHF, as described in Fig. 3. In all of sm1 to sm3, the α subunit was designed through genetically linking the N-terminus of R$_{FAD}$-ΔHα to the C-terminus of a huHF subunit (huHF$_α$) via a glycine-rich linker (L). The β subunits of sm1 and sm2 were designed by genetically linking the N-terminus of retroB and B, respectively, to the C-terminus of another huHF subunit (huHF$_β$) via the same L, while the huHF$_β$ of sm3 corresponds to native huHF subunit. We emphasize that the β subunit of sm1 was designed by linking the C-terminus of native MMOB to the C-terminus of huHF$_β$, allowing the N-terminus of MMOB to freely interact with the ΔHα in the α subunit, whereas in the β subunit of sm2, the N-terminus of native MMOB is linked to the C-terminus of huHF$_β$ and thus is subject to being tied up to the surface of huHF scaffold. As a control, we also constructed smδ by directly linking R$_{FAD}$ and ΔHα without using the huHF scaffold (Fig. 3).

## Synthesis, catalytic activity, and characterization of mini-sMMOs

We synthesized the sm1 to sm3 using the recombinant *E. coli* BL21(DE3) [F-ompThsdSB(rB-mB-)] transformed with two plasmid expression vectors (Supplementary Fig. 2a–c). The culture medium contained ferrous sulfate of 0.4 mM, which is a sufficient amount for fully saturating the active sites of mini-sMMOs (Supplementary Fig. 3a) but never inhibits the recombinant *E. coli* growth (Supplementary Fig. 3b–d). Sodium dodecyl sulfate-polyacrylamide gel electrophoresis

(SDS-PAGE) analyses show that the expressed mini-sMMOs constitute ~32 to 38% of total *E. coli* proteins, 64 to 83% of which is a soluble protein, while smδ mostly aggregated to insoluble inclusion bodies (Supplementary Fig. 4a, b). The cytoplasmic, soluble mini-sMMOs were readily purified through one-step Ni$^{2+}$-affinity purification after cell lysis and subsequent centrifugal separation, whereas the insoluble smδ was solubilized through a laborious downstream process including dialysis-based refolding steps (Methods). Also, the number of α and β subunits comprising each mini-sMMO (24mer) is about 7 and 17, respectively (Supplementary Fig. 4a). It is also obvious that every mini-sMMO has the shape of native ferritin-like spherical particles with a diameter of 18 to 20 nm (Fig. 4a). The enlarged size of mini-sMMOs is consistent with the fact that the R$_{FAD}$-ΔHα and B/retroB linked to the C-terminus of huHF subunit are flipped outward and thus localized on the exterior surface of huHF scaffold, which was confirmed through immunoblotting analyses (Supplementary Fig. 5). Further, we verified that sm1 shows a negligible activity of FOC (ferroxidase center, located in the internal surface of huHF scaffold[40,41]) (Supplementary Fig. 6), meaning that the potential iron-binding site of internal FOC does not exert an influence on the methane-oxidizing activity of sm1.

We estimated the methane-oxidizing activities of sm1 to sm3, wild-type huHF, and refolded smδ at 30°C by measuring the amount of methanol produced in the methane-purged reaction buffer (1 mL, 1.2 μM of sm1 to sm3 and wild-type huHF, 8.4 μM of smδ, 0.3 mM NADH) in a closed 20-mL container, where the headspace (19 mL) was initially filled with the mixture of methane (15 mL) and air (4 mL). The amount of methanol was quantified using gas chromatography after evaporating all the produced methanol to the headspace of the closed container (Methods, Supplementary Fig. 7a–c). The NADH concentration of 0.3 mM was determined through the previous test experiments using sm1 (1.2 μM) and the same reaction system (Supplementary Fig. 8). Notably, approximately 2000 moles of methanol were produced per mole of sm1 within 20 h (Fig. 4b) due to the methane-oxidizing activity of sm1, as confirmed through $^{13}$C-NMR analysis (Fig. 4c and Supplementary Fig. 9). The turnover frequency (TF) of sm1 was about 0.32 s$^{-1}$ (Supplementary Fig. 10). The other two mini-sMMOs (sm2 and sm3) produced a far less amount of methanol, and smδ and wild-type huHF never produced methanol (Fig. 4b). The far lowered activity of sm2 and sm3 implies that the correct interaction between the N-terminal region of MMOB and ΔHα is critical to the catalytic activity of mini-sMMO, which is also supported by the earlier result of the contact map between ΔHα and retroB (Fig. 2g). To evaluate the catalytic efficiency in sm1, we estimated methanol-NADH coupling efficiency (ratio of methanol produced per enzyme (mol/mol) to NADH consumed per enzyme (mol/mol)) in the in vitro methane oxidation by sm1 and sm1-MMOR (that was constructed through replacing R$_{FAD}$ of sm1 with a complete reductase unit of MMOR (FAD domain + ferredoxin domain)), demonstrating that the coupling efficiency of sm1 is notably higher than that of sm1-MMOR (Supplementary Fig. 11) and thus suggesting that R$_{FAD}$ of sm1 provides an efficient passage of electrons from NADH to diiron catalytic center. Further, the following evidence seems to show that the iron quantity in the active center of sm1 is critical to the catalytic methane oxidation: (1) when synthesized using FeSO$_4$-free medium, sm1 showed a far lowered methane-oxidizing activity (Fig. 4d), (2) the methane-oxidizing activity of sm1-cat$_{mut}$—a mutant of sm1, prepared through the mutation of six ligands (E114A, E144A, H147A, E209A, E243A, and H246A) around the diiron center of native MMOHα—was almost negligible (Fig. 4d), and (3) approximately two iron atoms are loaded per α subunit of sm1 (i.e., per catalytic center of ΔHα), which is based on the result of Fig. 4e showing that the difference in iron content per mini-sMMO between sm1 (α$_7$ β$_{17}$, Supplementary Fig. 4a) and sm1-cat$_{mut}$ is nearly 14 iron atoms.

Next, sm1 was further characterized in detail through the analyses of electron paramagnetic resonance (EPR), X-ray absorption near-edge

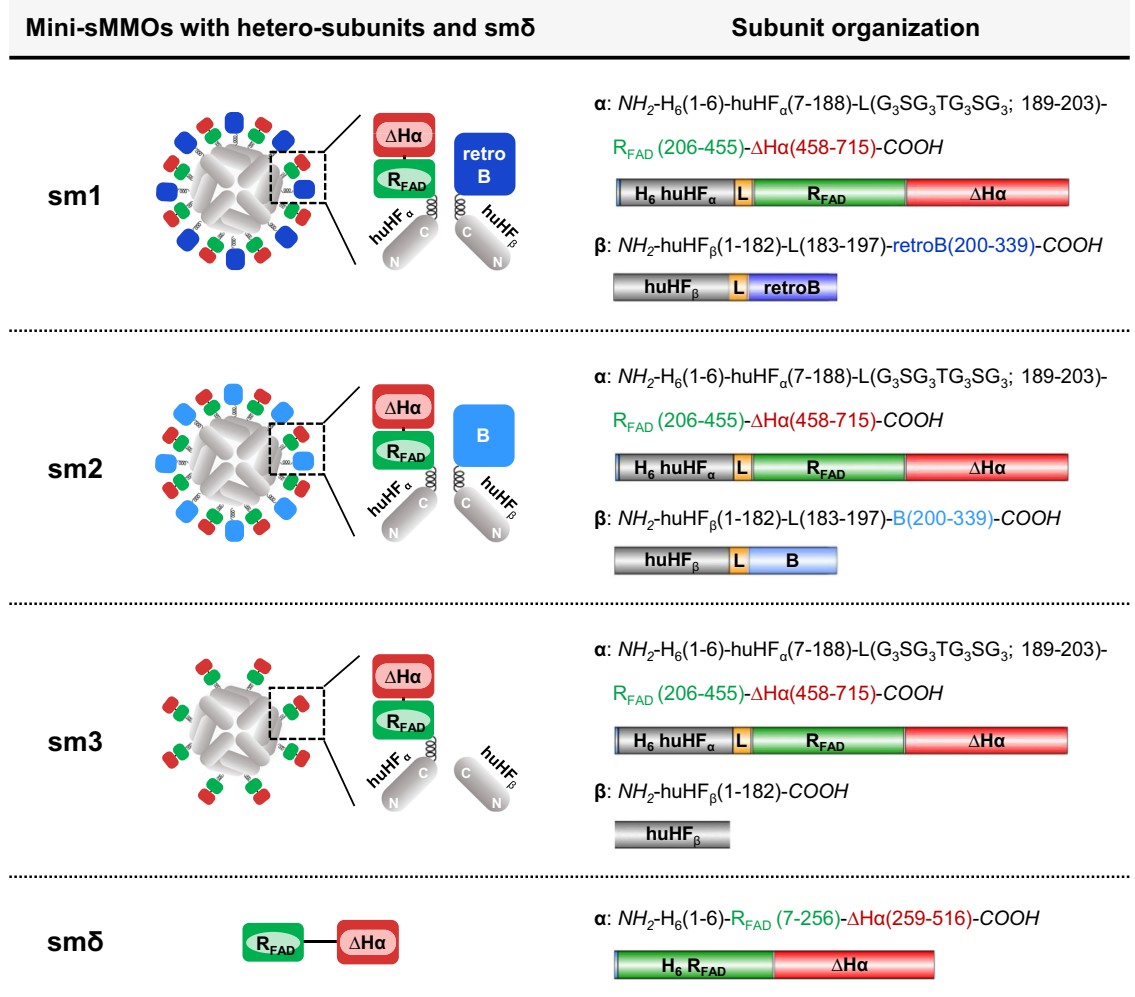

**Fig. 3 | Schematics of the whole enzymes, protomers, and subunits of mini-sMMOs (sm1 to sm3) and smδ.** sm1 and sm2 are comprised of both RFAD-ΔHα and B/retroB, while sm3 is comprised of RFAD-ΔHα only. Each N-terminus (NH2) of RFAD-ΔHα, B, and retroB is linked to the C-terminus (COOH) of huHF subunit (huHFα or huHFβ). That is, in sm1, the C-terminus of native MMOB was linked to the C-terminus of huHFβ, while in sm2, the N-terminus of native MMOB was linked to the C-terminus of huHFβ. In RFAD-ΔHα, the N-terminus of ΔHα (Ala64 to Leu321 of native MMOHα) is linked to the C-terminus of RFAD (FAD-binding domain of native MMOR). H6 is inserted for Ni2+-affinity purification, and L in sm1 to sm3 represents a linker peptide (G3SG3TG3SG3). smδ is only RFAD-ΔHα without linking to the huHF subunit.

structure (XANES), and extended X-ray absorption fine structure (EXAFS) using the sm1-afr sample (taken at 16 h after sm1 is used for in vitro methane oxidation in the presence of NADH). The strong iso-tropic signal at g = 2 in the EPR spectra of sm1-afr (Fig. 5a) indicates the generation of FAD radicals[42], which seems to be due to the active transfer of NADH electrons to FAD during methane oxidation. (The existence of FAD was also confirmed later through measuring FAD-specific fluorescence signal from sm1.) The XANES spectra (Fig. 5b) shows that both Fe(II) and Fe(III) are present in sm1-afr, the portion of Fe(III) being higher than Fe(II). Azide-diferric complex analysis[43,44] using sm1-afr and sm1-cat$_{mut}$-afr (prepared through the same proce-dure as sm1-afr) demonstrates that the distinct absorbance peaks (at 345 and 450 nm) for azide-diferric oxo-bridge complex were evidently detected with sm1-afr but never detected with sm1-cat$_{mut}$-afr, indicat-ing the presence of diiron cluster in the catalytic center of sm1 (Sup-plementary Fig. 12a,b). The absorption peaks of the EXAFS spectra of sm1-afr were fitted with the PDB structure (4GAM) of MMOH-MMOB complex from *M. capsulatus* (Bath), as presented in Fig. 5c with Debye-Waller factors (σ² between 0.0 and 0.013). The fitted EXAFS spectra demonstrate that the inter-iron distance of sm1-afr is 2.755 Å. Conse-quently, the fitted EXAFS (Fig. 5c) and azide-diferric complex analysis

(Supplementary Fig. 12a, b), as well as ICP-MS (Fig. 4e) and XANES (Fig. 5b) analysis, indicate the presence of diiron cluster in the catalytic center of sm1.

## FAD localization and electron shuttling within mini-sMMO structure

In addition to the result of EPR analysis showing the existence of FAD in sm1 (Fig. 5a), the MD-simulated structure of R$_{FAD}$-ΔHα further shows that FAD is likely to be localized in the interface between R$_{FAD}$ and ΔHα (Fig. 6a–c and Supplementary Movie 1), mainly comprised of five hydrophobic residues (Ile305, Leu325, Val328, Leu329, and Ile332) of ΔHα (Fig. 6d). We constructed a sm1 mutant (sm1-FAD$_{mut}$) by replacing the five hydrophobic residues of ΔHα with hydrophilic residues (Glu/Thr) (Fig. 6e and Supplementary Fig. 13a, b) and measured the fluor-escence intensity from FAD in and the methanol production by the sm1-FAD$_{mut}$. Notably, it was found with the sm1-FAD$_{mut}$ that the FAD content is only 35% of sm1 (Fig. 6f), and the methanol production decreased to only 20% of the methanol produced by sm1 under the same reaction condition (Fig. 6g), suggesting that the hydrophobic interface between R$_{FAD}$ and ΔHα play a critical role in the stable loca-lization of FAD in sm1.

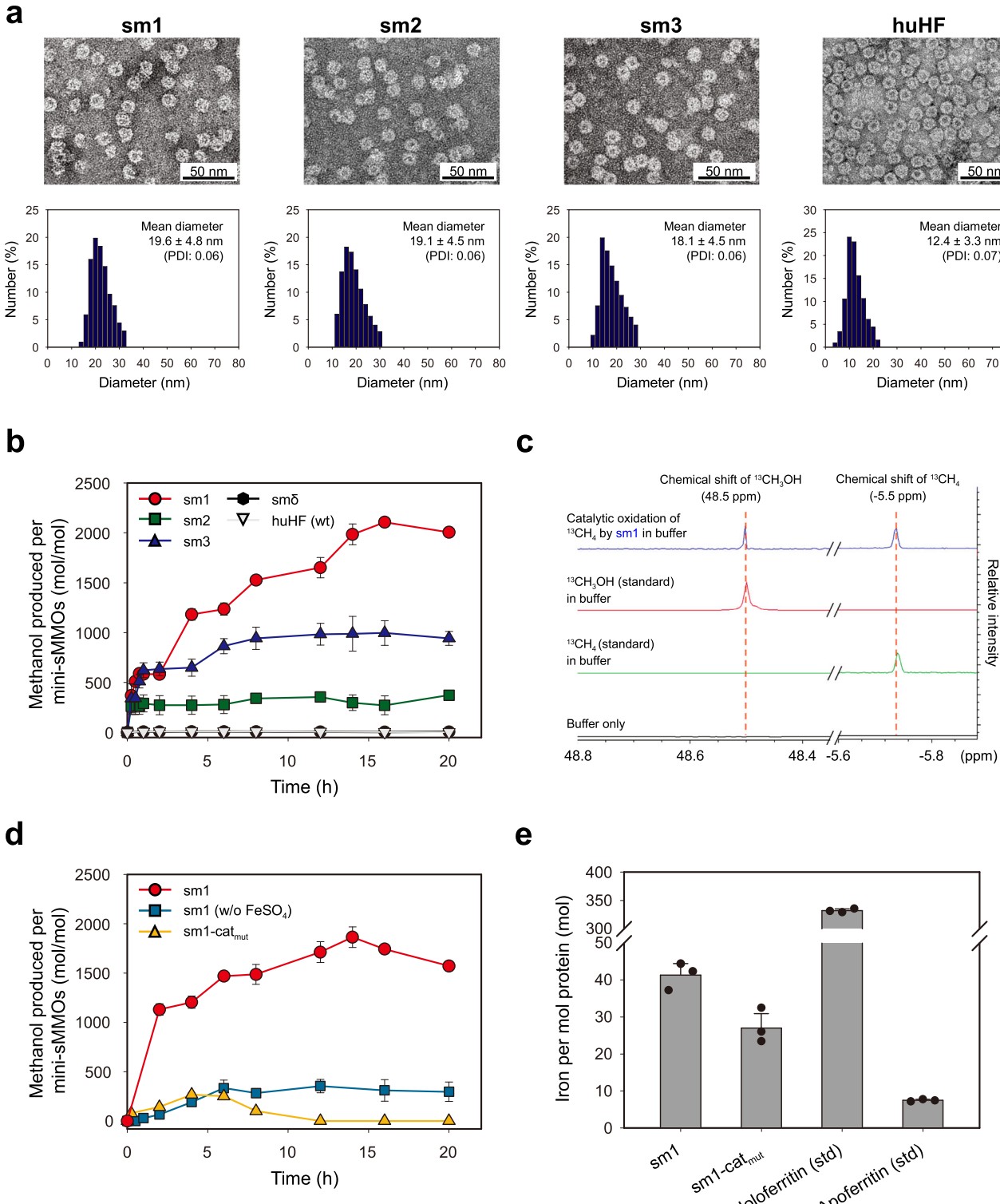

**Fig. 4 | Characterization and catalytic activity analysis of mini-sMMOs. a** TEM images (top) and DLS data (bottom) of synthesized mini-sMMOs. (PDI (poly-dispersity index) value of less than 0.08 indicates a sufficiently high size uni-formity.) This experiment has been repeated at least three times. **b** Cumulative amount of methanol produced through the methane oxidation by mini-sMMOs (sm1 to sm3), refolded smδ, and wild-type huHF. Data were obtained from three or more independent experiments. Mean ± s.d. **c** $^{13}$C-methane oxidation activity of sm1, analyzed through $^{13}$C-NMR spectroscopy. **d** Cumulative amount of methanol produced by sm1 synthesized using the growth medium with and without FeSO$_4$ supplementation and by sm1-cat$_{mut}$ (catalytic site mutant of sm1). NADH (0.3 mM)

was used in the in vitro methane oxidation in (**b–d**). Data were obtained from three or more independent experiments. Mean ± s.d. **e** Analysis of iron content in sm1, sm1-cat$_{mut}$, commercial standards (equine holo- and apoferritin (Sigma-Aldrich)). The iron content in sm1 (~41 mol) represents iron adsorbed on both active and non-specific sites, including H$_6$, while the iron content in sm1-cat$_{mut}$ (~27 mol) repre-sents iron adsorbed on non-specific sites. Accordingly, the difference in iron con-tent between sm1 and sm1-cat$_{mut}$ (~14 mol) is the amount of iron adsorbed solely on the active site. $N = 3$ independent experiments. Mean ± s.d. Source Data are pro-vided with this paper.

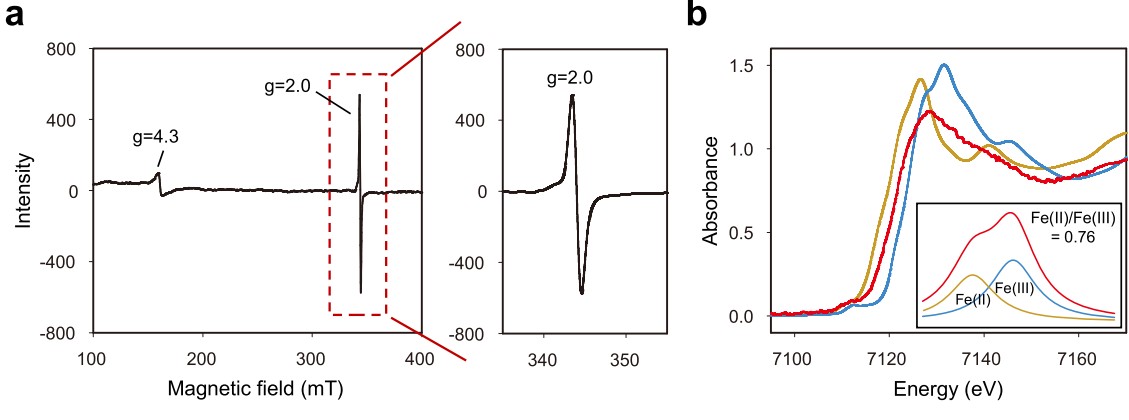

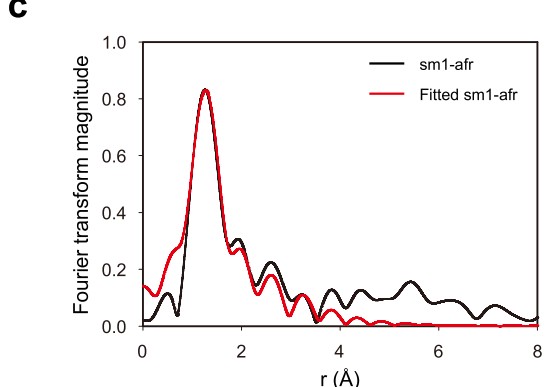

| bond | R (Å) | $\sigma^2$ | N |
|---|---|---|---|
| Fe-O | 1.863 | 0.009 | 5 |
| Fe-N | 2.496 | 0.003 | 2.5 |
| Fe-Fe | 2.755 | 0.013 | 1 |
| Fe-C | 3.661 | 0.000 | 0.3 |

**Fig. 5 | Spectroscopic analyses of sm1. a** EPR spectra of sm1-afr, prepared after methane oxidation. **b** XANES spectra of sm1-afr and the spectrum fitting (inset) to estimate the ratio of Fe(II) to Fe(III) in the designated region of energy (7106 to 7118 eV, indicated by a dotted circle). **c** EXAFS spectra of sm1-afr, fitted using *Artemis* software with the PDB crystal structure (4GAM) of MMOH-MMOB complex of *M. capsulatus* (Bath). (R(Å), $\sigma^2$, and *N* represent interatomic distance, bond variance (Debye-Waller factor), and degeneracy of the path, respectively). Source Data are provided with this paper.

From the MD-simulated structure of $R_{FAD}$-ΔHα, it seems that the side-chain oxygen and nitrogen atoms of three hydrophilic residues (Tyr305, Thr335, and His336) of ΔHα are very close to the FAD nitrogen atoms and thus likely to be involved in electron shuttling from FAD to the diiron center (Fig. 6b and Supplementary Movie 1). We constructed the four different mutants of sm1 (sm1-e$_{mut1}$ to sm1-e$_{mut4}$) by replacing the three hydrophilic residues above with hydrophobic Ala/Phe (Fig. 6e and Supplementary Fig. 13a, b) and estimated the methane-oxidizing activity. Notably, compared to sm1, the methanol production by the four mutants decreased by 55 to 80% under the same reaction condition, while the FAD content of the four mutants remained nearly equal to sm1 (Fig. 6f, g). Considering the non-significant change in secondary structures (Supplementary Fig. 14) of sm1 and its mutants (sm1-cat$_{mut}$, sm1-FAD$_{mut}$, and sm1-e$_{mut1}$), the changes in iron content (Fig. 4eand Supplementary Fig. 15) and methanol production (Fig. 6g) are not attributed to the changes in overall structure caused by the mutations and accordingly indicate that the three residues, Tyr305, Thr335, and His336 comprise the electron shuttling path in sm1, indispensable for the catalytic oxidation of methane, Tyr305 and His336 being more important for methanol production than Thr335 (Fig. 6g).

### Rapid and high-yield methanol production by sm1-expressing *E. coli*

We earlier confirmed the methane-oxidizing activity of sm1 in vitro with the external supply of NADH as a reductant. Here, we attempted a whole cell-based methane oxidation in a high-cell-density culture of sm1-expressing *E. coli* without the additional supply of any reductants, which might be a meaningful first step for the economically viable biomanufacturing of methanol. For the whole cell-based methanol production, we carried out the bioreactor operation—a pH-stat fed-batch operation with the feeding of glucose as a sole carbon and energy source—using the sm1-producing recombinant *E. coli*. When the optical density (absorbance at 600 nm) of the recombinant *E. coli* culture reached around 30, the sm1 expression was initiated by adding IPTG (1 mM), and after 12 h, methane and air mixture (3:7 v/v) was supplied to the fed-batch culture for methanol production (Fig. 7a). As a result, the maximum methanol concentration reached around 3.0 g/L within 8 h, indicating that the recoverable methanol productivity is 0.36 g/L/h (Fig. 7b and Supplementary Fig. 16). Through [13]C-NMR spectroscopy analysis of the [13]C-methane oxidation products in the recombinant *E. coli* culture, we confirmed that the methanol production was due to the supplied methane oxidation by sm1 expressed in the recombinant *E. coli* (Supplementary Fig. 17). Considering the entire culture period (27 h) covering initial cultivation (phase 1), sm1 expression (phase 2), and methane oxidation (phase 3), the recoverable methanol productivity is 0.11 g/L/h, which is still a notably much higher productivity compared to the methanol production by methanotrophs (0.002 to 0.005 g/L/h)[45–47]. Although the maximum peaks of intracellular NADH were detected before methanol production began (at 4 and 18 h), the NADH concentration remained nearly constant during the period of methanol production (Fig. 7b, c). After the methanol concentration reached the maximum in phase 3, the cell density and acetate, glucose, and methanol concentrations were nearly time-invariant (Fig. 7b, c), while the expression level of recombinant sm1 remained unchanged (Supplementary Fig. 18). This means that the fed-batch culture was under a steady state, that is, the specific rates of cell growth and methanol production are balanced with the dilution rate by fresh medium feeding.

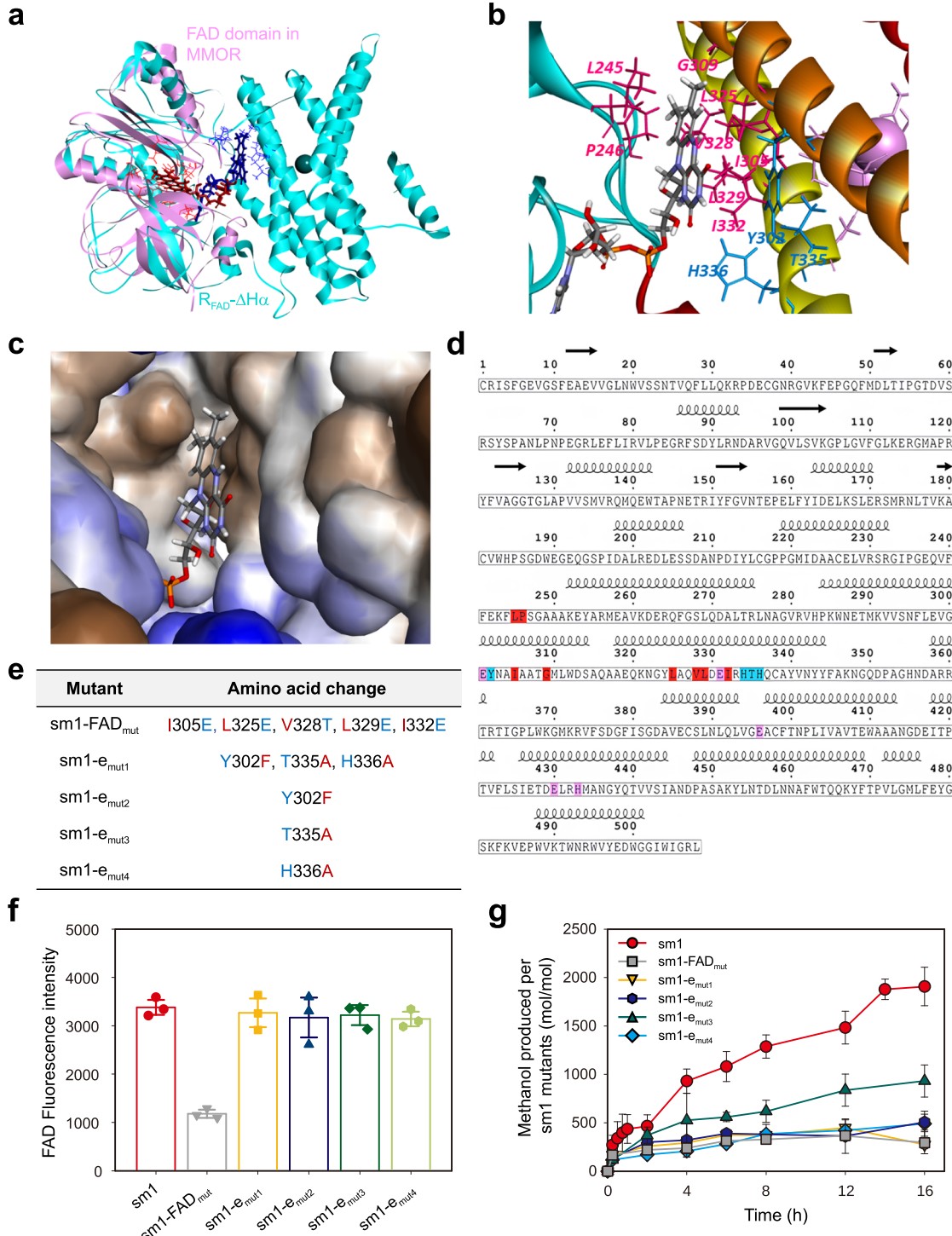

**Fig. 6 | Identification of FAD-binding region and electron shuttling path through MD simulation and mutant characterization of sm1. a** In silico super-position of FAD domain in native MMOR (pink) and $R_{FAD}$-ΔHα in sm1 (cyan). In the native MMOR, FAD and FAD-binding residues are marked by red sticks, while in $R_{FAD}$-ΔHα, they are marked by blue sticks, showing that FAD is localized much closer to the diiron center in $R_{FAD}$-ΔHα. **b, c** Closer view of FAD in the hydrophobic pocket of $R_{FAD}$-ΔHα, shown by in silico imaging. The putative residues comprising the hydrophobic pocket is visualized by magenta sticks in (**b**) and by a reddish groove on the molecular surface in (**c**), where hydrophobicity is indicated by a color gradient that ranges from red (representing hydrophobic) to blue (representing hydrophilic). In (**b**), the residues potentially involved in the electron shuttling are shown as blue sticks, and the diiron ligand residues are shown as pink sticks. All

results were obtained from isobaric-isothermal MD simulations of $R_{FAD}$-ΔHα equi-librated in water medium at 300 K and 1 bar for 500 ns. **d** Primary structure of $R_{FAD}$-ΔHα. The hydrophobic residues involved in FAD-binding residues, diiron ligands, and those potentially involved in electron shuttling are color-coded in red, pink, and blue, respectively. **e** Table showing the sm1 mutants in the putative FAD-binding region (sm1-$FAD_{mut}$) and electron transport path (sm1-$e_{mut1}$ to sm1-$e_{mut4}$). Sky-blue symbols indicate hydrophilic amino acids, and red symbols indicate hydrophobic amino acids. **f** Fluorescence intensity from FAD in sm1 and the mutants of (**e**) (sm1-$FAD_{mut}$ and sm1-$e_{mut1}$ to sm1-$e_{mut4}$). $N = 3$ independent experiments. Mean ± s.d. **g** Cumulative amount of methanol produced by sm1 and the mutants of (**e**). $N = 3$ independent experiments. Mean ± s.d. Source Data are provided with this paper.

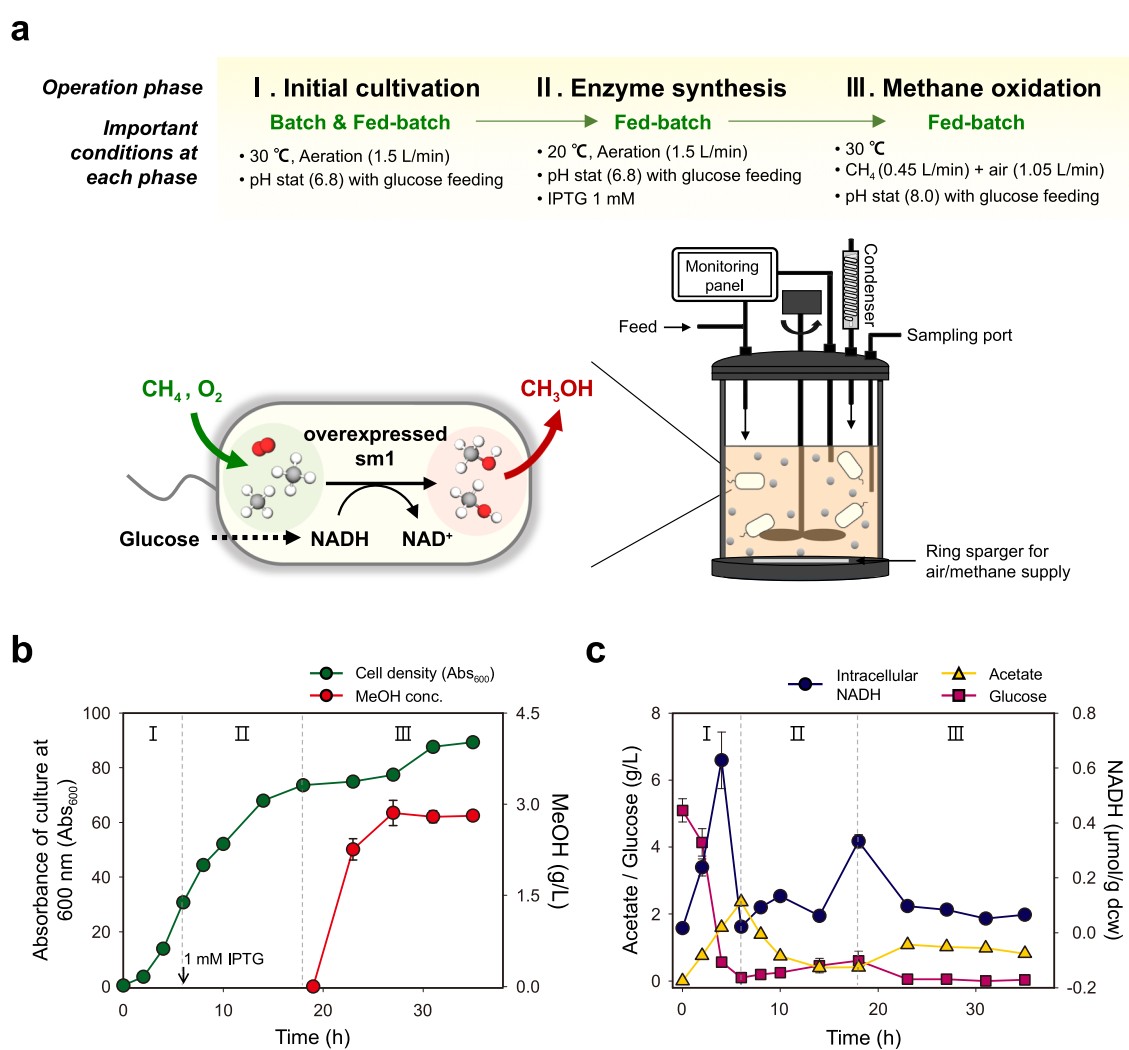

**Fig. 7 | In situ methane oxidation in the fed-batch culture of recombinant *E. coli.* a** A schematic diagram of the fed-batch bioreactor (3-L working volume) operation for sm1-expressing *E. coli* cultivation and methanol production. (A more detailed procedure is described in Methods). **b** Time-course variation in cell growth (OD$_{600}$) and methanol production (g/L) in the fed-batch bioreactor. I, II, and III represent the operation phases described in (**a**). **c** Time-course variation in intracellular NADH, acetic acid, and glucose concentrations during the fed-batch operation. (The culture absorbance was measured once, while all other measurements were done three times at a time. (Mean ± s.d.)). Source Data are provided with this paper.

The methanol production by the sm1-producing recombinant *E. coli* showed a notable improvement in productivity compared to the traditional methanotrophic production, which often requires the additional steps for cell concentration after cultivation due to the slow cell growth and low cell density[48,49]. As demonstrated above, the in situ methanol production in the high-cell-density culture of mini-sMMO-expressing recombinant *E. coli* is a promising strategy for rapid and high-yield methanol biomanufacturing, although important variables (e.g., bioreactor design, feeding strategy, production scale, etc.) need to be further optimized.

## Discussion

sMMO can oxidize a wide range of carbon feedstocks (C1 to C8) directly using intracellular NADH (a native reductant) and thus is a promising enzyme in developing green routes for industrial-scale manufacturing of chemicals; however, the high-throughput biosynthesis of active recombinant sMMO in *E. coli*—which rapidly grows to a high-cell density and has been used for producing many recombinant proteins at industrial scale—has long been exceptionally difficult because of the structural and functional complexity of sMMO, comprised of three components—MMOH, MMOR, and MMOB. Recently, the co-expression of recombinant sMMO and *E. coli* chaperonins was

reported[50]; however, the effectiveness of the expression system was not fully evidenced, although catalytic oxidation of p-nitrobenzene was verified with the EPR-based characterization of iron center of hydroxylase. To circumvent this issue, in this work, we attempted a molecular editing of sMMO (from *M. capsulatus* (Bath)), which is based on an optimal reassembly of the minimal sub-structures (*NH$_2$*-R$_{FAD}$-ΔHα-*COOH*) and modified version (retroB) of the three components on the catalytically inert and stable huHF scaffold. ΔHα is a minimal version of native MMOH (a.a. 64–321 covering the diiron active site and αA to αH helices out of the full 526 residues of native MMOHα), while R$_{FAD}$ is a ferredoxin domain-deleted version of native MMOR (a.a. 99–348 covering only FAD domain), and retroB is the reversed sequence of native MMOB. Based on the results of in silico analyses (MD and PPD simulations for assessing the localization of FAD and the interaction between ΔHα and retroB), we developed the recombinant *E. coli* expression system for a catalytically active mini-sMMO (sm1) through tethering the R$_{FAD}$-ΔHα and retroB in parallel to the endpoints (i.e., four C-termini) of each fourfold axis of huHF, where the R$_{FAD}$-ΔHα and retroB are closely interacting within a narrow area (2 nm$^2$), and the sm1 was successfully produced with high yield as a cytoplasmic soluble and active protein (with the turnover number of 0.32 s$^{-1}$ in the presence of NADH), demonstrating the successful reorganization of native sMMO-

derived minimal and modified components ($R_{FAD}$, $\Delta H\alpha$, and retroB) on the apoferritin scaffold. Through systematic spectroscopic and spectrometric analyses ($^{13}$C-NMR, EPR, XANES, EXAFS, and ICP-MS) and azide-diferric complex analysis of the mutated and/or intact sm1, we further investigated the iron valence and quantity in active center, existence and location of FAD, and electron shuttling path in detail, demonstrating that through the minimal model design, the FAD electrons are directly transferred to the diiron center without passing through the ferredoxin domain with keeping active orientation and interaction among $R_{FAD}$, $\Delta H\alpha$, and retroB.

Compared to the traditional methanol production by various methanotrophs (pure culture of *Methylocystis*[51], co-culture of *Methylosinus sporium* and *Methylocella tundrae*[52], etc.), other recombinant expression systems[50,53,54], or MMO-mimetic iron-embedded zeolite catalysts[55–57], the recoverable methanol yield and productivity were improved up to 2.9 g/L and 0.11 g/L/h (22 to 55-fold higher than methanotrophic production), respectively, through the in situ methane oxidation in the high-cell-density culture of mini-sMMO-expressing recombinant *E. coli*. This seems partly due to the fact that, unlike methanotrophs that utilize the methanol produced by MMO as a carbon and energy source, *E. coli* does not have such a methanol oxidation pathway[58], indicating that this approach is a superior way of achieving the highest rate of methanol production from methane in microbial cultures. A more systematic search for the optimal set of important variables (e.g., feeding strategy of fed-batch operation for enhancing intracellular reducing power and methane oxidation rate, bioreactor design to increase both dissolved methane concentration and methanol recovery yield, production scale, etc.) could improve further the methanol-manufacturing performance of the recombinant bacterial culture. We believe that this approach offers a promising platform for miniaturizing functionally valuable but structurally complex enzymes that are difficult to produce as active proteins in *E. coli* and thus for developing a variety of industrially useful biocatalysts that enable the environmentally and energetically benign production of chemicals.

## Methods
### Preparation of genetically engineered recombinant enzymes/proteins

Through polymerase chain reaction (PCR) amplification using appropriate primers, the gene clones were prepared from the coding sequence of huHF encoding the following: $NH_2$-*Nde*I-(His)$_6$-huHF-L(G$_3$SG$_3$TG$_3$SG$_3$)-*Xho*I-*COOH*, $NH_2$-*Nde*I-huHF-L-*Bam*HI-*COOH* and $NH_2$-*Nde*I-huHF-*Hind*III-*COOH*, $NH_2$-*Nde*I-(His)$_6$-huHF-*Hind*III-*COOH*. Furthermore, the additional five gene clones were prepared from *Methylococcus capsulatus* (Bath) genomic DNA through assembly PCR: $NH_2$-*Xho*I-$R_{FAD}$(Cys99-Ala348 of MMOR (PDB ID code: 1TVC, Uniprot: P22868))-*Bam*HI-*COOH*, $NH_2$-*Nde*I-(His)$_6$-$R_{FAD}$-*Bam*HI-*COOH*, $NH_2$-*Bam*HI-$\Delta H\alpha$(Ala64-Leu321 of MMOH (PDB ID code: 4GAM.B, Uniprot: P22869))-*Hind*III-*COOH*, $NH_2$-*Bam*HI-B(Ser2-Ala141 (PDB ID code: 4GAM.D, Uniprot: P18797))-*Hind*III-*COOH* and $NH_2$-*Bam*HI-(retroB)-*Hind*III-*COOH*. Subsequently, each gene clone was inserted into pT7-7 and/or pET28a plasmids to construct expression vectors: pT7-sm1, pET28a-sm1, pT7-sm2, pET28a-sm2, pT7-sm3, pET28a-sm3, pT7-sm$\delta$ and pT7-wt-huHF (Supplementary Fig. 2a–e). Using suitable primers and assembly PCR, some residues in the active site ligands, FAD-binding region, and electron shuttling path of sm1 were mutated, resulting in the following mutated clones: $NH_2$-*Bam*HI-$\Delta H\alpha$(E114A, E144A, H147A, E209A, E243A, H246A of MMOH)-*Hind*III-*COOH*, $NH_2$-*Bam*HI-$\Delta H\alpha$(I118E, L138E, V141T, L142E, I145E of MMOH)-*Hind*III-*COOH*, $NH_2$-*Bam*HI-$\Delta H\alpha$(Y115F, T148A, H149A of MMOH)-*Hind*III-*COOH*, $NH_2$-*Bam*HI-$\Delta H\alpha$(Y115F of MMOH)-*Hind*III-*COOH*, $NH_2$-*Bam*HI-$\Delta H\alpha$(T148A of MMOH)-*Hind*III-*COOH*, and $NH_2$-*Bam*HI-$\Delta H\alpha$(H149A of MMOH)-*Hind*III-*COOH*. Each gene clone was subsequently ligated into pT7-7 and/or pET28a plasmids to construct the expression vectors, pT7-sm1-

cat$_{mut}$, pT7-sm1-FAD$_{mut}$, pT7-sm1-e$_{mut1}$, pT7-sm1-e$_{mut2}$, pT7-sm1-e$_{mut3}$, and pT7-sm1-e$_{mut4}$ (Supplementary Fig. 2f–k). After complete DNA sequencing, the plasmid expression vector(s) were transformed into *E. coli* BL21(DE3) [F-ompThsdSB(rB-mB-)]. To synthesize sm1 to sm3, sm1-cat$_{mut}$, sm1-FAD$_{mut}$, and sm1-e$_{mut1}$ to sm1-e$_{mut4}$, *E. coli* BL21(DE3) was co-transformed with two different expression vectors, as shown in Supplementary Fig. 2, and both ampicillin- and kanamycin-resistant transformants were selected. After transformed with a single expression vector (pT7-sm$\delta$ or pT7-wt-huHF), ampicillin-resistant transformants were selected. The cultivation of recombinant *E. coli* cells and the purification of recombinant proteins are well described in the supplementary methods. In summary, we successfully constructed the plasmid expression vectors (Supplementary Fig. 2) for the recombinant synthesis of sm1, sm2, sm3, sm$\delta$, sm1-cat$_{mut}$, sm1-FAD$_{mut}$, sm1-e$_{mut1}$, sm1-e$_{mut2}$, sm1-e$_{mut3}$, sm1-e$_{mut4}$ and wild-type huHF in *E. coli* and finally selected the antibiotic-resistant transformants.

### Biosynthesis and purification of recombinant enzymes/proteins

The recombinant *E. coli* cells were cultivated at 37 °C using Luria-Bertani (LB) liquid medium supplemented with FeSO$_4$ (0.4 mM) and ampicillin (100 mg/L) for the expression of sm$\delta$ and wild-type huHF or with FeSO$_4$ (0.4 mM), ampicillin (100 mg/L), and kanamycin (60 mg/L) for the expression of sm1 to sm3 and sm1 mutants. When the optical density (absorbance at 600 nm) of the recombinant *E. coli* culture reached 0.5, the isopropyl β-d-1-thiogalactopyranoside (IPTG) (1 mM) was added to the culture for inducing recombinant gene expression. The culture was then further cultivated at 20 °C for 14 h after the IPTG induction, followed by the culture harvest by centrifugation (5311 × *g*, 5 min). The cell pellets were suspended in cell lysis buffer (50 mM NaH$_2$PO$_4$, 300 mM NaCl, 10 mM imidazole, 0.2 mM FeSO$_4$, pH 8.0) and disrupted using Branson Sonifier (Branson Ultrasonics Corp. Danbury, USA) and Sonopuls ultrasonic homogenizer (Bandelin Sonopuls HD 4000, Bandelin Ele., Germany) with 50% output energy (2.0 s pulses/5 s pause) in an ice bath for 2 h. The total cell lysates were then centrifuged (24,562 × *g*, 10 min) to separate the cell-free soluble supernatant from the insoluble fraction containing cell debris and protein aggregates. The soluble recombinant protein in the cell-free supernatant was purified using Ni$^{2+}$-affinity chromatography column (Ni-NTA agarose and column, Qiagen, Hilden, Germany) as follows: (1) prior to sample loading, the column resin was washed with 10 mL of buffer A (50 mM NaH$_2$PO$_4$, 300 mM NaCl, 10 mM imidazole, pH 8.0); (2) following sample loading and binding under batch mode conditions at 4 °C for 1 h, the resin was washed again with 10 mL of buffer B (50 mM NaH$_2$PO$_4$, 300 mM NaCl, 50 mM imidazole, pH 8.0); and (3) the recombinant protein was subsequently eluted with 2 mL of elution buffer (50 mM NaH$_2$PO$_4$, 300 mM NaCl, 250 mM imidazole, pH 8.0), followed by buffer exchange to Tris buffer (20 mM Tris-HCl, 250 mM NaCl, pH 8.0) using ultrafiltration (Amicon Ultra 100K, Millipore). In summary, we established the protocols for synthesizing sm1, sm2, sm3, sm1-cat$_{mut}$, sm1-FAD$_{mut}$, sm1-e$_{mut1}$, sm1-e$_{mut2}$, sm1-e$_{mut3}$, and sm1-e$_{mut4}$ through culturing the recombinant *E. coli* and for purifying the recombinant enzymes/proteins.

### Biosynthesis, purification, and refolding of sm$\delta$

The sm$\delta$ protein was produced by cultivating recombinant *E.coli* cells in an LB liquid medium containing ampicillin (100 mg/L) at 37 °C. After induction with IPTG and separation of the soluble and insoluble fractions of cell lysates, the insoluble protein aggregates were resuspended in denaturation buffer (50 mM NaH$_2$PO$_4$, 300 mM NaCl, 6 M Gdn-HCl, pH 8.0) for 1 h. The denatured and solubilized sm$\delta$ was separated from insoluble cell debris by centrifugation (24,562 × *g*, 20 min) and purified using Ni$^{2+}$-affinity chromatography, where the column resin was washed with 5 mL of the same denaturation buffer prior to the sample loading. After the sample loading and binding in batch mode at 4 °C for 1 h, the resin was washed using 10 mL of buffer

(50 mM NaH$_2$PO$_4$, 300 mM NaCl, 6 M Gdn·HCl, pH 6.3), and the solubilized smδ was eluted using 1 mL of elution buffer A (50 mM NaH$_2$PO$_4$, 300 mM NaCl, 6 M Gdn·HCl, pH 5.9), followed by second elution using 1 mL of elution buffer B (50 mM NaH$_2$PO$_4$, 300 mM NaCl, 6 M Gdn·HCl, pH 4.5). The smδ refolding process involved stepwise dialysis to gradually remove Gdn·HCl from the buffer. After the initial buffer exchange to the buffer (20 mM Tris·HCl, 250 mM NaCl, 1 mM FeSO$_4$, 6 M Gdn·HCl, pH 8.0) for 3 h, six additional dialysis steps were performed with a 1 M decrease in Gdn·HCl concentration at each step (3 h, 4 °C). The final step was performed using Gdn·HCl-free buffer (20 mM Tris·HCl, 250 mM NaCl, 1 mM FeSO$_4$, pH 8.0), which was repeated three times to fully remove the Gdn·HCl. The refolded smδ was recovered after removing protein aggregates by centrifugation (24,562 × $g$, 20 min, 4 °C). As a result, smδ synthesized as an insoluble protein in *E. coli* was successfully refolded to a soluble protein.

### Analysis of expression level and cytoplasmic solubility of mini-sMMOs and smδ

The expression level and cytoplasmic solubility of mini-sMMOs and smδ were analyzed through sodium dodecyl sulfate-polyacrylamide gel electrophoresis (SDS-PAGE). Total cell lysates, soluble and insoluble fractions, purified mini-sMMOs, and refolded smδ were loaded onto 12% SDS-PAGE gels and analyzed. Coomassie-stained protein bands were scanned and analyzed using a densitometer (GS-800 Calibrated Densitometer, Bio-Rad, California, USA) with Quantity One v4.6.9 software to estimate the expression level (%) in total cellular proteins and the soluble to insoluble fraction ratio of each recombinant protein.

### Characterization of mini-sMMOs (sm1 to sm3) and sm1 mutants

The morphological shapes of mini-sMMOs and sm1 mutants were analyzed using transmission electron microscope (TEM), operating at 200 kV (FEI, Hillsboro, Oreon, USA). TEM samples were prepared by placing one drop of colloidal solution onto 200 square mesh copper grids with carbon film (Electron Microscopy Science, Pennsylvania, USA), followed by negative staining using 2% (w/v) uranyl acetate solution and air drying for 1 h. TEM images were obtained using a charge-coupled device (CCD) camera and FEI-imaging software (Gatan Digital Micrograph v3.9.4, Pleasanton, USA) installed in the Tecnai 20.

The hydrodynamic size of mini-sMMOs and sm1 mutants was also analyzed through dynamic light scattering (DLS) analysis using an ELSZ-1000 Zeta Potential & Particle Size Analyzer (Otsuka Electronics Co., Osaka, Japan) with Photal ELSZ-1000 v5.01 software.

The iron content of sm1, sm1-cat$_{mut}$, sm1-e$_{mut1}$, and commercial standards of holo- and apoferritin (from equine spleen, Sigma-Aldrich, St. Louis, USA) was measured using inductively coupled plasma mass spectrometer (ICP-MS) (NexION 350D, Perkin-Elmer, USA) at National Center for Inter-university Research Facilities (NCIRF) at Seoul National University.

The stoichiometry of α- and β-subunits comprising each mini-sMMO (sm1, sm2, or sm3) was estimated by measuring the Coomassie-stained band intensity (optical density/mm2) of α- and β-subunit of each purified mini-sMMO loaded on SDS-PAGE gel using a GS-800 calibrated densitometer (Bio-Rad) and Quantity One v4.6.9 software (Bio-Rad). The relative % of the measured intensity of α- and β-subunit band was used to calculate the stoichiometry, i.e., the fraction of each subunit out of a total of 24 subunits comprising mini-sMMO.

To confirm whether the R$_{FAD}$-ΔHα and retroB in sm1 are localized on the huHF scaffold surface in the sm1, Western blot analysis of purified sm1 and wild-type huHF were performed after Native-PAGE. The separated proteins in the Native-PAGE gel were transferred to a nitrocellulose membrane. The membrane was blocked with 5% skim milk in Tris-buffered saline with Tween 20 (TBST) at 4 °C for 16 h, followed by incubation with 1000-fold diluted mouse anti-huHF IgG (ab77127, Abcam, Cambridge, UK) as primary antibody at 4 °C for 16 h.

Then the membrane was washed three times (15 min each) and then incubated with 1:1000 diluted horseradish peroxidase-conjugated goat anti-mouse secondary antibody (Catalog no. 31430, Pierce, Rockford, IL, USA) in TBST for 2 h. After drained by TBST, the membrane was reacted with enhanced chemiluminescence (ECL) solution (Catalog no. BWD0100, Biomax, Seoul, Korea). The chemiluminescent band was visualized using a Biomolecular Imaging System (LAS 3000M, Fuji Film, Tokyo, Japan) and Image Processing Software (Multi Gauge v2.3, Fuji Film, Tokyo, Japan).

The secondary structure of sm1, sm1-cat$_{mut}$, sm1-FAD$_{mut}$, and sm1-e$_{mut1}$ in reaction buffer (20 mM Tris·HCl, 250 mM NaCl, pH 8.0) was analyzed using Jasco J-1100 circular dichroism (CD) spectrometer (Jasco, Tokyo, Japan). The parameters for far-ultra-violet CD spectra were monitored from 180 to 250 nm. The percentage of each secondary structure (α-helix, β-sheet, turn, and random coil) involved in samples was calculated using the BeStSel program.

In summary, with mini-sMMOs (sm1 to sm3) and/or various sm1 mutants (sm1-cat$_{mut}$, sm1-FAD$_{mut}$, sm1-e$_{mut1}$, sm1-e$_{mut2}$, sm1-e$_{mut3}$, and sm1-e$_{mut4}$), their characteristics—morphological shape, hydrodynamic size, iron content, stoichiometry of α- and β-subunits, localization of R$_{FAD}$-ΔHα and retroB on apoferritin scaffold, and secondary structure—were analyzed using appropriate analytical tools.

### Xylenol Orange assay

The assay is conducted at 30 °C by adding Fe$^{2+}$ to a mixture of protein/enzyme (0.8 uM), Tris·HCl buffer (50 mM, pH 7.0), and (NH$_4$)$_2$Fe(SO$_4$)$_2$ (40 μM). A 10-μl aliquot is taken at 5 min after the assay reaction started and mixed with a 100-μL solution of Xylenol Orange (XO) (125 μM) and H$_2$SO$_4$ (25 mM). After incubation at room temperature for 5 min, the concentration of ferric ions is determined by measuring the absorbance at 595 nm using a TECAN microplate reader (Infinite M200 Pro, TECAN, Zürich, Switzerland) with i-control v1.7 software. The auto-oxidation of the ferrous ions is measured under the same assay conditions but without adding protein/enzyme.

### In vitro catalytic oxidation of methane to methanol

In vitro enzymatic oxidation of methane to methanol was performed using NADH as a reducing agent. About 1 mL solution of the purified recombinant protein (sm1 to sm3, smδ, wild-type huHF or sm1 mutants) containing NADH (0.3 mM) was added to a septa-sealed vial (20 mL) (Catalog no. 5182-0837, Agilent (California, USA)). The concentration of each recombinant enzyme above was determined to make the number of catalytic sites be equal as follows: 8.4 μM of smδ, 1.2 μM of sm1 to sm3, and wild-type huHF. Enzymatic oxidation was initiated by replacing 19 mL of headspace air with 15 mL of methane and 4 mL of air using a syringe, followed by incubation in a shaking incubator (30 °C, 180 rpm). The amount of methanol produced by the enzyme reaction was measured to estimate cumulative production (mol methanol per mol enzyme) and turnover frequency (TF, s$^{-1}$). TF was estimated for an initial 15 min after methane oxidation began. At least three vials were used at the time of each measurement.

### Whole cell-based methane oxidation in a closed vial system containing recombinant *E. coli* culture

About 3 mL of reaction solution containing sm1-overexpressing *E.coli* cells (OD$_{600}$ = 40), 5% (w/v) glucose, and LB medium with 100 mM phosphate buffer, were added to a 20 mL septa-sealed vial. To enhance the dissolved methane and air concentration, the reaction solution was pressurized by filling the headspace with 15 mL of methane/$^{13}$C-methane and 10 mL of air using a syringe, and the vials were placed in a shaking incubator (30 °C, 180 rpm) to initiate the methane oxidation.

### Analytical methods to measure the amount of methanol

**Analysis of methanol production in a closed vial system.** Prior to the methanol analysis at each time point, the septa-sealed vial (20 mL) was

placed in an oven at 70 °C for 15 min to totally evaporate the methanol produced in the reaction solution into the gas phase of headspace. Then, 1 mL of headspace gas was injected into a Supel-Q PLOT capillary gas chromatography (GC) column (Supelco, Pennsylvania, USA) installed in a GC equipment (7890B GC, Agilent, California, USA). The amount of methanol was quantified using a pre-determined correlation between the mass of commercial methanol standard (Sigma-Aldrich, St. Louis, USA) and calculated area of methanol peak using a GC software (OpenLAB CDS ChemStation C.01.07, Agilent).

**Analysis of in situ methanol production in fed-batch cultures of sm1-expressing *E. coli*.** About 1 mL of each culture sample was centrifuged at 4 °C, 2038 × $g$ for 10 min, and immediately 0.5 mL of supernatant solution was transferred to a 2-mL screw vial (parts number: 5182-0715, Agilent, California, USA) after filtering with PVDF syringe filter, followed by HPLC (1260 Infinity II, Agilent, California, USA) analysis. $NH_2SO_4$ solution (0.01 M) was used as the mobile phase after filtering and degassing for 30 min. 300 × 7.7 mm Hi-Plex H column (PL1170-6830, Agilent, California, USA) was heated to 70 °C at thermostats (1260 MCT, Agilent, California, USA) for analysis. The sample solution ran through the HPLC system at the rate of 0.7 mL/min powered by a quaternary pump (1260 Infinity II, Agilent, California, USA). A refractive index detector (1260 Infinity II, Agilent, California, USA) was used, and the detector was purged with the mobile phase solution for an hour and heated to 55 °C prior to analysis. Each sample was analyzed for 25 min after the HPLC system was stabilized, and the area of the methanol peak was calculated using HPLC software (OpenLAB CDS ChemStation v2.6, Agilent).

In summary, methanol production in closed vial systems and fed-batch cultures of sm1-expressing *E. coli* was monitored using GC and HPLC, respectively.

**Measurement of intracellular NADH concentration**
Intracellular NADH concentration in sm1-expressing *E.coli* cells was measured using the NAD⁺/NADH quantification kit (Sigma-Aldrich, MAK037) as per the manufacturer's instructions. For each sample, 1 mL of cell culture ($OD_{600}$ = 40) was collected and centrifuged (2038 × $g$, 10 min) in an Eppendorf tube. The cell pellet was immediately washed three times with cold PBS (2038 × $g$, 10 min) each time. To extract the NAD⁺/NADH from the centrifuged cells, the cell pellet was homogenized in 1 mL of extraction buffer, followed by centrifugation (2038 × $g$, 10 min). To detect total NADH and NAD⁺, the supernatant is eightfold diluted with extraction buffer, resulting in a total volume of 50 μL. About 50 μL of the diluted supernatant is transferred to a 96-well plate (Catalog no. 3599, Costar, NY, USA) followed by incubation with master reaction mix (98 μL of NAD Cycling Buffer and 2 μL of NAD Cycling Enzyme Mix) for 5 min at room temperature. Then, 10 μL of NADH developer is added to each well, and absorbance is measured at 450 nm at different time points. To detect NADH, 200 μL of the supernatant was incubated in a water bath (60 °C, 30 min), followed by centrifugation (17,005 × $g$, 10 min). About 50 μL of the NAD decomposed supernatant is transferred to a 96-well plate (Catalog no. 3599, Costar, NY, USA) followed by incubation with master reaction mix (98 μL of NAD Cycling Buffer and 2 μL of NAD Cycling Enzyme Mix) for 5 min at room temperature. Then, 10 μL of NADH developer is added to each well, and absorbance is measured at 450 nm at different time points. Absorbance is measured using a TECAN microplate reader (Infinite M200 Pro, TECAN, Zürich, Switzerland) with i-control v1.7 software.

**Fluorometric FAD assay of sm1 and sm1 mutants**
FAD quantification kit (FAD Colorimetric/Fluorometric assay kit (K357-100, BioVision, Waltham, USA)) was used to estimate the FAD content of sm1 and its mutants according to the manufacturer's protocol. The sm1 and its mutants at the same concentration (2.4 μM) were added into each well of a 96-well plate (Catalog no. 3599, Costar, NY, USA)

with the reaction mixture. Fluorescence intensity was measured at Ex/Em = 535/587 nm using a TECAN microplate reader (Infinite M200 Pro, TECAN, Zürich, Switzerland) with i-control v1.7 software up to 1 h at intervals of 15 min after the reaction started.

**Azide-diferric complex analysis**
To investigate the formation of a chromophoric complex of azide and oxo-bridged diiron clusters, 2 M buffered sodium azide was added to the enzyme (sm1-afr and sm1-$cat_{mut}$) at 1.5 μM, resulting in absorption bands at 345 and 450 nm. Absorbance is measured using a TECAN microplate reader (Infinite M200 Pro, TECAN, Zürich, Switzerland) with i-control v1.7 software.

**X-ray absorption spectroscopy and EPR analyses**
XANES and EXAFS analyses were performed by Aichi Synchrotron Radiation Center (Aichi, Japan) with the sm1-afr sample, prepared after the methane oxidation reaction for 16 h in the presence of NADH (0.3 mM) (sm1-afr). The sm1-afr was prepared as dried samples using freeze-dryer (FDU-2100, DRC-1000, EYELA, Tokyo, Japan): the tris buffer (20 mM Tris-HCl, 250 mM NaCl, pH 8.0) solution containing the sm1-afr was pre-frozen at −80 °C for 3 h, and the frozen sample was lyophilized at −80 °C using the freeze-dryer above. The X-ray absorption spectra of the dried sm1-afr were recorded with a hard X-ray XAFS technique at room temperature at the XAFS beam line (BL11S2) of Aichi Synchrotron Radiation Center (Aichi, Japan). The collected data were processed with the software *Athena*[59]. Especially, the magnitude of radial distances was obtained through the phase-corrected Fourier transformation. EXAFS spectra was fitted by calculating the theoretical EXAFS spectrum using *Artemis*[59] software. The model for MMOH-MMOB complex was derived from the PDB crystal structures 4GAM, selecting all atoms within 5 Å of diiron active site through the use of PyMOL v2. Individual scattering paths were selected and fitted within *Artemis* from paths calculated by FEFF6. The Fourier-transform spectrum was fitted over a range of $R$ = 1.0 to 3.5 Å (non-phase shift corrected).

Further, EPR analysis was performed with sm1-afr. All EPR measurements were carried out at the Metropolitan Seoul (Western Seoul) center, Korea Basic Science Institute (KBSI) in Seoul, Korea. CW X-band (9.6 GHz) EPR spectra were collected on a Bruker EMX plus 6/1 spectrometer equipped with an Oxford Instrument ESR900 liquid He cryostat using an Oxford ITC 503 temperature controller. All spectra were collected with the following experimental parameters: microwave frequency, 9.6 GHz; microwave power, 1 mW; modulation frequency, 100 kHz; modulation amplitude, 10 G; time constant, 40.96 ms; 2 scans; and temperature, 5 K. In summary, XANES, EXAFS, and EPR (5 K) analyses were performed with the freeze-dried sample of sm1-afr.

**NMR spectroscopy analysis**
¹³C-methane oxidation by the purified sm1 or sm1-overexpressing *E. coli* cells was performed in 20 mL septa-sealed vials containing reaction solution. The produced ¹³C-methanol was totally evaporated to headspace by placing the vials in an oven at 70 °C for 15 min. The headspace gas, containing the evaporated ¹³C-methanol and residual ¹³C-methane, was collected using syringe and injected into 1 mL ethanol. The same procedure above was repeated for the vials containing the enzyme- or cell-free reaction buffer (enzyme-free reaction buffer: 20 mM Tris-HCl, 250 mM NaCl, pH 8.0; cell-free reaction buffer: 5% (w/v) glucose, LB medium with 100 mM phosphate, pH 8.0) and also for the vials containing the same buffer with standard ¹³C-methanol (Sigma-Aldrich, St. Louis, USA) or standard ¹³C-methane (Sigma-Aldrich, St. Louis, USA). The ethanol solution (600 μL) containing the collected ¹³C-methanol and ¹³C-methane was mixed with ethanol-d₆ (60 μL), and the mixture was used as a sample for ¹³C-nuclear magnetic resonance (NMR) spectroscopy (Bruker Ascend 800 MHz NMR

spectrometers equipped with cryogenic z-gradient triple resonance probe) analysis at Korea Institute of Science and Technology (KIST). $^{13}$C-NMR spectra were processed and analyzed using Topspin 3.6.3 (Bruker) software. As a result, $^{13}$C-methanol produced from the $^{13}$C-methane oxidation by the purified sm1 or sm1-overexpressing *E. coli* cells was successfully detected through the $^{13}$C-NMR analysis described above.

### In situ methane oxidation in the high-cell-density cultures of sm1-expressing *E. coli* under fed-batch operation

The first seed culture (30 mL) of sm1-expressing *E. coli* were cultivated in a shaking incubator (VS-8480SR, Visionbionex, Republic of Korea, 230 rpm) at 37 °C for 10 h using LB medium containing ampicillin (100 mg/L) and kanamycin (60 mg/L), which was then inoculated to 300 mL LB medium and cultivated for 2.5 h at the same condition for preparing the second seed culture. Then 300 mL of the seed culture was inoculated to 3-L fresh medium (30 g/L yeast extract (Lot no. 016784.02, Duchefa Biochemie, Haarlem, Netherlands), 5 g/L glucose, 2.33 g/L MgSO$_4$·7H$_2$O, 20 mM phosphate (1.047 g/L KH$_2$PO$_4$, 1.047 g/L (NH$_4$)$_2$HPO$_4$, 0.761 g/L K$_2$HPO$_4$), 0.4 mM FeSO$_4$, 100 mg/L ampicillin, and 60 mg/L kanamycin) in 5-L jar fermenter (Marado-PDA, Biocns, Republic of Korea), followed by initial batch and fed-batch cultivation at 30 °C (phase 1), sm1 expression (1 mM IPTG) at 20 °C for 12 h in the fed-batch culture (phase 2), and methane oxidation at 30 °C with the supply of methane-air mixture (30% methane) in the same fed-batch culture (phase 3). The pH-stat-based fed-batch operation began when glucose was exhausted in the initial batch culture, with the feed rate of 2.5 mL/min using the feed medium containing 600 g/L glucose. The culture pH and dissolved oxygen (DO) were monitored using a pH electrode (F-695 Autoclavable pH Electrode, Broadley-James, Irvine, USA) and DO sensor (InPro6820/12/220, Mettler-Toledo, Urdolf, Switzerland). pH was controlled between 6.80 and 6.90 before phase 3 and between 7.90 and 8.10 during phase 3 using 15% ammonia solution, and the aeration rate was fixed at 1.5 L/min to maintain more than 30% DO level. A vertical condenser was installed on the fermenter to minimize the methanol loss. Antifoam SE-15 (Sigma-Aldrich, Germany) was used to minimize foaming in the fermenter. In summary, the operating procedures and conditions—seed culture preparation and inoculation, medium composition for initial batch and subsequent fed-batch cultures, and feeding strategy and operating conditions of fed-batch cultures—for in situ methanol production in the high-cell-density cultures of sm1-expressing *E. coli* were described above in detail.

### Computational details

All-atom atomistic molecular dynamics (MD) simulations were performed for model sMMO and mini-sMMO. The components of each model, including MMOHα, R$_{FAD}$, ΔHα, B, retroB, were prepared using initial structures obtained from the protein data bank (PDB ID codes: 1MTY, 4GAM, 1TVC)[28,60,61]. Our computational approach aimed at designing a regulatory protein for R$_{FAD}$-ΔHα, so that the binding characteristics observed in the native MMOH-MMOB complex system can be reproduced. We selected the MMOB structure of the native MMOB-MMOH complex (PDB 4GAM) as the initial framework for simulating retroB. That is, we constructed retroB structure model through assigning the N- and C-terminus of MMOB of 4GAM structure to the C- and N-terminus of retroB, respectively, which retains the reversed sequence of MMOB. Then, we evaluated the stability of retroB upon the binding with MMOH or R$_{FAD}$-ΔHα through MD simulation (RMSD and contact map analysis). Protein–protein docking (PPD) simulation for preparing the initial structure of R$_{FAD}$-ΔHα was carried out using ZDOCK server[62] with 6° rotational sampling, where shape complementarity, desolvation and electrostatic energy terms were considered in the ranking of the 2000 highest-scoring poses. High-scoring poses were filtered by constraining the distance between the C-terminus of R$_{FAD}$ and the N-terminus of ΔHα to within 10 Å. This

constraint was introduced to consider only configurations of R$_{FAD}$ linked to ΔHα by forming a single chain with the N-to-C orientation from R$_{FAD}$ to ΔHα, which allowed for stable R$_{FAD}$ binding to the NTH region by reducing the translational and rotational freedom of the proteins. The conformation samples of R$_{FAD}$-ΔHα chains were derived from the ten highest-scoring poses.

The R$_{FAD}$-ΔHα was then relaxed using an energy minimization process followed by isobaric-isothermal MD equilibration in a water medium with 100 mM NaCl in the simulation box with a periodic boundary condition at 300 K and 1 bar, using NAMD2 program[63] with CHARMm force field. The temperature and the pressure were maintained using the Langevin piston Nose-Hoover method[64]. A full system of periodic electrostatics was employed by the particle-mesh Ewald method with a 1 Å grid spacing[65]. The cutoff and switching distance for van der Waals force were set to be 12 and 10 Å, respectively. The bonds involving hydrogen were constrained to be rigid by using the SHAKE algorithm[66]. The MD system was equilibrated for 200 ns with a 2-fs time step, and the structure analysis was done for simulating a further 500 ns run with recording every 250 ps. The initial complex structures of MMOH/B, R$_{FAD}$-ΔHα/B, and R$_{FAD}$-ΔHα/retroB were prepared using the B structure from PDB file 4GAM and the previously derived R$_{FAD}$-ΔHα, where the initial bound forms were chosen to retain hydroxylase/regulator contacts as in the native MMOH/B. Structures for the regulator-bound samples were then relaxed through energy minimization and subsequently subjected to isobaric-isothermal MD equilibration, following the procedure previously described.

### Statistics and reproducibility

Unless otherwise stated in the text, data were shown as the mean ± standard deviation (s.d.). The number of independent biological repeats and technical replicates (*N*) are indicated in the text and figure legends. We confirmed a statistically significant difference in iron contents between samples (sm1-cat$_{mut}$, sm1-FAD$_{mut}$, sm1-e$_{mut1}$, Supplementary Fig. 15) using Student's *t*-test. No data were excluded from the analyses. The investigators were not blinded to allocation during experiments and outcome assessment.

### Reporting summary

Further information on research design is available in the Nature Portfolio Reporting Summary linked to this article.

## Data availability

The structural data for MMOR, MMOH, MMOB, and MMOH-MMOB complex are available under PDB accession numbers 1JQ4, 1TVC, 1MTY, and 4GAM, respectively. A sequence of MMOR, MMOH, and MMOB are available in UniProt databases under accession numbers P22868, P22869, and P18797, respectively. The molecular dynamics simulation data of R$_{FAD}$-ΔHα generated in this study is provided in Supplementary Movie 1. Authors can confirm that all relevant data are included in the paper and/or its Supplementary Information files. Source data are provided with this paper.

## Code availability

There are no custom codes or mathematical algorithms central to the results. The publicly available XAFS analysis code DEMETER can be found here: https://bruceravel.github.io/demeter/.

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

## Acknowledgements

This study was supported by grants from the National Research Foundation of Korea (NRF) funded by the Korean government (2019R1A2C3005771 and 2020M3D3A1A01080557 to J.L., Y.Y., Y.C., and Y.K.).

## Author contributions

J.L. conceived the development of mini-sMMOs, conducted the data analysis, and wrote the paper; Y.Y. performed the synthesis and activity analyses of the mini-sMMOs and collected the data; J.H. performed the in silico analyses and co-wrote the paper; Y.S. and J.-G.N. performed the fed-batch bioreactor operation for in situ methane oxidation; Y.W.K. performed EPR analysis and collected the data; Y.C. and Y.K. analyzed the experimental data.

## Competing interests

The authors declare no competing interests.
