## [Peer Review File · Nature Communications]

A rationally designed miniature of soluble methane monooxygenase enables rapid and high-yield methanol production in *Escherichia coli*REVIEWER COMMENTS

Reviewer #1 (Remarks to the Author):

Major comments:

This manuscript describes a generally well conducted study that constitutes a substantial breakthrough in expression of a methane-oxidising biocatalyst in *Escherichia coli*. The soluble methane monooxygenases (sMMOs) produced by aerobic methane-oxidising bacteria have proven very difficult to express in active form in heterologous systems. This study appears to have achieved this via a reductionist and synthetic biology approach. The manuscript describes elegant computational design of a new protein complex based on sMMO, substantial biophysical characterisation and small- and larger-scale biotransformations of methane to methanol.

My only substantial concern with the work is for the Authors to be clearer that all relevant controls have been done to establish methane oxidation activity by the protein complex they have made. With the pure protein work, it needs to be made clear whether the buffer controls contained all non-protein components of the reaction (including NADH, where commercial preparations sometimes contain volatiles that confound monooxygenase assays) and that the reaction was shown to be dependent on added NADH via no-NADH controls. In the whole-cell biotransformation work some more information would be helpful to be certain that methane is the source of methanol produced and that the carbon-13 methanol observed could not be due to natural abundance of carbon-13 in methanol produced from some source other than the added carbon-13 labelled methane.

In describing and discussing the new biocatalysts constructed, the Authors need to be careful to distinguish the properties that are experimentally verified in their work from those (such as many aspects of the three dimensional structure of the protein and the dependence of the enzyme on NADH in whole *E. coli* cells) which are assumed from *in silico* modelling without direct experimental verification.

Much of the discussion section is a recapitulation of the results and could be shortened. It is useful to evaluate the productivity of the new recombinant system for methane to methanol conversions in comparison with methanotroph-based systems, as the Authors do. It would also be relevant to discuss the extent to which the increased yield of methanol in the recombinant *E. coli* may be due to the absence of the methanol oxidation pathway that is present in methanotrophs and under most conditions consumes much of the methanol produced by methane monooxygenase.

Minor specific comments:

1. Page 1 (Abstract) “with demonstrating” would be better as “as well as demonstrating”.
2. Page 1 (Abstract). The usefulness of the abstract would be increased by including quantitative information about the turnover numbers and methanol yields achieved in the work.
3. Page 2. In describing the distribution of the particulate and soluble methane monooxygenases, the Authors should mention that some methanotrophs, such as the facultative methanotrophs of the genus *Methylocella*, possess the soluble enzyme but not the particulate.
4. Page 2 (end of first paragraph). “timely” would be better as “coupled”. “the best known being NADH” would be better as “NADH or NADPH”.
5. Page 2 (second paragraph). “seems still incomplete” would be better as “is still incomplete”. Also, consider replacing “many plausible” with “a number of possible”.
6. Later in the same paragraph. The lack of activity of recombinant pMMO cannot be solely attributed to its inability to receive electrons from NADH.
7. Next to last line of page 2. Fast reaction kinetics and other enzymological methods have been more important in establishing the mechanism of sMMO than crystallography and NMR.
8. First line of page 3. “enzymes” would be better as “components”.
9. Page 3, last sentence of first paragraph. Recombinant expression of sMMO and mutants of it has been achieved by homologous expression in methanotroph cells. It is the problem of heterologous expression in a genetically easily tractable host that had not previously been achieved.
10. Page 4. First sentence of last paragraph. It needs to be made clear what reaction of MMOH alone is being referred to here, presumably its activation by hydrogen peroxide in the peroxide shunt reaction.
11. Page 5. “slimed” should be “slimmed”.
12. Page 6. Second sentence of second paragraph. “inverted” here suggests that the sequence was reversed from the natural protein B, which clearly is not what was done. I presume some kind of domain inversion was done. A clearer explanation is needed and sufficient details for the reader to understand exactly what the construct was.
13. Page 6-7. The sentence that crosses between the pages here is one of the instances where it needs to be made clear that these are results from in silico modelling rather than experiment.
14. Page 8. Second sentence of second paragraph. There is some overinterpretation of the data in Fig. S3a-c here, which shows growth curves of recombinant *E. coli* and does not give information about the iron occupancy of the enzyme active sites.
15. Page 8. Next to last sentence. It is not clear how the SDS-PAGE shown in Fig. S4a shows the stoichiometry of the mini-sMMO complexes. An explanation is needed.
16. Page 8. Last line. “consistent with” would be more accurate than “indicating that”.

17. Bottom of page 8/top of page 9. Some more explanation is needed of how the absence of cross-reactivity with the antibodies shows the stated “flipped” conformation (maybe because the epitopes that the antibodies recognise are moved to the inside of the complex?).

18. Page 10. More explanation is needed for the “bfr” and “afr” samples. Exact conditions for how they were prepared is needed (ideally as a reference to the appropriate part of the Methods) and it needs to be clearly indicated why it is expected that the two forms should have iron in the different oxidations states (maybe because “bfr” has had the enzyme incubated with NADH in the absence of methane?).

19. Later on page 10, it is an overinterpretation of the results to say that the catalytic cycle is shown to be “exactly the same” as native MMOH. It would be reasonable to say that the results are consistent with a similar catalytic cycle.

20. Page 11. Last three lines of the paragraph that starts “Identification of electron shuttling path”. “elucidate” would be better as “suggest” or “indicate”. Consider replacing “looking more important” with “being more important for methanol production”.

21. Page 12. “we confirmed that methanol production was due to the catalytic activity of sm1” – the data in the figure cited (Fig. S11) do not show this, but (subject to the major comment above) that methane is the source of the methanol produced.

22. Later in the same paragraph. “the recombinant sm1 expression continued” – this suggests continued production of the protein, whereas the data indicate the amount remained roughly constant. Hence, “the recombinant sm1 protein remained” may be better here.

23. It may be worthy of mention in the Results or discussion that the main peak of intracellular NADH seen in Figure 6 is substantially before methanol production occurs, though there is a single elevated point that suggests a second smaller peak in NADH around the time of maximum methanol production (Fig. 6 b and c).

24. Throughout the methods, it would be better to use g (acceleration due to gravity) than rpm when specifying centrifugation conditions.

25. Page 13. Please provide full details of the cell breakage conditions (including duration, power and temperature of sonication), or a reference to where these can be found.

26. Fig. 3a and Fig. S9. TEM images and associated size distribution analysis of the scaffold protein without the sMMO components are needed.

27. Fig. 6 b and c. It is not clear why error bars are shown for only some of the data. Numbers of repeats, and whether the multiple measurements are technical or biological replicates, needs to be stated in the legend.

Reviewer #2 (Remarks to the Author):

The authors use MD computations to guide the construction of a severely contracted version of sMMOH using only a portion of the alpha subunit and then connect it to a truncated version of the MMOR reductase with the FAD domain moved to facilitate electron transfer to the active site of the modified MMOH. This is connected to apo human ferritin and an sequence inverted and truncated MMOB is connected nearby. The connection to ferritin pushes the “mini-sMMO” and inverted MMOB to the outside and allow expression in *E. coli*. Amazingly, this appears to allow conversion of methane to methanol in fairly high yields for the first time outside of a methanotroph.

If the production of methanol is correct from this completely disassembled and reconstructed mini-sMMO, then this is a remarkable breakthrough in enzyme engineering. The observation of methane to methanol conversion by both purified construct and in whole cell *E. coli* is most strongly supported by the use of $^{13}\text{CH}_4$ to produce $^{13}\text{CH}_3\text{OH}$, thereby eliminating artifacts from unlabeled methanol contamination, which is common.

I think the rates of production of methanol have been miscalculated or overstated in the text (see below), but any continuous production from a system redesigned in silico and produced in *E. coli* is unprecedented and worth publication. The problems with the study are in the details of the analysis of yields and the attempts to characterize the detailed structure of the new enzyme and its metal center. It is not clear that the characterization of the metal center is necessary for publication. However, it would be an excellent aspect of the study if correctly accomplished, and it would better connect the work to the extensive characterization of the diiron center and its catalytic mechanism in the sMMO system.

The authors present no actual structural characterization of the mini-MMO construct or evidence that a diiron cluster is actually formed. Everything is based on a computation of a massively modified and reconstructed (and even inverted in the case of MMOB) over-expressed protein complex. Some spectroscopic techniques are used to identify oxidation states of the presumed diiron cluster, but the way they are analyzed is either incorrect (see EPR comments below) or incomplete (no fits or quantification for the EXAFS, for example). The preparation of the samples for XANES and EXAFS by freeze drying the enzyme after 20 hours of reaction and expecting to get an accurate oxidized to reduced ratio is overly optimistic. It is not clear in any case how partial oxidation of the presumed diiron cluster in the construct is indicative of either MMO-like catalysis or a diiron cluster. Simple extensions like attempting to make a mixed valent cluster, which does have a characteristic EPR spectrum, or adding azide to the construct to attempt to detect the characteristic chromophore were not attempted. If enough enzyme for XANES can be obtained, then it should be possible to cheaply reconstitute with ^{57}Fe for definitive identification of the diiron cluster and its oxidation state by Mössbauer. Of course, this would also reveal whether only one type of iron center is present.

Significant effort was devoted to making mutants of residues predicted to be part of the diiron cluster ligation, FAD binding site, and electron transfer pathway. Changes were observed in iron content of the sample and/or methanol production, but all of the changes could be attributed to changes in overall structure caused by the mutations. Without a structure, or at least an attempt to show that the mutations don't change the structure, it is all guesswork.

Abstract: The first sentence of the abstract is 6 lines long, while the second sentence is 5 lines long. These are too complex for easy comprehension.

Line 30 (first line) I think it is misleading to start out by saying that methanotrophs consume atmospheric methane. As shown in the Curry reference, they are a very small contributor to methane consumption once it enters the atmosphere. They are a large contributor to consuming dissolved methane produced by biological and geological processes.

Line 34 "convert a wide range of carbon feedstocks to methanol" this needs to be rewritten for clarity. Only methane is converted to methanol.

Lines 37-42. This sentence is too long and complex

Line 42 "intrude an oxygen radical" Intrude is not used correctly and the reactive species is probably not an oxygen radical. It might be a hydroxyl radical if the proposed rebound mechanism is correct (ie. Formation of a carbon radical by hydrogen atom abstraction. followed by rebound of the resulting hydroxyl radical.)

Line 53 "owing to some seminal studies based on crystallographic and NMR investigations" While these studies contributed to understanding the mechanism they are far from the complete story. There are much more significant studies that actually address mechanism rather than simply structure. The structure references given are also very incomplete.

Line 54 and 55 and other places later in the manuscript "three different enzymes" only two of the components are enzymes

Lines 60-62 "(MMOB) that plays two simultaneous roles" The knowledge of the roles of MMOB began much earlier than the references provided and they have come much further since the referenced papers appeared. A much better knowledge and communication of the state of the knowledge in this

field will be necessary for the reader. The manner in which MMOB and MMOH interact is extremely complex and involves binding and release of MMOB from MMOH in concert with binding and release of MMOR. This makes it all the more remarkable that a fixed construct of MMOH-MMOR with an inverted MMOB could be reactive.

Introduction: In general, the introduction omits a great deal of insight from the recent literature. I think it would benefit from extensive rewriting, both to improve English presentation and content. In contrast, the English in the rest of the manuscript is much better.

Line 214 and on “Notably, approximately 2,000 moles of methanol were produced per mole of sm1 within 20 h” and “The turnover frequency (TF) of sm1 was about 0.32 s⁻¹” The computation here needs to be clarified for the reader. 20 h is 72000 s. 2000 turnovers in 72000 seconds is 0.03 s⁻¹ or 10 fold lower than claimed in Figure S8.

Line 235 “The EPR spectra (Figs. 4a,b) show that the ratio of Fe(II) to Fe(III) measured for sm1-bfr and sm1-afr is 1.91 and 0.81, respectively, which is very similar to that of native sMMO” Neither oxidized nor reduced MMOH have EPR spectra in the g = 4 to 1.8 region shown in the oxidized or reduced forms. Both forms are spin coupled with total net spin of 0 in the diferric state and 4 for the diferrous state. The spectra shown are mostly of a copper contaminant. They have no relationship to native MMOH spectra. The reduced diiron cluster of MMOH does have a distinctive EPR resonance at g = 16 but it cannot be quantified by simple double integration because it derives from an integer spin system. This region is not shown in the figure. Also, 77K is too high a temperature for accurate investigation of the EPR spectra of a diiron cluster. As stated above, the significance of the oxidation state of the presumed diiron cluster after 20 h of reaction is not clear.

Line 238 on. The XANES spectra are at best unconvincing that the observed iron is related to a diiron cluster in the enzyme. Also the statement on line 239-240: “The significantly higher portion of Fe(III) in sm1-afr demonstrates that the diiron center of sm1 follows exactly the same catalytic cycle as that of native MMOH_α.” claims much more than the data supports.

Figure 4d and 4e - The EXAFS is presented without any fits, quantification of N, O, Fe, DW factors for different trial fits, or other evidence that the assigned distances correspond to what it claimed.

Line 290 “methanol production by methanotrophs (0.002 to 0.005 g/L/h)” I believe this is severely underestimated. The referenced papers are using formate as a source of energy and as a means presumably to slow methanol to formate conversion in the culture. Blocking this conversion is the only way any methanol would be observed in solution. This is not representative of methanol production by freely growing methanotrophs. The literature reports 25 g/L of cell paste during an 8 h growth of

methanotrophs on methane in a fermentor at an OD600 of 8. All of the carbon would be derived from CH₄ (and subsequently methanol) so if 25% of the cell paste was carbon, this gives 6.25g/L divided by 8h = 0.78 g/L/h. This being compare to methanol production from an OD600 = 40 E. coli culture, so the value for native methanotrophs could be has high as 3.9g/L/hr at an equivalent cell mass. The native fermentation is run by dilution and regrowth of the bacteria during the 8 h period, so only at the end of each fermentation period is the maximal methanol per hour being produced. Consequently, the estimated 0.78 g/L/h average would be much lower than the actual rate of production. Since the turnover number for isolated MMOH/MMOB/MMOR is about 2.5 s⁻¹ at room temperature, and the recalculated value for the isolated mini-MMO is 0.03 s⁻¹, it seems unlikely that in whole cell E. coli, the mini-MMO is now 100 fold better than the native methanotroph.

Reviewer #3 (Remarks to the Author):

The authors present an interesting study that uses computational approaches to design and express soluble and active MMO enzyme. Overall the approach appears to work and the authors present compelling data on the activity of their designed MMO. This work is highly impactful and establishes a new biocatalyst for the production of methanol from methane.

I would recommend publication, with some clarification and rewrites of certain sections, with a few technical questions for clarification.

The introduction is rather dense and would benefit from significant rewriting to make the manuscript more compelling. Specific comments about the introduction:

The MMOH enzyme is not in a dimer organisation, which it has dimers of all the subunits, the overall arrangement of A₂B₂G₂ is a heterohexamer. Please clarify here.

The roles of the MMOH in gating and electron transfer should be explained more clearly.

High-throughput biosynthesis... using E. coli - reword for clarity. Is it really high throughput just growing E. coli?

Summarise key results at the end of the methods section please.

Results -

Table 1 isn't really a table, it is more of a figure and the layout is confusing with all of the domain boundaries shown above the cartoon of the constructs. Can this be clarified and inserted as a figure rather than a table. This is a key figure and needs to be clear to communicate your designs.

P5 of PDF - 'we first slimed down...' do you mean slimmed down?

Human H-ferritin - can you be sure that the ferritin in the growth conditions you use, especially those with iron, is actually free of iron? Ferritin is a great scaffold, but it is not an inert scaffold by any means. Particularly in the case here where you have an iron-dependent redox enzyme. You present results in Figure 3b to show that the HuHF does not produce methanol in your assay, however you should introduce a control experiment in which the FOC of the HuHF is mutated to inactivate it to control for the potential interaction/influence of the HuHF with the attached enzymes. Would also be instructive to show a HuHF control micrograph in figure 3A.

Construction of the B-subunit of SM1 - how did you facilitate a C-term-to-C-term fusion, this needs clarification.

Reviewer #4 (Remarks to the Author):

Yu et al report on the development of a miniature soluble methane monooxygenase (sMMO) assembled on an apoferritin scaffold. The native soluble MMO (as opposed to the particulate MMO, see later) is a three-component enzyme comprised of hydroxylase, reductase and regulatory units. The authors undertook the ambitious task to trim down the native enzyme to its most essential core parts that are still active enzymatically. The design strategy involves fusion of the short-forms of hydroxylase and reductase (FAD domain only, ferredoxin domain deleted). The fusion construct is linked to one subunit (alpha) of the apoferritin scaffold. The regulatory unit is linked the opposite subunit (beta) of the scaffold. It is presented in two orientations, one being inverted to the native enzyme (retroB). On expression in *E. coli*, the scaffold protein subunits self-assemble into protein complexes that expose the MMO components in a way to reconstitute activity. The authors use molecular dynamics simulations to guide and interpret their strategies of protein design. The scaffold-linked protein complexes are expressed well in soluble form whereas the non-scaffolded hydroxylase-reductase fusion is insoluble.

The enzyme constructs are active MMOs and their catalytic components (in particular the metal center of the hydroxylase unit) are characterized with biophysical methods. E. coli strain expressing the best-active construct referred to as sm1 (hydroxylase-reductase fusion with retroB) produced about 3 g/L methanol. The authors claim this to have been an important step towards bioproduction using MMO.

The study has been performed well and the results are interesting. However, there are concerns.

1: Earlier work by some of the authors (Nat Catal 2019, ref 17) has used effectively the same strategy of scaffolding on the particulate MMO (pMMO). The structure of the pMMO differs from the sMMO in respect of the electron supply chain but the hydroxylase unit is really very similar. Ref 17 is cited only once in the Introduction amongst other papers, apparently as a note in passing. The whole manuscript is then written to convey the idea that everything done in the current study is new. This reviewer noted only on third reading of the manuscript how important as precedent reference 17 is. The whole experimental approach is extremely similar in the two studies. It can be useful to compare the figures for similarity in appearance. This is not to say that the current study does not have important elements of novelty. But reference 17 certainly compromises novelty/originality of the approach.

2: The idea of shortening the electron transfer from NADH in the miniature MMO is interesting. However, it seems to be not fully worked out. The constructs should include a complete reductase unit (FAD domain and ferredoxin domain) and show how it compares to the redesigned unit containing only the FAD domain. My concern is this: a suitably exposed FAD domain will easily become reduced by NADH but the efficiency of electron transfer to the metal center is not clear. The authors should compare the ratio between NADH oxidized and methanol formed (a sort of coupling efficiency of the catalytic steps). It could easily be that the miniature MMOs show strongly altered coupling. An enzyme that requires many molecules of NADH to get a single molecule of methane hydroxylated is not an efficient catalyst. Coupling efficiency might represent a critical limitation of artificial MMOs. The FAD reduced from NADH may get reoxidized by O₂, resulting in peroxide and other protein-damaging reactive oxygen species. The structural control (full reductase unit) should be performed and the coupling efficiency of the construct is a critical to-do.

3: The evidence on E. coli strain for production of methanol is extremely preliminary and does not support the far-reaching claims (e.g., biomanufacturing etc.) made. Point of the reviewer is not that there is room for optimization, it is conceptual. The E. coli strain is first grown to a relatively high cell density (OD of almost 80!) and then methanol production is initiated by induction of sMMO synthesis. The system will burn glucose mostly for aerobic metabolism and just a very tiny part of it will go into methanol production. It seems relatively easy to find conditions in which such a process shows a higher productivity than a process based on methanogens that use completely different substrate conditions and lack the growth to high cell density. But the level of comparison is irrelevant and in no way does the higher productivity imply that the E. coli process would be more promising for a practical application.

4: Determination of the intracellular NADH requires rapid quenching of the cellular metabolism. In the way analyzed, the results in Figure 6c are not reliable.

5: The mutagenesis results identify residues somewhat important for activity but it is not clear how they would reveal the electron transfer path.

Other points

The structural description on pages 5 – 8 was very difficult to follow. It was not clear how unique/clear are the results of the molecular dynamics simulations.

The discussion of where the FAD molecule would have to be positioned with respect to the metal center was not clear in all parts. What is the optimal region (page 5)? Which constraints were used and why? (page 5)

The size of the assembled particles varies from 18 – 19 nm in this study and the earlier Nat Catal paper shows even larger particles. It is not clear how these sizes inform about protein exposed on the surface.

The characterization of the active sites (metal center, FAD) appear to have been carefully done and is interesting. But it wasn't always clear why these characterizations are needed. For example, the detailed assessment of the iron center with EPR and x-ray absorption.

Revision Details

Reviewer #1:

Major comments:

[Comment 1]

My only substantial concern with the work is for the Authors to be clearer that all relevant controls have been done to establish methane oxidation activity by the protein complex they have made. With the pure protein work, it needs to be made clear whether the buffer controls contained all non-protein components of the reaction (including NADH, where commercial preparations sometimes contain volatiles that confound monooxygenase assays) and that the reaction was shown to be dependent on added NADH via no-NADH controls. In the whole-cell biotransformation work some more information would be helpful to be certain that methane is the source of methanol produced and that the carbon-13 methanol observed could not be due to natural abundance of carbon-13 in methanol produced from some source other than the added carbon-13 labelled methane.

Revision:

We performed a couple of control experiments of *in vitro* methane oxidation - control 1 (enzyme-free buffer with 0.3 mM NADH) and control 2 (NADH-free buffer with sm1), and the results demonstrate that no methanol was produced in both of the control experiments, as shown in the figure below.

The figure above was added to Supplementary Information (Supplementary Fig. 9) and cited in the text (p. 10) of revised manuscript.

Further, we performed the additional methane oxidation experiments using sm1-expressing *E. coli* after each of natural- and C^{13} methane was dissolved in the reaction buffer and analyzed the oxidation products through C^{13} -NMR spectroscopy. As presented in the figure below, when the reaction buffer only (prepared without adding any methane) and the buffer containing natural C^{12} -methane were analyzed, no C^{13} -methanol peak was detected, while only in case of the methane oxidation using C^{13} -methane, C^{13} -methanol peak was evidently detected. This indicates that the C^{13} -methanol produced by sm1-expressing *E. coli* is the oxidation product of externally added C^{13} -methane, NOT the oxidation product of naturally existing C^{13} -methane. (Reportedly, natural C^{13} is only $\sim 1.1\%$ of total carbon.) The C^{13} -NMR spectroscopy of the original manuscript was replaced by the figure below in the revised manuscript (Supplementary Fig. 18).

[Comment 2]

In describing and discussing the new biocatalysts constructed, the Authors need to be careful to distinguish the properties that are experimentally verified in their work from those (such as many aspects of the three dimensional structure of the protein and the dependence of the enzyme on NADH in whole *E. coli* cells) which are assumed from in silico modelling without direct experimental verification.

Revision:

In the main figures, Fig. 1e, Fig. 2a-g, and Fig. 5a-c are the results of *in silico* modeling, which was more clearly described in the relevant text (p. 5-7, p 11-12) and figure captions to clarify the achievement by *in silico* modeling.

[Comment 3]

Much of the discussion section is a recapitulation of the results and could be shortened. It is useful to evaluate the productivity of the new recombinant system for methane to methanol conversions in comparison with methanotroph-based systems, as the Authors do. It would also be relevant to discuss the extent to which the increased yield of methanol in the recombinant *E. coli* may be due to the absence of the methanol oxidation pathway that is present in methanotrophs and under most conditions consumes much of the methanol produced by methane monooxygenase.

Revision:

According to the reviewer's comment, we shortened the Discussion with reducing the recapitulation part of the experimental results. We also emphasized that the methanol yield and productivity were significantly enhanced by the new recombinant *E. coli* system through comparing it with the methane to methanol conversion by methanotroph systems and explained about the reason. That is, with citing an appropriate reference (*Nat. Commun.* **13**, 5243 (2022), ref. 60), we added the following sentence in Discussion (p. 15): "...the methanol yield and productivity were remarkably improved up to 2.9 g/L and 0.11 g/L/h (22 to 55-fold higher than methanotrophic production), respectively, through the *in-situ* methane oxidation in the high-cell-density culture of mini-sMMO-expressing recombinant *E. coli* - which seems partly due to the fact that unlike methanotrophs that utilize the methanol produced by MMO as a carbon and energy source, *E. coli* does not have such a methanol oxidation pathway⁶⁰ - indicating that this novel approach is the best-ever strategy in the biomanufacturing of methanol."

Minor comments:

[Comment 1]

Page 1 (Abstract) "with demonstrating" would be better as "as well as demonstrating".

Revision:

We changed the phrase, "with demonstrating" to "as well as demonstrating".

[Comment 2]

Page 1 (Abstract). The usefulness of the abstract would be increased by including quantitative information about the turnover numbers and methanol yields achieved in the work.

Revision:

We added quantitative information about the turnover number and methanol yield to the Abstract of revised manuscript.

[Comment 3]

Page 2. In describing the distribution of the particulate and soluble methane monooxygenases, the Authors should mention that some methanotrophs, such as the facultative methanotrophs of the genus *Methylocella*, possess the soluble enzyme but not the particulate.

Revision:

Reportedly, some facultative methanotrophs such as the genus *Methylocella* and *Methyloferula* possess only sMMO, not pMMO (*J. Bacteriol.* **187**, 4303-4305 (2005); *Int. J. Syst. Evol. Microbiol.* **54**, 151-156 (2004); *Int. J. Syst. Evol. Microbiol.* **61**, 2456-2463 (2011)). Accordingly, we added the following phrase in the text (p. 2) of revised manuscript: "...found in some methanotrophs (including facultative *Methylocella* and *Methyloferula* species⁹⁻¹¹ that possess only sMMO)..."

[Comment 4]

Page 2 (end of first paragraph). "timely" would be better as "coupled". "the best known being NADH" would be better as "NADH or NADPH".

Revision:

We changed the word, "timely" to "coupled" and changed the phrase, "the best known being NADH" to "NADH or NADPH".

[Comment 5]

Page 2 (second paragraph). "seems still incomplete" would be better as "is still incomplete". Also, consider replacing "many plausible" with "a number of possible".

Revision:

We changed the phrase, “seems still incomplete” to “is still incomplete” and also replaced “many plausible” with “a number of possible”.

[Comment 6]

Later in the same paragraph. The lack of activity of recombinant pMMO cannot be solely attributed to its inability to receive electrons from NADH.

Revision:

pMMO activity depends on the efficiency of proton transfer and electron transport chain in membrane and also requires additional protein components necessary for stabilization and copper loading (*Chem. Soc. Rev.* **50**, 3424-3436 (2021); *Biochemistry*, **54**, 2283–2294, (2015); *J. Biol. Chem.* **293**, 10457–10465, (2018)). Accordingly, we added the following sentence, “Moreover, pMMO activity depends on electron/proton transfer efficiency in cytoplasmic membrane, requires additional protein components necessary for stabilization and copper loading¹⁹⁻²¹, and...” to the text (p. 2) of revised manuscript.

[Comment 7]

Next to last line of page 2. Fast reaction kinetics and other enzymological methods have been more important in establishing the mechanism of sMMO than crystallography and NMR.

Revision:

The enzymological nature of sMMO such as fast reaction kinetics was elucidated through some important studies (*Biochem. J.* **236**, 155-162, (1986), *Biochemistry*, **38**, 12768-12785, (1999), *Biochemistry*, **48**, 12145-12158, (2009), *Angew. Chem. Int. Ed.* **40**, 2782-2807, (2001)), which significantly contributed to establishing the catalytic mechanism of sMMO as well as the structural nature that was revealed through crystallography and NMR investigations. Accordingly, we modified the relevant sentence in the text (p. 3) of revised manuscript as follows: “...owing to some seminal studies on enzymological nature including reaction kinetics²⁵⁻²⁸ as well as crystallographic and NMR investigations²⁹⁻³², ...”.

[Comment 8]

First line of page 3. “enzymes” would be better as “components”.

Revision:

We changed the word, “enzymes” to “components”.

[Comment 9]

Page 3, last sentence of first paragraph. Recombinant expression of sMMO and mutants of it has been achieved by homologous expression in methanotroph cells. It is the problem of heterologous expression in a genetically easily tractable host that had not previously been achieved.

Revision:

We modified the relevant sentence in the text (p. 3) of revised manuscript as follows: “Due to this structural and functional complexity, the heterologous expression of catalytically active recombinant sMMO in a genetically easily tractable host like *E. coli* remains still unsettled;...”.

[Comment 10]

Page 4. First sentence of last paragraph. It needs to be made clear what reaction of MMOH alone is being referred to here, presumably its activation by hydrogen peroxide in the peroxide shunt reaction.

Revision:

The oxidation reaction of MMOH alone that we referred to here is not peroxide shunt reaction (i.e., the reaction by MMOH in the presence of excessive H₂O₂) but is the slow hydroxylation of alkanes and alkenes by chemically reduced MMOH alone (*J. Biol. Chem.* **264**, 10023-10033 (1989), ref. 38). To avoid unnecessary confusion, we cited this paper in the relevant sentence (p. 5) as follows: “Reportedly, MMOH alone can oxidize methane but very slowly³⁸...”

[Comment 11]

Page 5. “slimed” should be “slimmed”.

Revision:

We corrected the typo-error.

[Comment 12]

Page 6. Second sentence of second paragraph. “inverted” here suggests that the sequence

was reversed from the natural protein B, which clearly is not what was done. I presume some kind of domain inversion was done. A clearer explanation is needed and sufficient details for the reader to understand exactly what the construct was.

Revision:

Because the sequence of retroB is the reversed sequence of native MMOB (that is, the N and C-terminus of retroB corresponds to the C- and N-terminus of native MMOB, respectively, as shown in the table below), we changed the word, “inverted” to “reversed” in the text (p. 6) of revised manuscript. To clarify the genetic linkage between huHF and native/reversed sequence of MMOB, done in the construction of miniatures of sMMO (sm1 and sm2), we added the following sentence in the caption of Table 1 (p. 34): “That is, in sm1, the C-terminus of native MMOB was linked to the C-terminus of huHF₁₃, while in sm2, the N-terminus of native MMOB was linked to the C-terminus of huHF₁₃.”

Protein component	Primary structure
MMOB (B)	SVNSNAYDAGIMGLKGFADQFFADENQVVHESDTVVLVLLKKSDEI NTFIEEILLTDYKKNVNPTVNVEDRAGYWWIKANGKIEVDCDEISELL GRQFNVDYDFLVDVSSTIGRAYTLGNKFTITSELMGLDR KLEDYHA
retroB	AHYDELKRDLGMLESTITFKNGLTYARGITSSVDVLFDYVNFQRGLLE SIEDCDVEIKGNAKIWWYGARDEVNVTNPNVNKKYDTLLIEEIFTNIEDS KKLVLVVTDSEHVQVEDAFFQDAFDKGGKLGMIADYANSNVS

[Comment 13]

Page 6-7. The sentence that crosses between the pages here is one of the instances where it needs to be made clear that these are results from *in silico* modelling rather than experiment.

Revision:

To discriminate the results of *in silico* modeling from experimental work, we re-described the relevant text (p. 7) as follows: “The interaction of Y8(B)-R307(□H□) in the region 1 of MD-simulated contact map - which was also experimentally investigated through a previous study³⁹ - causes...”

[Comment 14]

Page 8. Second sentence of second paragraph. There is some overinterpretation of the

data in Fig. S3a-c here, which shows growth curves of recombinant *E. coli* and does not give information about the iron occupancy of the enzyme active sites.

Revision:

Through ICP-MS analysis, we estimated the variation in iron quantity (moles) loaded on a mole of sm1 synthesized in *E. coli* as the concentration of iron sulfate in the culture medium increased up to 0.5 mM. As presented in the figure below, sm1 is fully saturated with iron at the iron sulfate concentration of 0.4 mM. This figure was added to Supplementary Information (Supplementary Figure 3a) and was cited in the text (p. 9).

[Comment 15]

Page 8. Next to last sentence. It is not clear how the SDS-PAGE shown in Fig. S4a shows the stoichiometry of the mini-sMMO complexes. An explanation is needed.

Revision:

The stoichiometry was estimated through quantifying the α - and β -subunits of mini-sMMOs (sm1 to sm3, comprised of 24 subunits ($\alpha+\beta$)) expressed in *E. coli*. The quantification was done through measuring the Coomassie-stained band intensity (optical density/mm²) of α - and β -subunit of each mini-sMMO loaded on SDS-PAGE gel. That is, the Coomassie-stained protein bands were scanned and analyzed using a GS-800 calibrated densitometer (Bio-Rad) and Quantity One software (Bio-Rad). The relative % of the measured intensity of α - and β -subunit band was used to calculate the stoichiometry, i.e., the fraction of each subunit out of total 24 subunits comprising a mini-sMMO. Consequently, the calculated stoichiometry shows that the number of α - and β -subunits comprising each mini-sMMO (24mer) is about 7 and 17, respectively (p. 9 of the text of revised manuscript). This protocol was added to the Methods (p. 19) of revised manuscript.

[Comment 16]

Page 8. Last line. “consistent with” would be more accurate than “indicating that”.

Revision:

We added the following modified sentence in the revised text (p. 9): “The enlarged size of mini-sMMOs is consistent with the fact that the R_{FAD}-ΔH α and B/retroB linked to the C-terminus of huHF subunit are flipped outward and thus localized on the exterior surface of huHF scaffold,...”.

[Comment 17]

Bottom of page 8/top of page 9. Some more explanation is needed of how the absence of cross-reactivity with the antibodies shows the stated “flipped” conformation (maybe because the epitopes that the antibodies recognise are moved to the inside of the complex?).

Revision:

As described in Methods (p. 18) of revised manuscript, we investigated whether the R_{FAD}-ΔH α and retroB in sm1 are localized on the huHF surface through Western blot analysis of purified sm1 and wild-type huHF using mouse **polyclonal anti-huHF IgG** (ab77127, Abcam, Cambridge, UK), **NOT anti-sMMO IgG** as primary antibody. Therefore, the non-reactivity for huHF in Western blot analysis of sm1 indicates that the huHF scaffold of sm1 is fully masked by R_{FAD}-ΔH α and retroB and thus that R_{FAD}-ΔH α and retroB were localized on the huHF scaffold surface.

[Comment 18]

Page 10. More explanation is needed for the “bfr” and “afr” samples. Exact conditions for how they were prepared is needed (ideally as a reference to the appropriate part of the Methods) and it needs to be clearly indicated why it is expected that the two forms should have iron in the different oxidations states (maybe because “bfr” has had the enzyme incubated with NADH in the absence of methane?).

Revision:

“That is, sm1-bfr is the sample prepared before it is used for *in vitro* methane oxidation, while sm1-afr was sampled at 16 h after methane oxidation by sm1 began in the presence of NADH.” This sentence was added to the text (p. 10-11) of revised manuscript to clearly

describe the difference between the two sm1 samples.

It seems reasonable to presume that a larger portion of sm1-bfr is at reduced state because it was synthesized in *E. coli* cytoplasm containing a substantial amount of NADH, and thus the ratio of Fe⁺² to Fe⁺³ is far higher than 1.0 (Figs. 4a,c). As for the sm1-afR sampled during actively producing methanol, it seems evident based on the catalytic cycle of sMMO (shown in the attached schematic below) that the portion of the enzyme at oxidized state increases, which is believed to be the reason why the ratio of Fe⁺² to Fe⁺³ notably decreased below 1.0 (Figs. 4b,c). We modified the relevant sentence in the text (p. 11) as follows: “Based on the catalytic cycle of sMMO, it seems reasonable to assume that a larger portion of sm1-bfr is at reduced state, while sm1-afR are largely at oxidized state.”

[Comment 19]

Later on page 10, it is an overinterpretation of the results to say that the catalytic cycle is

Curr Opin Chem Biol 2002, 6(5), 568–576.

shown to be “exactly the same” as native MMOH. It would be reasonable to say that the results are consistent with a similar catalytic cycle.

Revision:

We modified the sentence in the text (p. 11) of revised manuscript as follows: “The significantly higher portion of Fe(III) in sm1-afR demonstrates that the diiron center of sm1 follows a similar catalytic cycle as that of native MMOH α ⁵³.”

[Comment 20]

Page 11. Last three lines of the paragraph that starts “Identification of electron shuttling path”. “elucidate” would be better as “suggest” or “indicate”. Consider replacing “looking more important” with “being more important for methanol production”.

Revision:

We changed the word, “elucidate” to “indicate” and replaced the phrase, “looking more important” with “being more important for methanol production” in the text (p. 12).

[Comment 21]

Page 12. “we confirmed that methanol production was due to the catalytic activity of sm1” – the data in the figure cited (Fig. S11) do not show this, but (subject to the major comment above) that methane is the source of the methanol produced.

Revision:

We modified the sentence in the revised manuscript (p. 13) as follows: “...we confirmed that the methanol production was due to the supplied methane oxidation by sm1 expressed in the recombinant *E. coli*...”

[Comment 22]

Later in the same paragraph. “the recombinant sm1 expression continued” – this suggests continued production of the protein, whereas the data indicate the amount remained roughly constant. Hence, “the recombinant sm1 protein remained” may be better here.

Revision:

We changed the relevant sentence as follows: “...the expression level of recombinant sm1 remained unchanged...” (p. 13).

[Comment 23]

It may be worthy of mention in the Results or discussion that the main peak of intracellular NADH seen in Figure 6 is substantially before methanol production occurs, though there is a single elevated point that suggests a second smaller peak in NADH around the time of maximum methanol production (Fig. 6 b and c).

Revision:

We added the following sentence in the text (p. 13) of revised manuscript: “Although the

maximum peaks of intracellular NADH were detected before methanol production began (at 4 and 18 h), the NADH concentration remained nearly constant during the period of methanol production (Figs. 6b,c).”

[Comment 24]

Throughout the methods, it would be better to use g (acceleration due to gravity) than rpm when specifying centrifugation conditions.

Revision:

Using the equation of ‘g (acceleration due to gravity) = rpm² x 1.118 x 10⁻⁵ x r’, we converted the unit of all centrifugal speed values from rpm to g value in the revised manuscript.

[Comment 25]

Page 13. Please provide full details of the cell breakage conditions (including duration, power and temperature of sonication), or a reference to where these can be found.

Revision:

We provided full details of the cell breakage conditions in Methods of the revised manuscript (p. 17) as follows: “The cell pellets were suspended in cell lysis buffer (50 mM NaH₂PO₄, 300 mM NaCl, 10 mM imidazole, 0.2 mM FeSO₄, pH 8.0) and disrupted using Branson Sonifier (Branson Ultrasonics Corp. Danbury, U.S.A.) and Sonopuls ultrasonic homogenizer (Bandelin Sonopuls HD 4000, Bandelin Ele., Germany) with 50 % output energy (2.0 s pulses/ 5s pause) in an ice bath for 2 h.”

[Comment 26]

Fig. 3a and Fig. S9. TEM images and associated size distribution analysis of the scaffold protein without the sMMO components are needed.

Revision:

We added the TEM images and associated size distribution data (the attached figure below) of the scaffold protein (huHF) to Fig. 3a and Supplementary Fig.14b in the revised manuscript.

[Comment 27]

Fig. 6b and c. It is not clear why error bars are shown for only some of the data. Numbers of repeats, and whether the multiple measurements are technical or biological replicates, needs to be stated in the legend.

Revision:

The culture absorbance (OD₆₀₀) of Fig. 6b was measured quickly once to minimize the unnecessary cell growth during the measurement, which was done immediately after the culture samples were taken from the bioreactor and properly diluted with saline solution. The values of other parameters (methanol, NADH, glucose, and acetate concentrations (Figs. 6b,c)) were measured three times with the cell-free samples prepared through centrifuging the culture samples taken from the bioreactor, followed by the calculation of average and standard deviation for each measurement. Some data points do not show error bars, which is because the width of error bar is smaller than the symbol size. The bioreactor experiment was repeated twice. In one experiment, NADH concentration was not measured, while the above measurement and analysis protocols were equally applied to the other parameters. We included the results of one bioreactor experiment in Figs. 6b,c, because the variation in the measured values between the two bioreactor experiments was negligible.

We added the following sentence in the Fig. 6 caption: “(The culture absorbance was measured once, while all other measurements were done three times at a time.)”

Reviewer #2:

Major comments:

[Comment 1]

The authors present no actual structural characterization of the mini-MMO construct or

evidence that a diiron cluster is actually formed. Everything is based on a computation of a massively modified and reconstructed (and even inverted in the case of MMOB) over-expressed protein complex. Some spectroscopic techniques are used to identify oxidation states of the presumed diiron cluster, but the way they are analyzed is either incorrect (see EPR comments below) or incomplete (no fits or quantification for the EXAFS, for example). The preparation of the samples for XANES and EXAFS by freeze drying the enzyme after 20 hours of reaction and expecting to get an accurate oxidized to reduced ratio is overly optimistic. It is not clear in any case how partial oxidation of the presumed diiron cluster in the construct is indicative of either MMO-like catalysis or a diiron cluster. Simple extensions like attempting to make a mixed valent cluster, which does have a characteristic EPR spectrum, or adding azide to the construct to attempt to detect the characteristic chromophore were not attempted. If enough enzyme for XANES can be obtained, then it should be possible to cheaply reconstitute with ^{57}Fe for definitive identification of the diiron cluster and its oxidation state by Mössbauer. Of course, this would also reveal whether only one type of iron center is present.

Revision:

The mini-MMO construct contains $\Delta\text{H}\alpha$ (i.e. a partial sequence of native MMOH α , where diiron active center is located, Table 1), and through the ICP-MS analysis, we detected about 2 moles of iron per mole of $\Delta\text{H}\alpha$. The EXAFS analysis of mini-sMMO (sm1) shows that the local arrangement of atoms around an iron atom of the diiron center is in good agreement with the previous results for native sMMO (Figs. 4d,e). (The results of X-ray spectroscopy of sm1-bfr and sm1-aftr are plotted using the raw data provided by the Aichi Synchrotron Radiation Center (Aichi, Japan) without any fitting process. The meaning of “no fits” in the comment above is not clear to us. Please refer to our response to the minor comment 13 below.) Thus, it seems reasonable to presume that like native MMOH, the mini-sMMO contains diiron center. The remaining issue to be clarified is that the diiron of mini-sMMO is involved in the catalytic methane oxidation of native sMMO. If it is so, the oxidation/reduction state of diiron must change in a logical, scientific manner before and after methane oxidation begins. We prepared “sm1-bfr” and “sm1-aftr” samples as follows: the sm1 that was synthesized in *E. coli* and purified was mixed with a buffer containing neither NADH nor dissolved methane, followed by freeze drying, resulting in the preparation of “sm1-bfr” sample, while the sm1 taken from the reaction buffer at 16 h after methane oxidation began in the presence of NADH was freeze-dried, resulting in the preparation of “sm1-aftr” sample. Then, we performed XANES analysis with the sm1-bfr and sm1-aftr samples to analyze the oxidation/reduction state of diiron (Fig. 4c),

demonstrating that both ferrous and ferric ions were detected in the sm1-bfr and sm1-afr and that the portion of ferric ions is notably higher in the sm1-afr than that in the sm1-bfr, which corresponds to the previous findings for the catalytic cycle of native sMMO (*Coordination Chemistry Reviews* **322**, 142-158 (2016)). In summary, the mini-sMMO (sm1) contains diiron per mole of $\Delta H\alpha$; the local arrangement of atoms around the diiron of sm1 is in good agreement with native MMOH α ; the diiron is comprised of both ferrous and ferric ions; and finally, the portion of ferric ions significantly increases after methane oxidation begins, corresponding to the native catalytic cycle of sMMO. Accordingly, it seems reasonable to conclude that the mini-sMMO (sm1) contains native sMMO-like diiron active center.

According to the reviewer's comment, we performed a new azide experiment to confirm the presence of ferric ions in sm1. In the presence of ferric ions, sodium azide combines with ferric ions, followed by the formation of chromophoric azide-ferric complex, which generates specific absorbance peaks at 345 and 450 nm (*PNAS*, **90**, 2486-2490, (1993), *PNAS*, **107**, 15391-15396, (2010)). First, we examined the light absorption profiles of sm1-bfr and sm1-afr in the range of UV-visible light and found a main absorbance peak at 281 nm and another weak peak at 395-420 nm for the both samples as shown in the inset plots (red arrows) of the attached figures below, which exactly corresponds to the previous reports for MMOH (*Biosci. Biotechnol. Biochem.*, **71**, 122-129, (2007), *Catalysts*, **8**, 582 (2018)). Then, we performed the azide experiment by adding sodium azide (2 M) to the sm1-bfr and sm1-afr solution (1 μ M) and observed two distinct absorbance peaks at 345 and 450 nm as shown in the figures below (blue arrows), indicating that both of sm1-bfr and sm1-afr have ferric ions, which corresponds to the earlier results of EPR and XANES analysis (Figs. 4a-c). We added the figure below to Supplementary Information (Supplementary Fig. 12), which was cited in the text (p. 11) of revised manuscript as follows: "(The presence of Fe(III) in sm1 was also confirmed through an azide experiment (Supplementary Fig. 12))."

Reportedly, the diiron center of other bacterial enzymes (small subunit (R2F) of native ribonucleotide reductase from *Corynebacterium ammoniagenes* and iron-sulfur cluster repair protein (YtfE) from *E. coli*) was analyzed through EPR at 77K (*J. Biol. Chem.*, **275**, 25365-25371 (2000); *Chem. Eur. J.*, **22**, 1-10 (2016)); however, we performed a new EPR analysis at 5 K with the same samples of sm1-bfr and sm1-af_r, according to the reviewer's comment that we need to characterize more complete structural features of the mini-*s*MMO construct. As presented in the attached figure below, both ferrous and ferric ions were detected in the sm1-bfr and sm1-af_r and that the portion of ferric ions is notably higher in the sm1-af_r than in the sm1-bfr. We replaced Figs. 4a,b of original manuscript with this figure below in the revised manuscript.

[Comment 2]

Significant effort was devoted to making mutants of residues predicted to be part of the diiron cluster ligation, FAD binding site, and electron transfer pathway. Changes were observed in iron content of the sample and/or methanol production, but all of the changes could be attributed to changes in overall structure caused by the mutations. Without a structure, or at least an attempt to show that the mutations don't change the structure, it is all guesswork.

Revision:

We performed additional analyses (ICP-MS and CD (circular dichroism) spectroscopy) of sm1 and sm1 mutants (sm1-cat_{mut}, sm1-FAD_{mut}, and sm1-e_{mut1}) to investigate the iron content and secondary structure of the mutants. First, ICP-MS analysis (figure attached below, Supplementary Fig. 16 in the revised manuscript) show that the iron content of sm1-cat_{mut} is significantly lower than that of sm1-FAD_{mut} and sm1-e_{mut1}, which is because

sm1-cat_{mut} was prepared through the mutation of key ligands (E114A, E144A, H147A, E209A, E243A, and H246A) around the diiron center. On the other hand, sm1-FAD_{mut} and sm1-e_{mut1} are shown to keep a significantly higher quantity of iron, because the mutated residues in sm1-FAD_{mut} and sm1-e_{mut1} are irrelevant to the diiron center.

Second, we performed CD analysis of sm1, sm1-cat_{mut}, sm1-FAD_{mut}, and sm1-e_{mut1} and compared their CD spectra as shown in the figure below (Supplementary Fig. 15 in the revised manuscript), demonstrating no significant difference in the secondary structures of sm1 and the three mutants. In particular, % of secondary structural elements in sm1 and sm1-e_{mut1} is nearly identical.

	% of each secondary structural element in sm1-mutants			
	sm1	sm1-cat _{mut}	sm1-FAD _{mut}	sm1-e _{mut1}
α -helix	43.9 (\pm 0.8) %	40.2 (\pm 6.9) %	38.9 (\pm 2.2) %	45.7 (\pm 1.5) %
β -sheet	21.1 (\pm 3.7) %	25.3 (\pm 8.9) %	28.1 (\pm 4.9) %	21.0 (\pm 2.8) %
Turn & Random coil	35.0 (\pm 4.4) %	34.5 (\pm 15.8) %	33.0 (\pm 7.1) %	33.3 (\pm 2.9) %

The above results indicate that the changes in iron and FAD content and methanol production (Fig. 3e, Figs. 5f,g) are not attributed to the changes in overall structure caused by the mutations. With citing the two figures above, the following sentence was added in the text (p. 12) of revised manuscript: “Considering the non-significant change in secondary structures (Supplementary Figs. 15) of sm1 and its mutants (sm1-cat_{mut}, sm1-FAD_{mut}, and sm1-e_{mut1}), the changes in iron content (Fig. 3e, Supplementary Fig. 16) and methanol production (Fig. 5g) are not attributed to the changes in overall structure caused by the mutations.”.

Minor comments:

[Comment 1]

Abstract: The first sentence of the abstract is 6 lines long, while the second sentence is 5 lines long. These are too complex for easy comprehension.

Revision:

The first long sentence of Abstract was divided into 2 sentences for easier comprehension.

[Comment 2]

Line 30 (first line) I think it is misleading to start out by saying that methanotrophs consume atmospheric methane. As shown in the Curry reference, they are a very small contributor to methane consumption once it enters the atmosphere. They are a large contributor to consuming dissolved methane produced by biological and geological processes.

Revision:

The first sentence of Introduction (p. 2) was modified to “Methanotrophs consume methane as much as approximately 30 million metric tons every year^{1,2} through the oxidation of aqueous methane using methane monooxygenase (MMO)...”.

[Comment 3]

Line 34 “convert a wide range of carbon feedstocks to methanol” this needs to be rewritten for clarity. Only methane is converted to methanol.

Revision:

The sentence was changed to “they can convert methane and a wide range of other carbon feedstocks to methanol and various value-added chemicals under mild conditions³⁻⁵.” (p. 2).

[Comment 4]

Lines 37-42. This sentence is too long and complex

Revision:

According to the comment, the sentence was rewritten as follows (p. 2): “Two different,

complex forms of MMO have been identified in methanotrophs: particulate MMO (pMMO) found in nearly all methanotrophs under copper-rich conditions and soluble MMO (sMMO) found in some methanotrophs (including facultative *Methylocella* and *Methyloferula* species⁹⁻¹¹ that possess only sMMO) under copper-limited conditions^{12,13}. pMMO is comprised of multiple transmembrane and periplasmic domains, while sMMO is comprised of functionally differentiated three components (hydroxylase, reductase, and regulatory enzyme).”.

[Comment 5]

Line 42 “intrude an oxygen radical” Intrude is not used correctly and the reactive species is probably not an oxygen radical. It might be a hydroxyl radical if the proposed rebound mechanism is correct (ie. Formation of a carbon radical by hydrogen atom abstraction. followed by rebound of the resulting hydroxyl radical.)

Revision:

Annu. Rev. Biochem. **88**, 409–431, (2019)

According to the recent report on the mechanism of sMMO (*Annu. Rev. Biochem.* **88**, 409-431, (2019)), reduced MMOH reacts with O₂ and is rapidly converted to Q state, and subsequently, methyl radical species and diiron cluster-bound hydroxyl radical are formed by the reaction of hydrogen atom abstraction at R state, followed by methanol production through the recombination of above radicals. The relevant sentence was rewritten as follows (p. 2): “In particular, sMMO produces methanol through the recombination of methane-derived methyl radical and diiron-bound hydroxyl radical¹⁴⁻¹⁶, which...”.

[Comment 6]

Line 53 “owing to some seminal studies based on crystallographic and NMR investigations” While these studies contributed to understanding the mechanism they are far from the complete story. There are much more significant studies that actually address mechanism rather than simply structure. The structure references given are also very incomplete.

Revision:

This comment is the same as the minor comment 7 of Reviewer #1. Please see our response to the comment in page 6 above.

[Comment 7]

Line 54 and 55 and other places later in the manuscript “three different enzymes” only two of the components are enzymes.

Revision:

This comment is the same as the minor comment 8 of Reviewer #1. Please see our response to the comment 8 in page 7 above.

[Comment 8]

Lines 60-62 “(MMOB) that plays two simultaneous roles” The knowledge of the roles of MMOB began much earlier than the references provided and they have come much further since the referenced papers appeared. A much better knowledge and communication of the state of the knowledge in this field will be necessary for the reader. The manner in which MMOB and MMOH interact is extremely complex and involves binding and release of MMOB from MMOH in concert with binding and release of MMOR. This make it all the more remarkable that a fixed construct of MMOH-MMOR with an inverted MMOB could be reactive.

Revision:

We included more detailed knowledge about the roles of MMOB and rewrote the relevant sentence in the text (p. 3) of Introduction as follows: “...3) a regulatory enzyme (MMOB) alternates between binding to MMOH and MMOR, the binding pair depending on the phase of the catalytic cycle of sMMO and plays a critical role in regulating proton transfer and methane/O₂ gating³⁴⁻³⁶.”.

We put forth the hypothesis that reversing the MMOB sequence would reproduce the native binding characteristics between MMOB and MMOH. This hypothesis was corroborated by our MD investigation, which revealed that the binding characteristics between $\square H\square$ and *retroB* in the designed minimal model is nearly the same as in the binding between native MMOH and MMOB (Figs. 2e-g, Fig. 5a). Moreover, the binding-stability tests exhibited a significantly improved stability of FAD within the mini-sMMO (sm1) (Supplementary Movie 1), showing a preservation of native contacts.

[Comment 9]

Introduction: In general, the introduction omits a great deal of insight from the recent literature. I think it would benefit from extensive rewriting, both to improve English presentation and content. In contrast, the English in the rest of the manuscript is much better.

Revision:

We have rewritten the Introduction with improving English.

[Comment 10]

Line 214 and on “Notably, approximately 2,000 moles of methanol were produced per mole of sm1 within 20 h” and “The turnover frequency (TF) of sm1 was about 0.32 s^{-1} ” The computation here needs to be clarified for the reader. 20 h is 72000 s. 2000 turnovers in 72000 seconds is 0.03 s^{-1} or 10 fold lower than claimed in Figure S8.

Revision:

In general, turnover frequency (TF) of an enzyme is estimated *in vitro* during an initial, short period when the enzyme activity is maintained at maximum level. Here we estimated the turnover frequency for initial 15 min (900 sec) - when methanol production rapidly increases (Figs. 3b,d) - after methane oxidation began. The average amount of methanol produced for initial 15 min is 289.8 ± 65.0 moles, and therefore, TF was calculated to be about 0.32 s^{-1} . We added the following sentence in the Methods (p. 20) of revised manuscript: “TF was estimated for an initial 15 min after methane oxidation began.”

[Comment 11]

Line 235 “The EPR spectra (Figs. 4a,b) show that the ratio of Fe(II) to Fe(III) measured for sm1-bfr and sm1-afr is 1.91 and 0.81, respectively, which is very similar to that of

native sMMO” Neither oxidized nor reduced MMOH have EPR spectra in the $g = 4$ to 1.8 region shown in the oxidized or reduced forms. Both forms are spin coupled with total net spin of 0 in the diferric state and 4 for the diferrous state. The spectra shown are mostly of a copper contaminant. They have no relationship to native MMOH spectra. The reduced diiron cluster of MMOH does have a distinctive EPR resonance at $g = 16$ but it cannot be quantified by simple double integration because it derives from an integer spin system. This region is not shown in the figure. Also, 77K is too high a temperature for accurate investigation of the EPR spectra of a diiron cluster. As stated above, the significance of the oxidation state of the presumed diiron cluster after 20 h of reaction is not clear.

Revision:

As stated by the reviewer, the fully oxidized, diferric state (MMOHox) is anti-ferromagnetic at low temperature and does not give an EPR signal in the $g \sim 1.8$ and ~ 4 region (*J. Biol. Chem.* **267**, 261-269, (1992)), while the fully reduced form of the hydroxylase exhibits a signal at $g = 15 \sim 16$ (*J. Am. Chem. Soc.*, **112**, 15 (1990)). The XANES analysis (Fig. 4c) shows that the diiron of both sm1-bfr and sm1-afr is at mixed valence state. Because the fully reduced state can be achieved only by chemical reducing under anerobic condition, it is hardly possible that sm1-bfr prepared under aerobic condition exists at fully reduced state. (We prepared the sm1-bfr and sm1-afr samples as follows: the sm1 that was synthesized in aerobically cultured *E. coli* and purified was mixed with a buffer containing neither NADH nor dissolved methane, followed by freeze drying, resulting in the preparation of sm1-bfr, while the sm1 taken from the reaction buffer at 16 h after methane oxidation began in the presence of NADH was freeze-dried, resulting in the preparation of sm1-afr.)

According to the previous literatures, the diiron center of other bacterial enzymes (small subunit (R2F) of native ribonucleotide reductase from *Corynebacterium ammoniagenes* and iron-sulfur cluster repair protein (YtfE) from *E. coli*) was analyzed through EPR at 77K (*J. Biol. Chem.*, **275**, 25365-25371 (2000); *Chem. Eur. J.*, **22**, 1-10 (2016)); however, we performed a new EPR analysis at 5 K with the same sm1-bfr and sm1-afr samples, according to the reviewer’s comment. The EPR results at 5 K (figure attached below, Figs. 4a,b of revised manuscript) show the unique triplet signal of iron at the magnetic field of ~ 150 mT and thus are not the spectra of copper contaminant at all. Also, the EPR at 5 K does not show any spectra at $g = 15 \sim 16$, meaning that fully reduced state of iron does not exist. Reportedly, the EPR spectrum of MMOHmv (MMOH at mixed valence state) is

found in the $g \approx 2$ region, (*J. Am. Chem. Soc.*, **115**, 9 (1993); *J. Biol. Inorg. Chem.*, **1**, 297-304, (1996)), and the new EPR at 5 K also shows the spectra at the $g \approx 2$ region and thus indicate that the diiron of sm1-bfr and sm1-afr is at mixed valence state, corresponding to the earlier results of XANES analysis (Fig. 4c). From the EPR spectra at 5 K, the ratio of ferrous to ferric irons is far higher than 1.0 in sm1-bfr, while the ratio decreases below 1.0 in sm1-afr, demonstrating that the portion of ferric ions notably increases after methane oxidation begins, which corresponds to the catalytic cycle of native sMMO (*Annu. Rev. Biochem.* **88**, 409–431, (2019)).

[Comment 12]

Line 238 on. The XANES spectra are at best unconvincing that the observed iron is related to a diiron cluster in the enzyme. Also the statement on line 239-240: “The significantly higher portion of Fe(III) in sm1-afr demonstrates that the diiron center of sm1 follows exactly the same catalytic cycle as that of native MMOH α .” claims much more than the data supports.

Revision:

The mini-sMMO constructs (sm1 to sm3) contain the native MMOH α sequence with diiron active center (i.e., R_{FAD}- Δ H α , Table 1), and through the ICP-MS analysis, we detected about 2 moles of iron per mole of the MMOH α sequence of sm1. The EXAFS analysis of sm1 shows that the local arrangement of atoms around an iron atom of the

diiron is in good agreement with the previous results for native sMMO (Figs. 4d,e). Thus, it seems reasonable to presume that like native MMOH, the mini-sMMO (sm1) contains diiron center. The remaining issue to be clarified is that the diiron of sm1 is involved in the catalytic methane oxidation of native sMMO. If it is so, the oxidation/reduction state of diiron must change in a logical, scientific manner before and after methane oxidation begins. Then, we performed XANES analysis with the sm1-bfr and sm1-afr (freeze-dried samples) to analyze the oxidation/reduction state of diiron (Please refer to our response to the comment 11 about how we prepared the sm1-bfr and sm1-afr samples.), demonstrating that both ferrous and ferric ions were detected in the sm1-bfr and sm1-afr, and the portion of ferric ions is notably higher in the sm1-afr than in the sm1-bfr (Fig. 4c), which corresponds to the previous findings for the catalytic cycle of native sMMO (*Coordination Chemistry Reviews*, **322**, 142-158 (2016)). In summary, the mini-sMMO (sm1) contains diiron per mole of $\Delta H\alpha$ (i.e. the native MMOH α sequence, where diiron active center is located); the local arrangement of atoms around the diiron is in good agreement with the active diiron center of native sMMO; the diiron is comprised of both ferrous and ferric ions; and the portion of ferric ions significantly increases after methane oxidation begins. Accordingly, it seems reasonable to conclude that the mini-sMMO (sm1) contains native sMMO-like diiron catalytic center.

We modified the relevant sentence in the text (p. 11) of revised manuscript as follows: “The significantly higher portion of Fe(III) in sm1-afr demonstrates that the diiron center of sm1 follows a similar catalytic cycle as that of native MMOH α ⁵³.”.

[Comment 13]

Figure 4d and 4e - The EXAFS is presented without any fits, quantification of N, O, Fe, DW factors for different trial fits, or other evidence that the assigned distances correspond to what it claimed.

Revision:

The results of X-ray spectroscopy of sm1-bfr and sm1-afr (Figs. 4c-e) are plotted using the raw data provided by the Aichi Synchrotron Radiation Center (Aichi, Japan) without any fitting process. The meaning of “any fits” in the comment above is not clear to us. To double-check the EXAFS results, we investigated the local arrangement of atoms around diiron center using crystal structures of native MMOHox (PDB 1FZ1) and MMOHred (PDB 1FYZ) and PyMOL software. We estimated the local distance between one iron atom of diiron center of MMOHox and MMOHred and each of other atoms that are

located within 5 Å from the diiron center and calculated the average distance. The results in the attached figure below show that the first and second nearest atoms are all O/N; the third nearest atom is another iron; and the distance between one iron and second nearest atom and the inter-iron distance are nearly the same as the EXAFS results (Figs. 4d,e). We added the figure below to Supplementary Information (Supplementary Fig. 13), and the following sentence was added to the text (p. 11) of revised manuscript: “The similar results were also obtained through analyzing the crystal structures of oxidized (PDB 1FZ1) and reduced MMOH (PDB 1FYZ) as presented in Supplementary Fig. 13.”

Native sMMOH-ox (1FZ1)			Native sMMOH-red (1FYZ)		
Scattering Path	R (Å)	SD.	Scattering Path	R (Å)	SD.
Fe-O/N	2.12	0.011	Fe-O	2.15	0.058
Fe-O	2.4	-	Fe-O/N	2.35	0.058
Fe-Fe	3.2	-	Fe-Fe	3.3	-

[Comment 14]

1)Line 290 “methanol production by methanotrophs (0.002 to 0.005 g/L/h)” I believe this is severely underestimated. The referenced papers are using formate as a source of energy and as a means presumably to slow methanol to formate conversion in the culture. Blocking this conversion is the only way any methanol would be observed in solution. This is not representative of methanol production by freely growing methanotrophs.

2)The literature reports 25 g/L of cell paste during an 8 h growth of methanotrophs on methane in a fermentor at an OD600 of 8. All of the carbon would be derived from CH₄ (and subsequently methanol) so if 25% of the cell paste was carbon, this gives 6.25g/L divided by 8h = 0.78 g/L/h. This being compare to methanol production from an OD600

= 40 E. coli culture, so the value for native methanotrophs could be as high as 3.9 g/L/hr at an equivalent cell mass. The native fermentation is run by dilution and regrowth of the bacteria during the 8 h period, so only at the end of each fermentation period is the maximal methanol per hour being produced. Consequently, the estimated 0.78 g/L/h average would be much lower than the actual rate of production. Since the turnover number for isolated MMOH/MMOB/MMOR is about 2.5 s⁻¹ at room temperature, and the recalculated value for the isolated mini-MMO is 0.03 s⁻¹, it seems unlikely that in whole cell E. coli, the mini-MMO is now 100 fold better than the native methanotroph.

Revision:

1) Methanol accumulation hardly happens in freely growing methanotrophs because methanol is utilized as a sole carbon and energy source as soon as it is produced through methane oxidation by MMO. Therefore, quite many researchers have added formate to methanotroph cultures either to slow down methanol oxidation or to regenerate reducing power. Some people have used phosphate or EDTA to inhibit the activity of methanol dehydrogenase (mdh) (*J. Chem. Technol. Biotechnol.* **92**, 311-318, (2017); *Biotechnol. biofuels bioprod.* **15**, 1-13, (2022); *Bioresour. Technol.* **102**, 7349-7353 (2011); *Appl. Microbiol. Biotechnol.* **83**, 669-677, (2009)). Here for rapidly producing a large quantity of methanol, we used *E. coli* as a host for the mini-sMMO expression, because *E. coli* does not have any methanol oxidation activity and grows quickly to high cell density. As a result, the methanol of ~ 3 g (dry cell mass)/L was produced for the entire culture period (27 h) covering initial cultivation, mini-sMMO expression, and methane oxidation phases, yielding the productivity of 0.11 g/L/h.

Reportedly, the highest methanol yield and productivity are 1.34 g/L and 0.332 g/L/h, respectively, in the methanotroph culture (*Methylomonas* sp. DH-1) at 2.4 g (dry cell mass)/L (OD 9), which contains sodium formate (40 mM) and EDTA (0.5 mM) (*J. Chem. Technol. Biotechnol.* **92**, 311-318 (2017)); however, this productivity is estimated for only 4-h period of methane oxidation. According to another literature published by the same research group (*Biotechnol. Biofuels Bioprod.* **15**, 1-13, (2022)), it takes 184~194 h until the *Methylomonas* culture reached OD 9, meaning that the methanol productivity for the entire culture period is only 0.007 g/L/h. The second-best methanol yield and productivity are 1.13 g/L and 0.028 g/L/h, respectively in the methanotroph culture (*Methylosinus trichosporium* OB3b) at 17.3 g (dry cell mass)/L (OD 25.8), which contains sodium formate (20 mM) and phosphate (400 mM) (*Bioresour. Technol.* **102**, 7349-7353, (2011)). Similarly, however, this productivity estimation is for 40 h of methane oxidation.

Considering that it takes 240 h until the culture density of the same *M. trichosporium* OB3b reaches 14 g (dry cell mass)/L (*Appl. Microbiol. Biotechnol.* **83**, 669-677 (2009), published by the same research group), the methanol productivity is only 0.004 g/L/h. Consequently, the methanol productivity of 0.11 g/L/h, achieved in this work, is evidently best-ever performance, not overestimated result.

2) Because the reviewer did not suggest the published references concerning the comment above (including a very surprising result that methanotroph grew to 25 g (dry cell mass)/L from OD 8 within 8 h), we tried to find the references but failed. Even if methanol production is not directly proportional to cell mass concentration (i.e., methanol yield decreases as cell mass concentration increases to such a high concentration (*J. Chem. Technol. Biotechnol.* **92**, 311-318 (2017))), the reviewer simply assumed that 25% of 25 g/L cell paste comes from methanol (6.25 g/L); however, this means that all methanol is completely consumed for cell mass production, that is, no methanol remains in the culture. As stated in the paragraph above, it is extremely difficult to produce a large quantity of “extra” methanol in methanotrophs because methanol is used as a primary carbon source. This is the reason why we used *E. coli* as a methanol-producing host in this work, leading to the high yield and productivity of “extra” methanol, which has been never achieved using methanotroph cultures.

Furthermore, in this work, the *E. coli* culture at OD 40 was used only in closed vial system where the methane supply is severely limited, and thus methanol productivity was not assessed with this system. Methanol produced in the closed vial system was used for C¹³-NMR analysis for investigating whether methanol is produced through the oxidation of added methane by recombinant *E. coli* cells. And the turnover frequency (TF) of mini-sMMO (sm1) was 0.32 s⁻¹, estimated through *in vitro* methane oxidation (p. 10 of the revised manuscript), not through the whole cell experiment. Reportedly, the TF of native sMMO is 0.2 to 1.0 s⁻¹ (*Biochemistry*, **54**, 14, 2283-2294, (2015)).

Reviewer #3:

[Comment 1]

The introduction is rather dense and would benefit from significant rewriting to make the manuscript more compelling. Specific comments about the introduction: The MMOH enzyme is not in a dimer organisation, which it has dimers of all the subunits, the overall arrangement of A2B2G2 is a heterohexamer. Please clarify here.

Revision:

We rewrote the Introduction as marked by blue color with changing the word, “dimer” to “heterohexamer” in the text (p. 3).

[Comment 2]

The roles of the MMOH in gating and electron transfer should be explained more clearly.

Revision:

We added the following sentence to the Introduction (p. 3) of revised manuscript to explain the role of MMOH and MMOB more clearly: “...a regulatory enzyme (MMOB) alternates between binding to MMOH and MMOR, the binding pair depending on the phase of the catalytic cycle of sMMO and plays a critical role in regulating proton transfer and methane/O₂ gating³⁴⁻³⁶.”. The electron transfer to diiron active center is described in detail in the sub-section (entitled, “Important considerations for the design of mini-sMMOs”) (p. 4-5) of Results of the revised manuscript.

[Comment 3]

High-throughput biosynthesis... using *E. coli* - reword for clarity. Is it really high throughput just growing *E. coli*?

Revision:

“High-throughput” was used to describe the rapid and high-yield biosynthesis of mini-sMMO in *E. coli* culture. The word, “high-throughput” was changed to “rapid and high-yield” in the text (p. 4) of revised manuscript.

[Comment 4]

Summarise key results at the end of the methods section please.

Revision:

We added the brief summary at the end of lengthy sub-sections of Methods as follows.

Preparation of genetically engineered recombinant enzymes/proteins

“In summary, we successfully constructed the plasmid expression vectors (Supplementary Fig. 2) for the recombinant synthesis of sm1, sm2, sm3, sm δ , sm1-cat_{mut},

sm1-FAD_{mut}, sm1-e_{mut1}, sm1-e_{mut2}, sm1-e_{mut3}, sm1-e_{mut4} and wild-type huHF in *E. coli* and finally selected the antibiotic-resistant transformants.”

Biosynthesis and purification of recombinant enzymes/proteins

“In summary, we established the protocols for synthesizing sm1, sm2, sm3, sm1-cat_{mut}, sm1-FAD_{mut}, sm1-e_{mut1}, sm1-e_{mut2}, sm1-e_{mut3}, and sm1-e_{mut4} through culturing the recombinant *E. coli* and for purifying the recombinant enzymes/proteins.”

Biosynthesis, purification, and refolding of sm δ

“As a result, sm δ synthesized as a insoluble protein in *E. coli* was successfully refolded to a soluble protein.”

Characterization of mini-sMMOs (sm1 to sm3) and sm1 mutants

“In summary, with mini-sMMOs (sm1 to sm3) and/or various sm1 mutants (sm1-cat_{mut}, sm1-FAD_{mut}, sm1-e_{mut1}, sm1-e_{mut2}, sm1-e_{mut3}, and sm1-e_{mut4}), their characteristics - morphological shape, hydrodynamic size, iron content, stoichiometry of α - and β -subunits, localization of R_{FAD}- Δ H α and retroB on apoferritin scaffold - were analyzed using appropriate analytical tools.”

Analytical methods to measure the amount of methanol

“In summary, methanol production in closed vial systems and fed-batch cultures of sm1-expressing *E. coli* was monitored using GC and HPLC, respectively.”

X-ray absorption spectroscopy and EPR analyses

“In summary, with the freeze-dried samples of sm1-bfr and sm1-afr and/or their mutants (sm1-emut1-bfr and sm1-emut1-afr), the oxidation/reduction status of diiron center was analyzed using XANES and EPR (5 K), while the local arrangement of atoms around diiron center was analyzed using EXAFS.”

NMR spectroscopy analysis

“As a result, ¹³C-methanol produced from the ¹³C-methane oxidation by the purified sm1 or sm1-overexpressing *E. coli* cells was successfully detected through ¹³C-NMR analysis described above.”

***In situ* methane oxidation in the high-cell-density cultures of sm1-expressing *E. coli* under fed-batch operation**

“In summary, the operating procedures and conditions - seed culture preparation and inoculation, medium composition for initial batch and subsequent fed-batch cultures, and feeding strategy and operating conditions of fed-batch cultures - for *in situ* methanol production in the high-cell-density cultures of sm1-expressing *E. coli* were described above in detail.”

[Comment 5]

Results - Table 1 isn't really a table, it is more of a figure and the layout is confusing with all of the domain boundaries shown above the cartoon of the constructs. Can this be clarified and inserted as a figure rather than a table. This is a key figure and needs to be clear to communicate your designs.

Revision:

Upon the manuscript revision, Table 1 was modified to describe each domain boundary and domain assembly of mini-sMMO constructs more clearly.

[Comment 6]

P5 of PDF - 'we first slimed down...' do you mean slimmed down?

Revision:

We corrected the typo-error by changing “slimed” to “slimmed”.

[Comment 7]

Human H-ferritin - can you be sure that the ferritin in the growth conditions you use, especially those with iron, is actually free of iron? Ferritin is a great scaffold, but it is not an inert scaffold by any means. Particularly in the case here where you have an iron-dependent redox enzyme. You present results in Figure 3b to show that the HuHF does not produce methanol in your assay, however you should introduce a control experiment in which the FOC of the HuHF is mutated to inactivate it to control for the potential interaction/influence of the HuHF with the attached enzymes. Would also be instructive to show a HuHF control micrograph in figure 3A.

Revision:

In our previous report (*Nat. Catal.* **2**, 342-353 (2019), ref. 23), we already proved that huHF scaffold contains no iron, and thus we call huHF apoferritin scaffold.

To examine whether mini-sMMO (sm1) has the activity of FOC (ferroxidase center, located in the internal surface of huHF^{47,48}) or not, we performed xylenol orange assay (see the assay protocol described below) with sm1, huHF scaffold, apoferritin standard (Sigma Aldrich), and HBVC (hepatitis B virus capsid, negative control). The results are presented in the figure below, demonstrating that the FOC activity of sm1 is negligible, i.e. nearly equal to that of HBVC (negative control), while both of huHF scaffold and apoferritin standard show FOC activity. This implies that ferrous ions cannot be transported to the internal FOC of huHF, which is probably because the ferrous ion transport is severely blocked by R_{FAD}-H₂O and retroB that are densely located on the surface of huHF scaffold (Table 1, Supplementary Fig. 5). Accordingly, it is concluded that the methane oxidation by sm1 is not influenced by the FOC within the apoferritin scaffold. We added the figure below to Supplementary Information (Supplementary Fig. 6) and the following sentence to the text (p. 9) of revised manuscript: “Furthermore, we verified that sm1 shows a negligible activity of FOC (ferroxidase center, located in the internal surface of huHF^{47,48}) (Supplementary Fig. 6), meaning that methane oxidation is not interrupted by FOC within the apoferritin scaffold.”

Xylenol Orange assay: The assay is conducted at 30 °C by adding Fe²⁺ to a mixture of protein/enzyme (0.8 μM), Tris-HCl buffer (50 mM, pH 7.0), and (NH₄)₂Fe(SO₄)₂ (40 μM). A 10-μl aliquot is taken at 5 min after the assay reaction started and mixed with a 100- μl solution of Xylenol Orange (XO) (125 μM) and H₂SO₄ (25 mM). After incubation at room temperature for 5 min, the concentration of ferric ions is determined by measuring the absorbance at 595 nm. The auto-oxidation of the ferrous ions is measured under the same assay conditions but without adding protein/enzyme.

[Comment 8]

Construction of the B-subunit of SM1 - how did you facilitate a C-term-to-C-term fusion, this needs clarification.

Revision:

The sequence of retroB was the reversed sequence of native MMOB, that is, the N- and C-terminus of retroB corresponds to the C- and N-terminus of native MMOB, respectively. The coding sequence of retroB was synthesized by a company (Bionics, Seoul, Republic of Korea) and was fused to the huHF-coding sequence so that the resulting fusion gene codes for the synthesis of fusion protein with the N-to-C orientation from huHF to retroB, where the N-terminus of retroB is linked to the C-terminus of huHF.

Reviewer #4:**Major comments:****[Comment 1]**

Earlier work by some of the authors (Nat Catal 2019, ref 17) has used effectively the same strategy of scaffolding on the particulate MMO (pMMO). The structure of the pMMO differs from the sMMO in respect of the electron supply chain but the hydroxylase unit is really very similar. Ref 17 is cited only once in the Introduction amongst other papers, apparently as a note in passing. The whole manuscript is then written to convey the idea that everything done in the current study is new. This reviewer noted only on third reading of the manuscript how important as precedent reference 17 is. The whole experimental approach is extremely similar in the two studies. It can be useful to compare the figures for similarity in appearance. This is not to say that the current study does not have important elements of novelty. But reference 17 certainly compromises novelty/originality of the approach.

Revision:

Both of our current and former study (*Nat. Catal.*) use commonly apoferritin that has very advantageous features such as high overall stability, flexible surface topology, narrow surface area around each 4-fold axis, *etc.*; however, apoferritin plays a quite different role in each study. Furthermore, the catalytic domains or components (assembly units) to be assembled on the surface of apoferritin is totally different in terms of sequence and conformation of each assembly unit, intrinsic nature of inter-unit interaction, and catalytic

mechanism to be preserved. The former study is about the construction of catalytic mimics of pMMO ($\alpha\beta\gamma$, i.e. homo-trimer of a single chain, $\alpha\beta\gamma$), which is a highly hydrophobic, membrane-associated enzyme. The α subunit (pmoB) of pMMO has two soluble, periplasmic domains (pmoB1 and pmoB2, interacting closely with each other) that are linked to a couple of insoluble, transmembrane domains, pmoB1 having two catalytic sites A and B. In this study, apoferritin is used as a scaffold to replace the transmembrane region of native pmoB, that is, the pmoB1 and pmoB2 were linked in parallel to the 4-fold axis of apoferritin and thus were localized on the surface of apoferritin with closely interacting with each other, resulting in the construction of pMMO-mimetic enzymes. Consequently, the apoferritin scaffold is a simple substitute of transmembrane region of parent enzyme (pMMO).

On the other hand, the current study is about the construction of miniaturized enzymes (mini-sMMOs) of sMMO, which has a highly complex structure, comprised of three components (MMOH, MMOR, and MMOB). MMOH is a hetero-hexamer ($\alpha_2\beta_2\gamma_2$) with diiron active center; MMOR has two electron transfer domains (FAD and ferredoxin domain); and MMOB interacts alternately and precisely with MMOH and MMOR for efficient proton and substrate (methane and oxygen) transfer to catalytic center. Therefore, the construction of mini-sMMOs requires a highly exquisite work, including the optimal design of functionally active, minimal or modified version of three components, and their assembly with preserving native catalytic function of sMMO. Here apoferritin is NOT a simple substitute of a particular component of parent enzyme BUT a novel platform that enables parent enzyme complex (sMMO) to be miniaturized with preserving the native catalytic function through exquisitely assembling the minimal or modified versions of functionally and structurally different components of sMMO, leading to the construction of new enzymes derived from sMMO.

[Comment 2]

The idea of shortening the electron transfer from NADH in the miniature MMO is interesting. However, it seems to be not fully worked out. The constructs should include a complete reductase unit (FAD domain and ferredoxin domain) and show how it compares to the redesigned unit containing only the FAD domain. My concern is this: a suitably exposed FAD domain will easily become reduced by NADH but the efficiency of electron transfer to the metal center is not clear. The authors should compare the ratio between NADH oxidized and methanol formed (a sort of coupling efficiency of the catalytic steps). It could easily be that the miniature MMOs show strongly altered

coupling. An enzyme that requires many molecules of NADH to get a single molecule of methane hydroxylated is not an efficient catalyst. Coupling efficiency might represent a critical limitation of artificial MMOs. The FAD reduced from NADH may get reoxidized by O₂, resulting in peroxide and other protein-damaging reactive oxygen species. The structural control (full reductase unit) should be performed and the coupling efficiency of the construct is a critical to-do.

Revision:

Through additional experiment, we constructed another mini-sMMO by replacing R_{FAD} (FAD domain of MMOR) of sm1 with a full reductase unit of MMOR (FAD domain + ferredoxin domain), named sm1-MMOR. Then, we performed the *in vitro* methane oxidation in closed vial systems, which started with enzyme (sm1 or sm1-MMOR) at 1.25 μM and NADH at 0.3 mM, and estimated coupling efficiency (methanol produced per NADH consumed) through measuring methanol and NADH concentration in the reaction solution at the pre-determined time points (4, 8, 12 h). As shown in the figure below, the coupling efficiency of sm1 is notably higher than that of sm1-MMOR at all time points, suggesting that the electron transfer to the diiron center is far more efficient in sm1 ([NADH]– [FAD]– [diiron center]) than in sm1-MMOR ([NADH]– [FAD]– [Fe-S]– [diiron center]).

We added the figure above in Supplementary Information (Supplementary Fig. 11) and the following sentence to the text (p. 10) of revised manuscript: “To evaluate the catalytic efficiency in sm1, we estimated methanol-NADH coupling efficiency (ratio of methanol produced per enzyme (mol/mol) to NADH consumed per enzyme (mol/mol)) in the *in vitro* methane oxidation by sm1 and sm1-MMOR (that was constructed through replacing R_{FAD} of sm1 with a complete reductase unit of MMOR (FAD domain + ferredoxin domain)), demonstrating that the coupling efficiency of sm1 is always notably higher than that of sm1-MMOR (Supplementary Fig. 11) and thus suggesting that R_{FAD} of sm1 provides an efficient passage of electrons from NADH to diiron catalytic center.”

[Comment 3]

The evidence on *E. coli* strain for production of methanol is extremely preliminary and does not support the far-reaching claims (e.g., biomanufacturing etc.) made. Point of the reviewer is not that there is room for optimization, it is conceptual. The *E. coli* strain is first grown to a relatively high cell density (OD of almost 80!) and then methanol production is initiated by induction of sMMO synthesis. The system will burn glucose mostly for aerobic metabolism and just a very tiny part of it will go into methanol production. It seems relatively easy to find conditions in which such a process shows a higher productivity than a process based on methanogens that use completely different substrate conditions and lack the growth to high cell density. But the level of comparison is irrelevant and in no way does the higher productivity imply that the *E. coli* process would be more promising for a practical application.

Revision:

sMMO can oxidize a wide range of carbon feedstocks (C1 to C8) directly using intracellular NADH and thus is a promising enzyme in developing environmentally and energetically benign, industrial manufacturing of chemicals. However, despite a longterm, tremendous effort through engineering methanotrophs or recombinant expression of sMMO, low yields and conversion efficiencies remain the major hurdle in developing methanotroph- and/or sMMO-based, economically viable biomanufacturing processes. As a means to overcome this unresolved issue, the high-throughput production of active recombinant sMMO in *E. coli* - which does not have a methanol oxidation pathway unlike methanotrophs and rapidly grows to a high cell density even at industrial scale - has

attracted much attention but long been exceptionally difficult because of the structural and functional complexity of sMMO comprised of three components - MMOH, MMOR, and MMOB.

In this work, we constructed an mini-sMMO (sm1) through a molecular editing of sMMO - which is based on an exquisite reassembly of the minimal sub-structures and modified version of the three sMMO components on the catalytically inert and stable huHF scaffold - and developed the sm1-overexpressing recombinant *E. coli* system, leading to the remarkable (22 to 55-fold) improvement in methanol yield and productivity through the *in-situ* methane oxidation in the high-cell-density culture of the recombinant *E. coli*. The current result seems preliminary at this stage, but a more systematic search for the optimal set of important variables (e.g. feeding strategy of fed-batch operation for enhancing intracellular reducing power and methane oxidation rate, bioreactor design to increase both dissolved methane concentration and methanol recovery yield, production scale, *etc.*) could improve further the methanol-manufacturing performance of this recombinant bacterial culture to an economically viable level. We believe that this novel approach offers a promising platform for developing sMMO-derived industrially useful bioprocesses that enable the production of a variety of value-added chemicals.

[Comment 4]

Determination of the intracellular NADH requires rapid quenching of the cellular metabolism. In the way analyzed, the results in Figure 6c are not reliable.

Revision:

NADH concentration of Fig. 6c was measured using the *E. coli* cell pellet prepared immediately after *E. coli* culture samples taken from bioreactor was centrifuged (Methods, p. 21 of revised manuscript). To evaluate the reliability of this assay method (“assay A”), we attempted another assay protocol (“assay B”) which is based on quenching the cell pellet of “assay A” in liquid nitrogen for 30 sec prior to the assay. The performance of two assay methods (A and B) was compared in the figure below, showing no significant difference in NADH and NAD⁺ concentration measured by the two assay methods and thus indicating that the intracellular NADH concentration of Fig. 6c was measured using a reliable assay method.

A: assay with *E. coli* cell pellet prepared immediately after
E. coli culture was sampled from bioreactor and subsequently centrifuged
 B: assay after quenching the *E. coli* cell pellet of A in liquid nitrogen for 30 sec

[Comment 5]

The mutagenesis results identify residues somewhat important for activity, but it is not clear how they would reveal the electron transfer path.

Revision:

First, we performed EPR analysis with the samples of sm1-bfr and sm1-afr (Figs. 4a,b). sm1-bfr is the sm1 prepared before used for *in vitro* methane oxidation, while sm1-afr was the sm1 sampled after 16-h methane oxidation in the presence of NADH. Fig. 4a shows that the ratio of Fe⁺² to Fe⁺³ in the sm1-bfr is far higher than 1.0, meaning that a larger portion of sm1 in the sm1-bfr is at reduced state, which is probably because sm1 was synthesized in *E. coli* cytoplasm containing a substantial amount of NADH. The ratio of Fe⁺² to Fe⁺³ in the sm1-afr, sampled during actively producing methanol, notably decreased below 1.0. (Fig. 4b), meaning that the portion of oxidized sm1 in the sm1-afr significantly increased, which is well explained based on the catalytic cycle of sMMO (shown in the attached figure below).

Then, we synthesized a sm1 mutant (sm1-e_{mut1}) through replacing the three hydrophilic

Curr Opin Chem Biol. **6**, 568–576 (2002).

residues (Y, T, and H, which are presumed to be key residues in electron shuttling path from FAD to diiron center) with hydrophobic residues (F and A) and similarly prepared the two different samples, sm1-e_{mut1}-bfr and sm1-e_{mut1}-afr. The EPR results of sm1-e_{mut1}-bfr and sm1-e_{mut1}-afr (Fig. 5h,i) are totally different from the EPR results of sm1-bfr and sm1-afr (Figs. 4a,b). That is, the ratio of Fe⁺² to Fe⁺³ in sm1-e_{mut1} is always below 1.0, which means that the electron transfer to diiron center in sm1-e_{mut1} is significantly less efficient than in sm1. Further, the methanol production by sm1-e_{mut1} is negligible compared to that by sm1 (Fig. 5g). Consequently, from the results of Figs. 4a,b and 5g-i, it seems reasonable to claim that the three hydrophilic residues (Y, T, and H) play an important role in the electron transfer from FAD to diiron center.

Minor comments:

[Comment 1]

The structural description on pages 5 – 8 was very difficult to follow. It was not clear how unique/clear are the results of the molecular dynamics simulations.

Revision:

To more clearly describe the results of the molecular dynamics simulations, we added the following modified sentences to the text (p. 5-6) of revised manuscript:

“Our first step in designing the mini-sMMO was to create a minimal form of MMOH-MMOR complex. The key strategy in creating the minimal model involved removing the ferredoxin domain (a.a.1-98) from MMOR (a.a.1- 348) (Figs. 1a,b) and relocating the remaining FAD domain (a.a. 99-348, referred to as R_{FAD}) close to the diiron center of MMOH, enabling direct electron transfer from FAD to the active site. To achieve this, we performed a series of sequence editing trials, aided by structure stability analysis using various *in silico* approaches including molecular dynamics (MD) and protein-protein docking (PPD) simulations. The relocation of R_{FAD} was attempted by the following two steps. First, a minimal version of hydroxylase (denoted as AHa) was created from MMOHa, retaining only 258 residues (a.a. 64 - 321) out of the full 526 residues, while removing other MMOH subunits, MMOH13 and MMOHy, for the minimal model (Figs. 1c,d). The resulting trimmed sequence covers only the helices from aA to aH, without MMOH13 and MMOHy, which exposes aA, aB, aC, and aD and provide more potential binding sites for the R_{FAD} (Fig. 1e).

The second step involved identifying R_{FAD} candidate that poses on the $\Delta H\alpha$ surface using PPD simulations. This revealed a bimodal distribution with R_{FAD} binding to either the canyon or the region near aA, aB and aC (referred to as N-terminal half (NTH) region) (Fig. 1e). High-scoring poses were predominantly found in the NTH region, which were further narrowed down by constraining the distance between the C-terminus of R_{FAD} and the N-terminus of $\Delta H\alpha$ to within 10 Å. This constraint was introduced to consider only configurations of R_{FAD} linked to AHa by forming a single chain with the N-to-C orientation from R_{FAD} to AHa (denoted as R_{FAD}-AHa, where the C-terminus of R_{FAD} is linked to the N-terminus AHa).”.

[Comment 2]

What is the optimal region (page 5)? Which constraints were used and why? (page 5)

Revision:

The phrase, “optimal region” sounds somewhat ambiguous, and we modified the relevant sentence. That is, the sentence, “...relocate the remaining FAD domain (a.a. 99-348, denoted as R_{FAD}) to an optimal region near the diiron center of MMOH α .” was changed to the sentence, “...relocating the remaining FAD domain (a.a. 99-348, referred to as R_{FAD}) close to the diiron center of MMOH,...” as shown in the text (p. 5) of revised manuscript. In the response to minor comment 1 of reviewer 4, we explained in detail about the design strategy and constraint we used for constructing the mini-sMMO. Please see the response above.

[Comment 3]

The size of the assembled particles varies from 18 – 19 nm in this study and the earlier Nat Catal paper shows even larger particles. It is not clear how these sizes inform about protein exposed on the surface.

Revision:

The mini-sMMOs of this study were constructed through localizing R_{FAD-H} and B/retroB on the surface of apoferritin, while the pMMO-mimics of Nat. Catal. display the two periplasmic domains of pmoB (pmoB1 and pmoB2) on the apoferritin surface. The size of the mini-sMMOs and pMMO-mimics is, of course, larger than apoferritin (~12 nm), i.e. the average diameter of sm1, sm2, pMMO-m2, pMMO-m3, and pMMO-m4 is 19.6, 19.1, 20.8, 23.9, and 21.9 nm, respectively. (We excluded sm3 and pMMO-m1 because sm3 contains only R_{FAD-H}, and in pMMO-m1, pmoB1 and pmoB2 are serially linked and fused to the apoferritin scaffold.) Because the exact particle size depends on the conformation/structure and orientation of the surface-localized polypeptides (R_{FAD-H}, B, retroB, pmoB1, and pmoB2), it is presumed that the pmoB1 and pmoB2 are slightly (0.5~2 nm) more protruded outward from the apoferritin surface.

[Comment 4]

The characterization of the active sites (metal center, FAD) appear to have been carefully done and is interesting. But it wasn't always clear why these characterizations are needed. For example, the detailed assessment of the iron center with EPR and x-ray absorption.

Revision:

The objective of this study is to create structurally miniaturized but catalytically active enzymes (mini-sMMOs) derived from native sMMO, comprised of MMOH, MMOR, and MMOB. The mini-sMMOs were designed to retain diiron catalytic center of MMOH, electron transfer domain (FAD domain) of MMOR, and modified MMOB with reversed sequence. As demonstrated, one of the mini-sMMOs (sm1) showed clearly high methane-oxidizing activity like native sMMO. Then we needed to clarify 1) whether the activity of sm1 is actually based on catalytic performance of diiron center, 2) how efficiently the electrons transfer to diiron center via FAD domain, and 3) whether the modified MMOB and the minimal version of MMOH/MMOR interact actively with each other. For clarifying the first issue above, it is essential to analyze the changes in valence state (reduced/oxidized state) of diiron center before and after the methane oxidation begins,

which is the reason why EPR and X-ray spectroscopy analyses were performed.

REVIEWER COMMENTS

Reviewer #1 (Remarks to the Author):

The Authors have addressed most of the issues that I raised on the original version, including addition of substantial new data. The result is a manuscript that describes a major achievement in development of a biocatalyst for oxidation of methane that does not have the problem of further metabolism of the methanol product that is seen with wild-type methanotroph systems. A small number of issues need to be addressed, as indicated below.

1. Page 3. Last sentence of first paragraph. In the revised text here, it would be relevant to mention the study of Zill et al. (DOI 0.1002/cbic.202200195), which achieved expression of a Methylomonas sMMO in *E. coli*.
2. Page 9. End of first paragraph. In the revised text: “meaning that methane oxidation is not interrupted by FOC within the apoferritin scaffold.” This was unclear to me. Maybe the Authors mean that the sMMO-derived structures occlude the potential iron-binding centres of the scaffold?
3. Supplementary Figure 12. The insets in the graphs are not sufficiently described. What samples (native MMOH expressed in the original methanotroph?) were used for the spectra in the insets, and what comparison to the data in the main graphs should the reader make?
4. Page 11. First paragraph. “The significantly higher portion of Fe(III) in sm1-*afr* demonstrates that the diiron center of sm1 follows a similar catalytic cycle as that of native MMOH α ” – this is still an overinterpretation of the data. I suggest rewording to say that this result is consistent with a similar catalytic cycle to native MMOH α .
5. Page 12. “indicating that electron transfer to the diiron center hardly happens” – this is an overinterpretation. It would be fine to say that the result is consistent with abolition of electron transfer in the mutant.
6. Page 13. Four lines up from the bottom. “fed-bath” should be “fed-batch”.
7. Page 15. First paragraph. The careful analysis of the recombinant proteins that the Authors report give some insights into the possible electron transfer pathway, but do not establish it in detail as the Authors claim here. A more circumspect form of wording is needed.
8. Page 15. Second paragraph. To be more specific about the nature of the major achievement that the Authors have made, I recommend replacing “best-ever strategy” with a more specific statement, maybe that this study reports the highest rate of methanol production from methane per unit volume of culture.

Reviewer #2 (Remarks to the Author):

The authors have improved several aspects of this manuscript, notably a better presentation of the methods used. However, the references to the literature for the structure and reaction of sMMO in the introduction remain incomplete, and the characterization of the metal site in the preparations of the mini-sMMO remains quite problematic (see below). The fundamental discovery of a remarkable reworking of the sMMO system to allow expression in *E. coli* with high methanol production represents an important advance. The comparison to the metal center structure and the reaction cycle dynamics of the native sMMO system would be great to include, but it is not necessary for the fundamental story. In the revision, the authors try again to demonstrate the presence of the diiron cluster and to associate observed oxidation of the iron in the sample over a 16 h period to the native reaction cycle. Neither goal is accomplished, and it detracts from the overall goal of the study. In my opinion, the authors should consider omitting the comparison to sMMO mechanism and try simpler techniques to demonstrate a diiron cluster in the mini-sMMO.

The addition of the spectra of the azide complex is a step in the right direction toward demonstrating a diiron cluster, but the intensity of the spectrum presented is far below that expected for a diiron cluster. There is a good example of the expected spectrum in the reference cited by the authors (PNAS, 107, 15391-15396, (2010), figure 2). It appears from spectrum in the revised manuscript that some fraction of the mini-sMMO may be in the diiron cluster state, but the amount is unknown without further comparison to examples from the literature. Similarly, the measurement of the EPR at 5K is a good step. However, the simple additional step of adding dithionite to an anaerobic liquid sample, freezing the sample, and then attempting to detect the $g = 16$ EPR signal was not done. This signal could only emerge from a ferromagnetically coupled diferrous diiron cluster, and it would be a straightforward way to provide evidence that it is present. Instead, it is reported that no $g = 16$ signal is seen when reducing with NADH. NADH in combination with the reductase should have reduced the cluster to the diferrous state based on the redox potentials of the native cluster in sMMOH. However, the process used of freeze drying the sample and then attempting to interpret a powder sample is unlikely to give interpretable results. The procedures for doing this experiment have been reported many times in the literature.

The EPR spectra shown in Figures 4a,b and 5h,i from power samples are difficult to interpret in the context of a diiron cluster. Perhaps labeling the figures with g -values and enlarging the $g = 2$ region would help the reader. However, the presentation is confusing because Fe(II) has no EPR signal in the way these spectra were measured (perpendicular mode EPR), so the authors may be attempting to interpret them based on fits to a spin coupling model. If so, the details and assumptions of this method are not made clear. The signal the authors attribute to Fe(III) at 150 mT ($g = 4.3$) is due to mononuclear Fe(III) not diiron cluster iron. The very broad signal shown in the figures is often due to non-specifically bound iron rather than the typical sharp signal from active site mononuclear Fe(III). The observation of this signal shows that at least some of the iron quantified in the sample is not in a diiron cluster. The EPR

signal for the mixed-valent form of the diiron cluster has 3 g-values, all below $g = 2$. This is quite characteristic for all antiferromagnetically coupled diiron clusters and would be another good way to argue for the presence of a diiron cluster in mini-sMMO if clearly presented. See for example in the paper cited by the authors, *J. Am. Chem. Soc.*, 115, 3688-3701 (1993), figure 7. In contrast, the spectra shown in Fig. 4 have many signals in the $g = 2$ region, both above and below $g = 2$. The superposition of the bold simulation spectra over the data makes it impossible to see the data for accurate comparison. It is not possible from this figure to identify the characteristic signals of a diiron cluster, and there is nothing here that would allow a Fe(II) to Fe(III) ratio to be assigned.

The authors argue that the observation of the signal from Fe(III) at 150 mT means that the signal seen in Fig 5i (and maybe others) is not due to Cu. However, to me the spectrum in the $g = 2-3$ region looks like classic type II copper (for example, see Fig 1 of *Angew. Chem. Int. Ed.* 2007, 46, 1992–1994)

The other spectra in Fig 4 designed to either show the presence of a diiron cluster or its oxidation during turnover also suffer from over-interpretation and/or lack of detail in describing how they were interpreted. Overall, in my opinion, the entire premise that the oxidation of iron over a 16 hour reaction somehow aligns the mini-sMMO reaction with the reaction cycle of the native enzyme is suspect. In the native cycle, the iron is reduced and oxidized each cycle on a seconds time scale. It oscillates between diferric, diferrous, diferric, di-Fe(IV) and back to diferric, all at 3 times per second at room temperature. What happens on an hours time scale is completely unrelated to this cycle. The XANES data shown in Fig. 4 can be interpreted to show a shift from Fe(II) to Fe(III), but this is on freeze-dried samples with oxidation occurring over 16 hours. It is not realistically possible to relate this data back to the native enzyme cycle. Also, the claim that this data is consistent with a mixed-valent diiron cluster of the enzyme is not correct. The data shows the presence of Fe(III) and Fe(II), not their presence in a coupled cluster. As previously pointed out, the EXAFS data in Fig 4d is shown without fits or Debye-Waller factors to characterize the quality of the fits. An EXAFS fit is based on distance, number of atoms and their electron count relative to a given iron. Without fits to show that the position and intensity of 2 irons at 3.2 Å (in combination with the other atoms expected in the MMOH cluster ligation) gives a better fit than other possibilities, the spectra are meaningless. The claimed shift due to iron oxidation in the peak assigned to the Fe-Fe scattering is too small to claim given the quality of the data and intensity of the peak. Moreover, the claimed shift (0.02 Å) is much too small based on the most recent EXAFS measurements cited by the authors, where it is shown to be about 0.3 Å (see *J. Am. Chem. Soc.* 2018, 140, 16807–16820, table 2).

The addition of CD data to support proper folding of the mutated mini-sMMO is good. The mutations of the proposed diiron cluster ligands could also have been used to support the presence of the diiron cluster and its ligation by looking for loss of the azide complex UV-Vis spectrum, or loss of a $g=16$ in the diferrous dithionite-reduced state, or loss of a g -below 2 three line EPR spectrum in the mixed valent state.

The authors have added the very useful information concerning how the measurement of the turnover frequency was made by stating that it is based on changes only the beginning of the reaction. It is unclear why the turnover frequency decreases during the reaction, but there are many possibilities that do not affect the inherent maximum turnover frequency. Based on comparisons to the turnover frequencies reported for several growth studies by groups trying to develop industrially relevant methanol production by methanotrophs, the authors argue that the native sMMO enzyme system has a much lower turnover frequency and ability to produce methanol than mini-sMMO. I continue to believe that the authors in the current work severely underestimate the turnover frequency of the native sMMO system. The grow characteristics of a commonly used methanotroph as well as the turnover frequency of the native sMMO system can be found in early papers describing the *M. trichosporium* OB3b system (see *Methods in Enzymology* (1990) vol. 188, 191-202). This study (and earlier primary literature on which it is based) showed 25 g/L/h of cells produced by continuous growth at an OD600 of 8. My lab has verified the work reported in the *Methods* article and shown that this rate of production can be sustained for 10 days with production of 3-4 kg of cells. The *Methods* article also lists the turnover frequency for the purified sMMO components for several substrates. For methane, the turnover frequency to yield methanol is reported to be 3.7 s⁻¹ (at 23 °C) which is far greater than the 0.32 s⁻¹ (at 30 °C) claimed for the mini-sMMO system by the authors. It is also in line with the stopped-flow studies of the single turnover rate constants for the purified components, which show about 0.17 s⁻¹ at 4 °C for *M. trichosporium* OB3b sMMO with 10-20 fold higher rate constants at 23 or 30 °C (see for example *Chem. Rev.* 1996, 96, 2625-2657, figure 4).

As stated above, I think the important comparison the authors wish to make is the yield of actual recoverable methanol from the native methanotrophs vs the mini-sMMO. This differs from the comparison of the actual rate of methanol production by methanotrophs prior to it being utilized for cell growth. This comparison of actual recoverable methanol is valid in the current study, but it should not be claimed that the mini-sMMO is comparable to the rate of turnover for sMMO itself.

Line 43 – the regulatory component is not an enzyme

Line 56 – 57 The references cited here do not come close to identifying all of the important contributions to kinetic and NMR characterization of the sMMO cycle and components. If the authors do not want to provide a list that includes the contributions from many research group, then citing only recent reviews is an alternative.

Line 64 – MMOB is not an enzyme

Line 66 - The references cited here do not come close to identifying all of the studies that identified the roles played by MMOB

Line 229 – See comments above for the actual turnover frequency for native sMMO at room temperature. Also Rosensweig is not spelled correctly in ref 19.

Lines 234-241 – This shows that Fe is important for methanol production but not necessarily that it is a diiron cluster. The diiron cluster could be demonstrated directly (see comments above)

Lines 248-267 – Much of this is incorrect or over-interpreted. All the authors need to show is that a diiron cluster is formed (methods listed above). As stated above, it does not seem necessary for the main point of the paper to try to show that the mini-MMO has the same cycle as native sMMO.

Line 264 – The difference in these distances is much too small and are not comparable to the results given in the cited reference.

Line 266 – The relevant comparison here would be to the structures of the completely oxidized and fully reduced MMOH-MMOB complexes (J. Am. Chem. Soc. 2020, 142, 14249–14266, fig 7) which agrees with the EXAFS from Cutsail but not the EXAFS difference in Fe-Fe distance in the current manuscript. The cited PDB structures do not have MMOB associated with MMOH and are not in the pure oxidation states due to reduction in the synchrotron beam.

Line 340 – omit the word “enzyme”

Line 357 – nuclearity was not established

Line 334- end The discussion just restates what is in the results and then says the system the authors have developed can be fine-tuned. None of this is necessary or provided context. A comparison to previously published attempts to make a viable methanol producing system by using methanotrophs or using zeolites with embedded iron sites might be more useful.

Reviewer #3 (Remarks to the Author):

We thank the authors for their consideration of all of the reviewers comments and the valuable additional data presented. The manuscript is now easier to read and the results better contextualised.

We have one further query before the manuscript would be suitable for publication.

The rationale behind the generation of the retro-B construct is still not clear. We thank you for clarifying that the sequence was actually reversed fully. The description of the modelling methods for the retro-B indicate that the structures for MD simulation were taken from the PDB for 4GAM and others. Alphafold predictions of retro-B do not give models with significant fold similarity to MMOB. RMSD Ca of over 10 Å!

Could the authors please comment further on the rationale for the retro-B construct and their modelling and what this might mean for the role of the modified retro-B.

Reviewer #4 (Remarks to the Author):

I agree with the authors' responses and the revisions made.

Revision Details

Reviewer #1:

[Comment 1]

Page 3. Last sentence of first paragraph. In the revised text here, it would be relevant to mention the study of Zill et al. (DOI 0.1002/cbic.202200195), which achieved expression of a Methylomonas sMMO in *E. coli*.

Revision:

The reference that the reviewer suggested (*ChemBioChem* **23**, e202200195 (2022)) is about the heterologous expression of sMMO from *Methylomonas methanica* MC09 through co-expressing *E. coli* chaperonin molecules (GroES/EL) but does not show any verified, qualitative and quantitative characteristics of recombinant sMMO such as structure, function, expression yield, catalytic efficiency, etc. With citing this reference, we added the following sentence in the second revised manuscript (p. 14): “Although the co-expression of recombinant sMMO and *E. coli* chaperonins was recently reported⁵³, the effectiveness of the expression system remains unknown, because any qualitative and quantitative characteristics (structure, function, expression yield, catalytic efficiency, etc.) of the enzyme were not revealed yet.”

[Comment 2]

Page 9. End of first paragraph. In the revised text: “meaning that methane oxidation is not interrupted by FOC within the apoferritin scaffold.” This was unclear to me. Maybe the Authors mean that the sMMO-derived structures occlude the potential iron-binding centres of the scaffold?

Revision:

FOC (ferroxidase center) is located on the internal surface of the huHF scaffold with a hollow, spherical nanostructure, while the native sMMO-derived minimal/modified components (R_{FAD}, ΔH_α, and retroB) of mini-sMMO (sm1) are localized on the outer surface of the huHF scaffold. Reviewer’s concern is that because ΔH_α contains the diiron center that is essential for methane oxidation, the FOC with potential iron-binding site might affect the methane oxidizing performance of sm1. Supplementary Fig. 6 of the revised manuscript demonstrates that contrary to the huHF scaffold and apoferritin standard, sm1 never shows ferroxidase activity, meaning that either ferrous ions cannot

be transported to the internal FOC site or FOC is inactivated, due probably to the significant interference by R_{FAD}, ΔH_α, and retroB that are densely localized on the outer surface of huHF scaffold. Accordingly, it is obvious that the effect of FOC inside the huHF scaffold on the methane oxidizing activity of sm1 is negligible. In the second revised manuscript (p. 9), the phrase (“... meaning that methane oxidation is not interrupted by FOC within the apoferritin scaffold.”) was changed to: “... meaning that the potential iron-binding site of internal FOC does not exert an influence on the methane oxidizing activity of sm1.”.

[Comment 3]

Supplementary Figure 12. The insets in the graphs are not sufficiently described. What samples (native MMOH expressed in the original methanotroph?) were used for the spectra in the insets, and what comparison to the data in the main graphs should the reader make?

Revision:

According to the previous literatures (*Biosci. Biotechnol. Biochem.*, **71**, 122-129 (2007); *Catalysts*, **8**, 582 (2018)), native MMOH shows a unique absorbance peak at 395~420 nm, which results from the oxo-bridged diiron in MMOH. In order to investigate if the oxo-bridged diiron is also present in the mini-sMMO (sm1), we prepared “sm1-afr” sample (see below) and measured absorbance. The results were presented as dashed lines in both of inset and main plots of Supplementary Figure 12, showing the absorbance peak at 395~420 nm (indicated by red arrows). (The “sm1-afr” was the sample taken from the reaction buffer at 16 h after methane oxidation began in the presence of NADH.)

Subsequently, through performing an azide experiment we tried to evaluate if the oxo-bridged diiron of sm1 is di-ferric oxo-bridges. In case of the presence of diferric ions, sodium azide combines with diferric ions, followed by the formation of chromophoric azide-diferric complex, which generates the specific absorbance peaks at 345 and 450 nm (*PNAS*, **90**, 2486-2490, (1993); *PNAS*, **107**, 15391-15396, (2010)). The results are plotted as solid lines in main plot of Supplementary Figure 12a, clearly showing the absorbance peaks at 345 and 450 nm and thus indicating the presence of di-ferric oxo-bridges. In the second revised manuscript, we modified the figure caption of Supplementary Figure 12 to describe the figure more comprehensively as follows: “The red arrows on the dashed lines of inset and main plots indicate the peak at 395-420 nm for oxo-bridged diiron of sm1-afr⁴⁵, while the blue arrows on the solid lines of main plots indicate the peaks at 345

and 450 nm for azide-diferric ion complex⁴³.”

Comment 4]

Page 11. First paragraph. “The significantly higher portion of Fe(III) in sm1-afr demonstrates that the diiron center of sm1 follows a similar catalytic cycle as that of native MMOH α ” – this is still an overinterpretation of the data. I suggest rewording to say that this result is consistent with a similar catalytic cycle to native MMOH α .

Revision:

We agree that the sentence is an overinterpretation of the EPR data and thus removed the sentence. Instead, we discussed about only the co-existence of Fe(II) and Fe(III) with the XANES data. We modified the relevant sentence as follows: “The XANES spectra (Fig. 4b) shows that both Fe(II) and Fe(III) are present in sm1-afr, the portion of Fe(III) being higher than Fe(II).”.

[Comment 5]

Page 12. “indicating that electron transfer to the diiron center hardly happens” – this is an overinterpretation. It would be fine to say that the result is consistent with abolition of electron transfer in the mutant.

Revision:

We agree that the sentence is an overinterpretation of the EPR data and thus modified the relevant sentence in the second revised manuscript (p. 12): “Considering the non-significant change in secondary structures (Supplementary Figs. 14) of sm1 and its mutants (sm1-cat_{mut}, sm1-FAD_{mut}, and sm1-e_{mut1}), the changes in iron content (Fig. 3e, Supplementary Fig. 15) and methanol production (Fig. 5g) are not attributed to the changes in overall structure caused by the mutations and accordingly indicate that the three residues, Tyr305, Thr335, and His336 comprise the electron shuttling path in sm1, indispensable for the catalytic oxidation of methane, Tyr305 and His336 being more important for methanol production than Thr335 (Fig. 5g).”.

[Comment 6]

Page 13. Four lines up from the bottom. “fed-bath” should be “fed-batch”.

Revision:

We corrected the typo-error by changing “fed-bath” to “fed-batch”.

[Comment 7]

Page 15. First paragraph. The careful analysis of the recombinant proteins that the Authors report give some insights into the possible electron transfer pathway, but do not establish it in detail as the Authors claim here. A more circumspect form of wording is needed.

Revision:

In the second revised manuscript (p.15), we modified the sentence with more cautious wording as follows: “Through systematic spectroscopic and spectrometric analyses (¹³C-NMR, EPR, XANES, EXAFS, and ICP-MS) and azide-diferric complex analysis of the mutated and/or intact sm1, we further investigated the iron valence and quantity in active center, existence and location of FAD, and electron shuttling path in detail, demonstrating that through the minimal model design, the FAD electrons are directly transferred to the diiron center without passing through the ferredoxin domain with keeping active orientation and interaction among R_{FAD}, ΔH_α, and retroB..”.

[Comment 8]

Page 15. Second paragraph. To be more specific about the nature of the major achievement that the Authors have made, I recommend replacing “best-ever strategy” with a more specific statement, maybe that this study reports the highest rate of methanol production from methane per unit volume of culture.

Revision:

To more specifically state our achievement, we modified the phrase with removing the words, “best-ever strategy” as follows: “... - indicating that this novel approach is the best way of achieving the highest rate of methanol production from methane in microbial cultures.”.

Reviewer #2:**Major comments:****[Comment 1]**

The authors have improved several aspects of this manuscript, notably a better presentation of the methods used. However, the references to the literature for the structure and reaction of sMMO in the introduction remain incomplete, and the

characterization of the metal site in the preparations of the mini-sMMO remains quite problematic (see below). The fundamental discovery of a remarkable reworking of the sMMO system to allow expression in *E. coli* with high methanol production represents an important advance. The comparison to the metal center structure and the reaction cycle dynamics of the native sMMO system would be great to include, but it is not necessary for the fundamental story. In the revision, the authors try again to demonstrate the presence of the diiron cluster and to associate observed oxidation of the iron in the sample over a 16 h period to the native reaction cycle. Neither goal is accomplished, and it detracts from the overall goal of the study. In my opinion, the authors should consider omitting the comparison to sMMO mechanism and try simpler techniques to demonstrate a diiron cluster in the mini-sMMO.

Revision:

According to the reviewer's comment, we newly added two recent review articles (*Angew. Chem. Int. Ed.* **40**, 2782-2807 (2001) (ref. 25); *Annu. Rev. Biochem.* **88**, 409-431 (2019) (ref. 16)) to complete the references in Introduction of the second revised manuscript (p. 3).

Also, because sm1-bfr and sm1-afr are neither fully reduced nor oxidized form, it seems meaningless to compare the reaction cycle between sm1 and native sMMO based on the ratio of Fe(II) to Fe(III) of two sm1 samples above. Accordingly, in the second revised manuscript, we included EPR-, XANES-, and EXAFS data for only sm1-afr that was sampled from the *in vitro* reaction solution where methane oxidation was actively going on. Also, we discussed the results supported by evident analytical data and thus omitted the EPR- and XANES data-based comparison between mini-sMMO and native sMMO in terms of diiron cluster of active center and reaction cycle. (Please see our responses below to Comments 2 to 5.)

[Comment 2]

The addition of the spectra of the azide complex is a step in the right direction toward demonstrating a diiron cluster, but the intensity of the spectrum presented is far below that expected for a diiron cluster. There is a good example of the expected spectrum in the reference cited by the authors (PNAS, 107, 15391-15396, (2010), figure 2). It appears from spectrum in the revised manuscript that some fraction of the mini-sMMO may be in the diiron cluster state, but the amount is unknown without further comparison to examples from the literature. Similarly, the measurement of the EPR at 5K is a good step.

However, the simple additional step of adding dithionite to an anaerobic liquid sample, freezing the sample, and then attempting to detect the $g = 16$ EPR signal was not done. This signal could only emerge from a ferromagnetically coupled diferrous diiron cluster, and it would be a straightforward way to provide evidence that it is present. Instead, it is reported that no $g = 16$ signal is seen when reducing with NADH. NADH in combination with the reductase should have reduced the cluster to the diferrous state based on the redox potentials of the native cluster in sMMOH. However, the process used of freeze drying the sample and then attempting to interpret a powder sample is unlikely to give interpretable results. The procedures for doing this experiment have been reported many times in the literature.

Revision:

We performed a new azide experiment - which has been widely used to detect azide-diferrous oxo-bridge complex (*PNAS*, 107, 15391-15396 (2010); *JACS*, 144, 17611-17621 (2022)) - using the increased amount of sm1-af_r (1.5 μ M) and sodium azide (2 M) and added the new results to Suppl. Fig. 12a in the second revised manuscript. From the previous and new figures attached below, it is obvious that the intensities of two distinct peaks at 345 and 450 nm from azide-diferrous ion complex (indicated by blue arrows) significantly increased in the new figures. As compared to the former result for fully

oxidized enzyme (desaturase with Fe(III)·Fe(III)) (*PNAS*, **107**, 15391-15396 (2010)), the peak intensity looks weaker, suggesting that sm1-af_r is not a fully oxidized enzyme. We also performed another azide experiment using catalytic mutant of sm1 (sm1-cat_{mut} (1.5 μ M), prepared through the mutation of six ligands around the diiron center of native MMOH α) and sodium azide (2 M), demonstrating that the azide-diferrous complex peaks at 345- and 450-nm disappeared and thus indicating that diferrous oxo-bridges are present in the catalytic center of sm1 (Suppl. Figs. 12a,b). (Please see the response below to Major Comment 6.)

We agree that our current EPR data obtained with the freeze-dried samples of sm1 does not clearly show the characteristic signals of diiron cluster and its mixed valence state. Accordingly, we removed the simulated data and the EPR data-based discussion about diiron cluster formation in sm1 in the second revised manuscript.

[Previous figures (Suppl. Fig. 12)]

[New figures (Suppl. Fig. 12)]
[Comment 3]

The EPR spectra shown in Figures 4a,b and 5h,i from power samples are difficult to interpret in the context of a diiron cluster. Perhaps labeling the figures with g-values and enlarging the $g = 2$ region would help the reader. However, the presentation is confusing because Fe(II) has no EPR signal in the way these spectra were measured (perpendicular mode EPR), so the authors may be attempting to interpret them based on fits to a spin coupling model. If so, the details and assumptions of this method are not made clear. The signal the authors attribute to Fe(III) at 150 mT ($g = 4.3$) is due to mononuclear Fe(III) not diiron cluster iron. The very broad signal shown in the figures is often due to non-specifically bound iron rather than the typical sharp signal from active site mononuclear Fe(III). The observation of this signal shows that at least some of the iron quantified in the sample is not in a diiron cluster. The EPR signal for the mixed-valent form of the diiron cluster has 3 g-values, all below $g = 2$. This is quite characteristic for all antiferromagnetically coupled diiron clusters and would be another good way to argue for the presence of a diiron cluster in mini-sMMO if clearly presented. See for example in the paper cited by the authors, *J. Am. Chem. Soc.*, 115, 3688-3701 (1993), figure 7. In

contrast, the spectra shown in Fig. 4 have many signals in the $g = 2$ region, both above and below $g = 2$. The superposition of the bold simulation spectra over the data makes it impossible to see the data for accurate comparison. It is not possible from this figure to identify the characteristic signals of a diiron cluster, and there is nothing here that would allow a Fe(II) to Fe(III) ratio to be assigned.

Revision:

First, we included the g values in EPR data (Fig. 4a in the second revised manuscript) and added a new figure showing the enlarged $g=2$ region for sm1-afr without the superposition of the bold simulation spectra. Based on the reviewer's comment and the EPR results of previous literature (*J. Am. Chem. Soc.* **113**, 9219-9235 (1991)), we agree that our current EPR data obtained with the freeze-dried samples does not clearly show the presence of diiron cluster in active center and diiron's mixed valence state. Accordingly, we removed the simulated data and the EPR data-based discussion about diiron cluster formation in sm1 in the second revised manuscript.

On the other hand, according to another recent literature about EPR analysis of MMOR (*Inor. Chem. Front.*, **8**, 1279-1289 (2021)), isotropic EPR signal at $g=2$ indicates the presence of FAD radical as shown in the figure attached below.

The result above is nicely reproduced in the new Fig. 4a showing the enlarged $g=2$ region.

Inor. Chem. Front., **8**, 1279-1289 (2021)

That is, the strong isotropic EPR signal at $g=2$ - which was clearly detected with the sm1-afr sample, too - demonstrates the existence of FAD and active transfer of NADH electrons to FAD during methane oxidation in the presence of NADH. (The existence of FAD was confirmed later through measuring FAD-specific fluorescence (Fig. 5f).) Accordingly, with removing the EPR-based discussion about diiron cluster, we newly added the following discussion about the FAD radical formation in the second revised

manuscript (p. 10-11): “The strong isotropic signal at $g=2$ in the EPR spectra of sm1-afr (Fig. 4a) indicates the generation of FAD radicals⁴², which seems to be due to the active transfer of NADH electrons to FAD during methane oxidation. (The existence of FAD was also confirmed later through measuring FAD-specific fluorescence signal from sm1.)”

[Comment 4]

The author argue that the observation of the signal from Fe(III) at 150 mT means that the signal seen in Fig 5i (and maybe others) is not due to Cu. However, to me the spectrum in the $g = 2-3$ region looks like classic type II copper (for example, see Fig 1 of Angew. Chem. Int. Ed. 2007, 46, 1992 –1994)

Revision:

We agree that our current EPR data obtained with the freeze-dried samples of sm1 does not clearly show the characteristic signals of diiron cluster. Accordingly, we removed the EPR data-based discussion about diiron cluster formation in sm1 and Figs. 5h,i in the second revised manuscript.

[Comment 5]

The other spectra in Fig 4 designed to either show the presence of a diiron cluster or its oxidation during turnover also suffer from over-interpretation and/or lack of detail in describing how they were interpreted. Overall, in my opinion, the entire premise that the oxidation of iron over a 16 hour reaction somehow aligns the mini-sMMO reaction with the reaction cycle of the native enzyme is suspect. In the native cycle, the iron is reduced and oxidized each cycle on a seconds time scale. It oscillates between diferric, diferrous, diferric, di-Fe(IV) and back to diferric, all at 3 times per second at room temperature. What happens on an hours time scale is completely unrelated to this cycle. The XANES data shown in Fig. 4 can be interpreted to show a shift from Fe(II) to Fe(III), but this is on freeze-dried samples with oxidation occurring over 16 hours. It is not realistically possible to relate this data back to the native enzyme cycle. Also, the claim that this data is consistent with a mixed-valent diiron cluster of the enzyme is not correct. The data shows the presence of Fe(III) and Fe(II), not their presence in a coupled cluster.

Revision:

As stated in our responses to Comments 2 to 4 above, we agree with the reviewer’s opinion. In addition to the removal of the EPR data-based discussion about diiron cluster in sm1 and Figs. 5h,i, we further removed the XANES data-based discussion about the

comparison of catalytic cycle between sm1 and native sMMO. In the second revised manuscript (p. 11), the relevant text was modified as follows: “The XANES spectra (Fig. 4c) shows that both Fe(II) and Fe(III) are present in sm1-afr, the portion of Fe(III) being higher than Fe(II).”.

[Comment 6]

As previously pointed out, the EXAFS data in Fig 4d is shown without fits or Debye-Waller factors to characterize the quality of the fits. An EXAFS fit is based on distance, number of atoms and their electron count relative to a given iron. Without fits to show that the position and intensity of 2 irons at 3.2 Å (in combination with the other atoms expected in the MMOH cluster ligation) gives a better fit than other possibilities, the spectra are meaningless. The claimed shift due to iron oxidation in the peak assigned to the Fe-Fe scattering is too small to claim given the quality of the data and intensity of the peak. Moreover, the claimed shift (0.02 Å) is much too small based on the most recent EXAFS measurements cited by the authors, where it is shown to be about 0.3 Å (see J. Am. Chem. Soc. 2018, 140, 16807–16820, table 2).

Revision:

Because sm1-bfr and sm1-afr are neither fully reduced nor oxidized form, it seems meaningless to compare the EPR-, XANES-, and EXAFS data of the two sm1 samples above, and accordingly, in the second revised manuscript, we included EPR-, XANES-, and EXAFS data of only sm1-afr that is sampled from the *in vitro* reaction solution where methane oxidation is actively going on. Considering that sm1-afr was at methane-oxidizing state in the presence of NADH and dissolved oxygen, it seems a reasonable speculation that sm1-afr is at an intermediate state between fully reduced and fully oxidized state, which is supported by the previous literature (*Science*, **275**, 515-518, (1997)) about the EXAFS analysis of MMOH at an intermediate state that was prepared by mixing fully reduced (FeII•FeII) MMOH with 100% O₂-saturated buffer in the presence of two equivalents of MMOB.

According to the reviewer’s comment, we performed EXAFS fitting using the PDB structure (4GAM) of MMOH-MMOB complex from *M. capsulatus* (Bath) sMMO and presented the results in Fig. 4c with Debye-Waller factors (σ^2 values between 0.0 and 0.013, showing a good quality of the fits (meaning a reasonably low structural and vibrational disorder of atoms)). The fitted EXAFS data demonstrate that the inter-iron distance of sm1-afr is 2.755 Å, which is shorter than the distance of fully oxidized MMOH

(MMOH_{ox}) (3.056 Å) (*J. Am. Chem. Soc.* **140**, 16807–16820 (2018)) but longer than the distance of MMOH at an intermediate state (MMOH_{int}, mostly comprised of Q state (44-61%)) (2.46, 2.47 Å) (*Science*, **275**, 515-518, (1997)). Considering that sm1-afr was at methane-oxidizing state in the presence of NADH and dissolved oxygen, it seems that sm1-afr is at another intermediate state, closer to MMOH_{ox} than MMOH_{int} above. Both of the azide-diferric complex analysis (Supplementary Figs. 12a,b) and the fitted EXAFS (Fig. 4c) indicate the presence of diiron cluster in the active center of sm1.

We added the following sentence in the second revised manuscript (p. 11): “The absorption peaks of the EXAFS spectra of sm1-afr were fitted with the PDB structure (4GAM) of MMOH-MMOB complex from *M. capsulatus* (Bath), as presented in Fig. 4c with Debye-Waller factors (σ^2 between 0.0 and 0.013). The fitted EXAFS spectra demonstrate that the inter-iron distance of sm1-afr is 2.755 Å, which is between the distances of fully oxidized MMOH (MMOH_{ox}) (3.056 Å)⁴⁶ and MMOH at an intermediate state (MMOH_{int}, mostly comprised of Q state (44-61%)) (2.46, 2.47 Å)⁴⁷. Considering that sm1-afr was at methane-oxidizing state in the presence of NADH and dissolved oxygen, it seems that sm1-afr is at another intermediate state, closer to MMOH_{ox} than MMOH_{int} above. Consequently, the fitted EXAFS (Fig. 4c) and azide-diferric complex analysis (Supplementary Figs. 12a,b) as well as ICP-MS (Fig. 3e) and XANES (Fig. 4b) analysis indicate the presence of diiron cluster in the catalytic center of sm1.”

[Comment 7]

The addition of CD data to support proper folding of the mutated mini-sMMO is good. The mutations of the proposed diiron cluster ligands could also have been used to support the presence of the diiron cluster and its ligation by looking for loss of the azide complex UV-Vis spectrum, or loss of a g=16 in the diferrous dithionite-reduced state, or loss of a g-below 2 three line EPR spectrum in the mixed valent state.

Revision:

According to the reviewer’s suggestion, we performed an additional azide-diferric complex analysis using catalytic mutant of sm1 (sm1-cat_{mut} (1.5 μM), prepared through the mutation of six ligands (E114A, E144A, H147A, E209A, E243A, and H246A) around the diiron center of native MMOH α) and sodium azide (2 M) and presented the results in the new Supplementary Figs. 12a,b (attached in the response above to Comment 2), demonstrating that the distinct 345- and 450-nm peaks (corresponding to azide-diferric

oxo-bridge complex) disappears completely in sm1-cat_{mut}-afr and thus indicating that the diiron cluster is present in the catalytic center of sm1. We added the following sentence in the second revised manuscript (p. 11): “Azide-diferric complex analysis^{43,44} using sm1-afr and sm1-cat_{mut}-afr (prepared through the same procedure as sm1-afr) demonstrates that the distinct absorbance peaks (at 345 and 450 nm) for azide-diferric oxo-bridge complex were evidently detected with sm1-afr but never detected with sm1-cat_{mut}-afr, indicating the presence of diiron cluster in the catalytic center of sm1 (Supplementary Figs. 12a,b).”

[Comment 8]

1) The authors have added the very useful information concerning how the measurement of the turnover frequency was made by stating that it is based on changes only the beginning of the reaction. It is unclear why the turnover frequency decreases during the reaction, but there are many possibilities that do not affect the inherent maximum turnover frequency.

2) Based on comparisons to the turnover frequencies reported for several growth studies by groups trying to develop industrially relevant methanol production by methanotrophs, the authors argue that the native sMMO enzyme system has a much lower turnover frequency and ability to produce methanol than mini-sMMO. I continue to believe that the authors in the current work severely underestimate the turnover frequency of the native sMMO system. The growth characteristics of a commonly used methanotroph as well as the turnover frequency of the native sMMO system can be found in early papers describing the *M. trichosporium* OB3b system (see Methods in Enzymology (1990) vol. 188, 191-202). This study (and earlier primary literature on which it is based) showed 25 g/L/h of cells produced by continuous growth at an OD₆₀₀ of 8. My lab has verified the work reported in the Methods article and shown that this rate of production can be sustained for 10 days with production of 3-4 kg of cells.

3) The Methods article also lists the turnover frequency for the purified sMMO components for several substrates. For methane, the turnover frequency to yield methanol is reported to be 3.7 s⁻¹ (at 23 °C) which is far greater than the 0.32 s⁻¹ (at 30 °C) claimed for the mini-sMMO system by the authors. 4) It is also in line with the stopped-flow studies of the single turnover rate constants for the purified components, which show about 0.17 s⁻¹ at 4 °C for *M. trichosporium* OB3b sMMO with 10-20 fold higher rate constants at 23 or 30 °C (see for example Chem. Rev. 1996, 96, 2625-2657, figure 4).

Revision:

1) The turnover frequency (TF) of sm1 was measured at the early phase of *in vitro* methane oxidation (i.e. for 15 min after *in vitro* methane oxidation begins). We initially added an enough amount of methane and air to the head space (19 mL) of container (a septa-sealed vial (20 mL)) and NADH (0.3 mM) to the reaction buffer (1 mL) in vial to start methane oxidation, never followed by additional supply of methane and NADH. When we measured time-course change in NADH concentration in the *in vitro* reaction solution after methane oxidation began, the NADH consumption rate was the highest for initial 15 min and then slowed down, as shown in the figure attached below.

As long as enzyme is structurally and functionally intact, the constant (time-invariant) rate of enzyme reaction implies that the rate of substrate inflow to active center is always balanced with the rate of product outflow from the active center, which is regarded as an ideal state of enzyme reaction. Because the active center is initially empty, the rate of substrate inflow to active center and thus the substrate conversion rate is initially very fast; however, as product formation goes on, substrate inflow might be interfered, which in turn might decrease substrate conversion rate per enzyme (turnover frequency). That is, TF cannot be actually time-invariant, and thus it seems generally reasonable to accept measured TF of enzyme as “maximum moles of product that can be converted from a mole of substrate per unit time” (*Nature Protocols* **13**, 1506–1520 (2018)). According to the previous literatures reporting the TF (0.19 s^{-1}) of sMMO (*Biochemistry* **54**, 2283-2294 (2015); *Biochem. J.* **236**, 155–162. (1986)), the TF was measured for initial 1 min after the reaction was initiated by the addition of NADH to the assay mixture.

2) We never argue that the native sMMO enzyme system has a much lower turnover frequency than mini-sMMO. In the manuscript (p. 10), we already discussed about the TF of mini-sMMO as follows: “The turnover frequency (TF) of sm1 was about 0.32 s^{-1} (Supplementary Fig. 10), which is within a TF range of native sMMO ($0.2 \sim 1.0 \text{ s}^{-1}$)¹⁹.” Furthermore, the content of the suggested article (*Methods in Enzymology*, **188**, 191-202 (1990)) is summarized as attached below:

That is, after a long, 3-step cultivation that takes at least 3 days (i.e. the culture period of only first step is nearly 3 days), the fermentation culture of 10 L was harvested, and finally 18 to 25 g cells were obtained per L culture. Because the entire, 3-step culture period is not clearly described in this reference, it seems not possible to exactly estimate cell mass productivity. Also, the reviewer commented that 3-4 kg of cells was obtained for 10 days in the reviewer's own lab, but it is still unclear whether the 3-4 kg is wet or dry cell mass, and furthermore, because culture volume was not stated here, it seems not possible to compare the cell mass productivity (g/L/h) achieved by reviewer's work with our result. 3) In the construction of mini-sMMO, we used the MMOH, MMOR, and MMOB sequences of *M. capsulatus* (Bath) sMMO and thus compared the TF of mini-sMMO with the TF of *M. capsulatus* (Bath) sMMO (0.2~1.0 s⁻¹, *Biochemistry*, **54**, 2283–2294, (2015)). 3.7 s⁻¹ that the reviewer introduced is the TF of *M. trichosporium* OB3b sMMO (*Methods Enzymology* **188**, 191-202 (1990)), which is the reason why we did not introduce it in our manuscript.

We concerned about the difference in sequence and structure between two sMMOs from the different methanotroph strains. The sequence homology between two sMMOs is not quite high, as described as follows: 1) MMOH ((□₂□₂□₂): 80.2, 57.1, and 50.0 % for α, β, and γ subunit, respectively; 2) MMOB: 66.0 %; and 3) MMOR: 48.7 % (*Appl. Environ. Microbiol.* **63**, 1898-1904 (1997)). The figure above is the superposition images of two

different structures of MMOH-MMOB complex - one from *M. capsulatus* (Bath) sMMO (blue and orange, PDB 4GAM) and the other one from *M. trichosporium* OB3b sMMO (light blue and yellow, PDB 6YDO) - where only MMOH structures (blue and light blue) and diiron (orange and yellow) are visualized, showing the significant difference in active center including key ligands around diiron. It seems that we cannot exclude such a possibility that this structural difference between the two sMMOs could result in the different TF values.

4) The reference (*Chem. Rev.* **96**, 2625-2657 (1996)) suggested by the reviewer reported on the turnover rate constant of individual reaction step involved in the methanol producing, multi-step reaction path, during which diferrous MMOH is converted to diferric MMOH, as shown in the attached figure below. 0.17 s^{-1} at $4 \text{ }^\circ\text{C}$ is the rate constant of only final-step reaction by *M. trichosporium* OB3b MMOH. Accordingly, it seems unreasonable to compare the rate constant (0.17 s^{-1} at $4 \text{ }^\circ\text{C}$) with the overall TF (0.32 s^{-1} at $30 \text{ }^\circ\text{C}$) of sm1 including *M. capsulatus* (Bath) MMOH α .

Chem. Rev. **96**, 2625-2657 (1996))

[Comment 9]

As stated above, I think the important comparison the authors wish to make is the yield of actual recoverable methanol from the native methanotrophs vs the mini-sMMO. This differs from the comparison of the actual rate of methanol production by methanotrophs prior to it being utilized for cell growth. This comparison of actual recoverable methanol is valid in the current study, but it should not be claimed that the mini-sMMO is comparable to the rate of turnover for sMMO itself.

Revision:

A main objective of this study is to construct sMMO-derived, structurally minimized recombinant enzyme (mini-sMMO) that can be overexpressed with catalytic activity in *E. coli*, which rapidly grows to a high cell density and does not have a methanol oxidation pathway unlike methanotrophs and to evaluate the feasibility and potential of the mini-sMMO-expressing *E. coli*-based, productive biomanufacturing of methanol. To clearly enunciate this objective, we already included the following statement in our manuscript (p. 12-13): “Here we attempted a whole cell-based methane oxidation in a high-cell-density culture of sm1-expressing *E. coli* without the additional supply of any reductants, which might be a meaningful first step for the economically viable biomanufacturing of methanol.”

In the second revised manuscript, to emphasize the objective above, we added the word, “recoverable” in describing methanol yield and productivity as follows: “... the recoverable methanol productivity...” (p. 13); “The recoverable methanol production by the sm1-producing recombinant *E. coli* showed a remarkable improvement in productivity compared to the traditional methanotrophic production, ...” (p. 13); and “... the recoverable methanol yield and productivity were ...” (p. 15).

Minor comments:**[Comment 1]**

Line 43 – the regulatory component is not an enzyme

Revision:

We replaced “regulatory enzyme” with “regulatory protein” in the second revised manuscript.

[Comment 2]

Line 56 – 57 The references cited here do not come close to identifying all of the important contributions to kinetic and NMR characterization of the sMMO cycle and components. If the authors do not want to provide a list that includes the contributions from many research group, then citing only recent reviews is an alternative.

Revision:

According to the reviewer’s comment, we removed the references (ref. 25-32) about the

kinetic and NMR characterization of sMMO cycle and components and newly cited two recent review articles (*Angew. Chem. Int. Ed.* **40**, 2782-2807 (2001); *Annu. Rev. Biochem.* **88**, 409-431 (2019)) in the second revised manuscript (p. 3).

[Comment 3]

Line 64 – MMOB is not an enzyme

Revision:

We replaced “regulatory enzyme” with “regulatory protein” in the second revised manuscript.

[Comment 4]

Line 66 - The references cited here do not come close to identifying all of the studies that identified the roles played by MMOB

Revision:

According to the reviewer’s comment, we removed the references (ref. 34-36) about the critical role of MMOB and newly cited more relevant articles (*J. Am. Chem. Soc.* **136**, 2244-2247 (2014); *Nature* **494**, 380-384 (2013); *Annu. Rev. Biochem.* **88**, 409-431 (2019)) in the second revised manuscript (p. 3).

[Comment 5]

Line 229 – See comments above for the actual turnover frequency for native sMMO at room temperature. Also Rosensweig is not spelled correctly in ref 19.

Revision:

Please see the response above to Major Comment 8-3. We corrected the last name of second author of ref 19 to “Rosenzweig”.

[Comment 6]

Lines 234-241 – This shows that Fe is important for methanol production but not necessarily that it is a diiron cluster. The diiron cluster could be demonstrated directly (see comments above)

Revision:

Please see the responses above to Major Comments 2 to 6.

(We removed the EPR-based discussion about diiron cluster in the active center of sm1 and discussed about the diiron cluster based on the new results of azide-diferric complex analysis using sm1 and sm1-cat_{mut} and fitted EXAFS.)

[Comment 7]

Lines 248-267 – Much of this is incorrect or over-interpreted. All the authors need to show is that a diiron cluster is formed (methods listed above). As stated above, it does not seem necessary for the main point of the paper to try to show that the mini-MMO has the same cycle as native sMMO.

Revision:

Please see the responses above to Major Comments 2 to 6 and 9.

[Comment 8]

Line 264 – The difference in these distances is much too small and are not comparable to the results given in the cited reference.

Revision:

Please see the responses above to Major Comment 6.

[Comment 9]

Line 266 – The relevant comparison here would be to the structures of the completely oxidized and fully reduced MMOH-MMOB complexes (J. Am. Chem. Soc. 2020, 142, 14249–14266, fig 7) which agrees with the EXAFS from Cutsail but not the EXAFS difference in Fe-Fe distance in the current manuscript. The cited PDB structures do not have MMOB associated with MMOH and are not in the pure oxidation states due to reduction in the synchrotron beam.

Revision:

Please see the response above to Major Comment 6.

The PDB structures of PDB 1FZ1 and PDB 1FYZ we cited in the manuscript are for the structure of MMOH only. In the second revised manuscript (p. 11), we cited another PDB structure of MMOH-MMOB complex (PDB 4GAM) of *M. capsulatus* (Bath) and used it for EXAFS fitting as shown in Fig. 4c.

[Comment 10]

Line 340 – omit the word “enzyme”

Revision:

In the second revised manuscript (p. 14), “three enzyme components” was changed to “three components”.

[Comment 11]

Line 357 – nuclearity was not established

Revision:

In the second revised manuscript, we remove the word, “nuclearity”.

[Comment 12]

Line 334- end The discussion just restates what is in the results and then says the system the authors have developed can be fine-tuned. None of this is necessary or provided context. A comparison to previously published attempts to make a viable methanol producing system by using methanotrophs or using zeolites with embedded iron sites might be more useful.

Revision:

According to the reviewer’s suggestion, we additionally cited the following recent literatures in Discussion:

1) *Bioresour. Technol.*, **297**, 122433, (2020) (ref. 54): pure culture of *Methylocystis bryophila* enables MeOH production of 0.99 g/L through a repeated batch process (8 cycles)

2) *Bioresour. Technol.*, **364**, 128032, (2022) (ref. 55): co-culture of *Methylosinus sporium* and *Methylocella tundrae* enables MeOH production of 0.7 g/L and productivity of 0.009 g/L/h through a repeated batch process (10 cycles)

3) *Angew. Chem. Int. Ed.*, **56**, 16464 – 16483 (2017) (ref. 60): MMO-mimetic iron-embedded zeolites produce MeOH but also produces various by-products such as HCHO, MeOOH (methyl hydroperoxide), HCOOH, CO, and CO₂ under atmospheric condition, reaction selectivity problem remaining unresolved yet.

4) *Chem. Eur. J.*, **18**, 15735–15745 (2012) (ref. 58), *ACS Catal.*, **3**, 689–699 (2013) (ref. 59): Zeolite catalyst containing various metals as well as iron enhances methanol selectivity (<95%).

In the second revised manuscript (p. 15), we added the following phrase, "... by various methanotrophs (pure culture of *Methylocystis*⁵⁴, co-culture of *Methylosinus sporium* and *Methylocella tundrae*⁵⁵, etc.), other recombinant expression systems^{53,56,57}, or MMO-mimetic iron-embedded zeolite catalysts⁵⁸⁻⁶⁰, ...".

Reviewer #3:

[Comment 1]

We have one further query before the manuscript would be suitable for publication. The rationale behind the generation of the retro-B construct is still not clear. We thank you for clarifying that the sequence was actually reversed fully. The description of the modelling methods for the retro-B indicate that the structures for MD simulation were taken from the PDB for 4GAM and others. Alphafold predictions of retro-B do not give models with significant fold similarity to MMOB. RMSD Ca of over 10 Å! Could the authors please comment further on the rationale for the retro-B construct and their modelling and what this might mean for the role of the modified retro-B.

Revision:

Because MMOH with diiron active center is much more activated through interacting with MMOB, it is of crucial importance to make MMOH and MMOB optimally interact with each other in the construction of mini-sMMO, which is based on the reassembly of minimal and modified version of native sMMO (R_{FAD} , $\Delta H\alpha$, and B/retroB, where B and retroB represent the normal and reversed sequence of native MMOB, respectively) on the huHF scaffold (Table 1). For the reassembly on the huHF scaffold, the N-termini of $N-R_{FAD}-\Delta H\alpha-C$ and B/retroB are genetically linked to the C-termini of huHF. Reportedly, a structural study of native MMOH-MMOB complex (PDB 4GAM) revealed that 1) the N-terminal region (~ 35 a.a.) of MMOB plays a critical role in the interaction with reduced MMOH α and 2) the N- and C-termini of MMOH α and MMOB, respectively, are facing the same direction (Supplementary Figs. 1a,b), which suggests that the N-terminal region of MMOB needs to be freely interact with $\Delta H\alpha$ within the construct of mini-sMMO. Accordingly, the C-terminus of MMOB needs to be linked to the C-terminus of huHF. This is the reason why we tried to link the N-terminus of retroB to the each C-terminus of huHF.

Instead of the simulation study based on Alphafold prediction of B or retroB alone, we

performed a series of MD simulations to investigate the contact maps between the residues of AHa and B/retroB within the mini-sMMO (sm1) (Figs. 2e-g). The contact map between native MMOHa and MMOB (Fig. 2e) reveals the 9 significant interactive regions, while in the contact map between AHa (in *N*-R_{FAD}-AH α -C) and B, the 9 interactive regions were notably weakened or disappeared (Fig. 2f). Surprisingly, the native contact pattern between native MMOHa and MMOB (Fig. 2e) was almost perfectly reproduced in the contact map between AHa (in *N*-R_{FAD}-AH α -C) and retroB (Fig. 2g). Based on these results of contact map analysis by MD simulation, we actually constructed the mini-sMMO (sm1) using the components of R_{FAD}-AH α and retroB and successfully proved that sm1 is catalytically active in methane oxidation.

We investigated the stability of B (MMOB) and retroB that exist alone (unbound) or at the state of binding to *N*-R_{FAD}-AH α -C through another MD simulation for RMSD analysis. As pointed out by reviewer, like Alphafold prediction, our MD simulations also revealed that the unbound form of retroB exhibits significantly poorer structural stability compared to unbound B: the RMSD of unbound retroB (~ 12 Å, red curve in Figure below) is roughly two-fold larger than that of the unbound B (~ 6.2 Å, blue curve in Figure below) with more substantial fluctuations.

RMSD comparison of B and retroB: unbound vs. bound to *N*-R_{FAD}-AH α -C forms.

A surprising feature of the retroB construct is its binding capability to *N*-R_{FAD}-AH α -C. Upon binding to *N*-R_{FAD}-AH α -C, the RMSD of retroB is notably reduced, showing a dramatic drop from 12 Å for unbound state to ~ 5.5 Å for bound state (pink curve Figure above). This is in a sharp contrast to B when bound to *N*-R_{FAD}-AH α -C, which exhibits

even larger RMSD compared to its unbound form (green curve in Figure above), implying that retroB binds better with $N\text{-R}_{\text{FAD}}\text{-}\Delta\text{H}\alpha\text{-C}$ than B. Considering that R_{FAD} is essential to electron transfer from NADH to diiron center of $\Delta\text{H}\alpha$ and that the $N\text{-R}_{\text{FAD}}\text{-}\Delta\text{H}\alpha\text{-C}$ is a viable option for securing FAD and $\Delta\text{H}\alpha$ (Figs. 2a-d), retroB seems to be much better partner for $N\text{-R}_{\text{FAD}}\text{-}\Delta\text{H}\alpha\text{-C}$ in the construction of mini-sMMO (sm1).

REVIEWERS' COMMENTS

Reviewer #1 (Remarks to the Author):

The Author have attended to all the issues that I raised on the previous version.

I suggest one remaining change, to the new text describing the Zill et al. study, where the Authors say (lines 340-342): "...the effectiveness of the expression system remains unknown, because any qualitative and quantitative characteristics (structure, function, expression yield, catalytic efficiency, etc.) of the enzyme were not revealed yet." This is not wholly accurate because in the Zill et al. study specific activity was quantified using purified recombinant protein with the sMMO assay substrate p-nitrobenzene, and the iron centre of the hydroxylase was characterised via electron paramagnetic resonance spectroscopy.

Reviewer #2 (Remarks to the Author):

The authors have improved the manuscript by de-emphasizing the comparison of the mini-MMO with the structure and reaction cycle of native sMMO. As I suggested before, this comparison is not necessary to support for the main point of the study. In the revised manuscript, the authors have introduced some new interpretations of the data, which are again incorrect (or incomplete) and not necessary for the story they wish to convey. It should be straightforward to correct these problems.

Line 229 – The reference here is to a review which itself references old literature (Green and Dalton) on *M. capsulatus* sMMO turnover frequency. The early methods used to purify *M. capsulatus* sMMO failed to increase the specific activity, meaning that activity was lost at about the same rate the protein was purified. This was corrected when a new procedure was developed for *M. trichosporium* OB3b (Fox et al). Before the new procedure, the preparation of sMMO from *M. trichosporium* also gave the same low specific activity as that from *M. capsulatus* (0.2 s⁻¹ or lower). However, using the new prep, the *M. trichosporium* OB3b showed a steady increase in specific activity during the prep and a turnover frequency of 3.5 s⁻¹ for the homogeneous enzyme. It is likely that *M. capsulatus* sMMO would show the same high activity if similarly prepared because single turnover experiments show similar rate constants for the reaction cycle steps for sMMOs from the two sources. These rate constants were determined from the optical characteristics of MMOH and are measured under pseudo first order conditions in substrates, so they reflect the active fraction of the MMOH preparation. The more recent preparations of the sMMO from *M. capsulatus* show turnover frequency of 1 s⁻¹, which is trending toward the value

observed for *M. trichosporium* sMMO. The bottom line is that the value seen for the mini-sMMO is not in the range of fully active sMMO from either source.

In the rebuttal letter, the authors show a superposition of the metal centers for *M. capsulatus* sMMO and *M. trichosporium* OB3b sMMO in the MMOH-MMOB complex. It is argued that they are somewhat different, so it is reasonable that the turnover frequencies would also be different. However, the resolution of the structure from *M. capsulatus* is much worse (2.9 Å) than that from *M. trichosporium* complex (1.9 Å). In protein crystallography, this is a major difference that would introduce a large errors in the distances from the *M. capsulatus* structure. Other spectroscopies and single turnover kinetic studies have failed to detect any significant differences between the complexes from the two sources. It is also true that the components from the two sources can be interchanged to give nearly full activity, so the important aspects of the interactions must be conserved. Consequently, all sources of native sMMO are likely to have a much higher turnover frequency than the mini-MMO. For the main point of the manuscript, this comparison does not matter and does not need to be made.

Lines 260-269 – This is an improvement in the reporting of the EXAFS data, but it is still lacking. EXAFS data from others is always presented as a series of fits of potential structures (i.e. 1 Fe vs 2 Fe, 4,5,6 ligands to the irons, 2 vs 3 nitrogen ligands etc.). The Debye-Waller values for these trials allows the reader to judge the best model. The values for the single fit seem to be reasonable based on the assumption that there is a diiron cluster present that is like that in the crystal structure. However, the Fe-Fe distance cannot be accurate to 3 decimal places as reported based on the quality of the fit. Also, the logic in the text that the Fe-Fe distance represents an average of diferric and an intermediate (Q) with a very short Fe-Fe distance cannot be supported by the data. Recent EXAFS studies have shown that Q has a much longer Fe-Fe distance than initially reported (in ref 47). The error in the earlier report is due to the geometry of the cell holder compartment of all traditional EXAFS instruments. When a weak (i.e. biological) sample is in the compartment, the traditional detector picks up scattering from the iron in the cell holder outside of the sample, leading to an erroneous distance. A new type of detector (HERFD) picks up scattering only from the sample and gives an Fe-Fe distance that is longer, not shorter, than that for diferric MMOH (ref 46). It is likely that the Fe-Fe distance reported in the current manuscript has a considerable error due to these factors. Also, Q has a very short lifetime in the presence of methane, so it will not be present in any concentration during turnover. One interpretation of the XANES data presented is that there is a mixture of Fe(III) and Fe(II) is present in the sample. This mixture should give a slightly longer average Fe-Fe distance than seen in the diferric enzyme. The bottom line is that the authors cannot claim that the short Fe-Fe distance observed is due to the mixture of diferric and a Q-like species in a freeze-dried sample after 16 h of reaction. These statements are not necessary for the main points of the manuscript and should be removed.

Reviewer #3 (Remarks to the Author):

We thank the authors for their consideration of the reviewer comments and appreciate the thoughtful additions and new data to support their conclusions.

Further to questions regarding the retro-B structure and modelling it is still not clear how the model for this protein was generated and prepared for MD simulations. In the methods section the authors state that models were taken from the PDB. In the case of retro-B there is not a suitable model for it in the PDB as it is my understanding that it is the inverted structure of the MMOB. One would assume that an alpha fold model would be used in the MD simulations. Could the authors clarify this please in the methods and comment on the rationale for the inversion of the sequence - which is really not clear at this point.

** See Nature Portfolio's author and referees' website at www.nature.com/authors for information about policies, services and author benefits

Revision Details

Reviewer #1:

[Comment 1]

I suggest one remaining change, to the new text describing the Zill et al. study, where the Authors say (lines 340-342): "...the effectiveness of the expression system remains unknown, because any qualitative and quantitative characteristics (structure, function, expression yield, catalytic efficiency, etc.) of the enzyme were not revealed yet." This is not wholly accurate because in the Zill et al. study specific activity was quantified using purified recombinant protein with the sMMO assay substrate p-nitrobenzene, and the iron centre of the hydroxylase was characterised via electron paramagnetic resonance spectroscopy.

Revision:

In the revised text (p. 14), we modified the relevant sentence as follows: "Recently, the co-expression of recombinant sMMO and *E. coli* chaperonins was reported⁵⁰; however, the effectiveness of the expression system was not fully evidenced although catalytic oxidation of p-nitrobenzene was verified with the EPR-based characterization of iron center of hydroxylase."

Reviewer #2:

[Comment 1]

The authors have improved the manuscript by de-emphasizing the comparison of the mini-MMO with the structure and reaction cycle of native sMMO. As I suggested before, this comparison is not necessary to support for the main point of the study. In the revised manuscript, the authors have introduced some new interpretations of the data, which are again incorrect (or incomplete) and not necessary for the story they wish to convey. It should be straightforward to correct these problems.

Line 229 – The reference here is to a review which itself references old literature (Green and Dalton) on *M. capsulatus* sMMO turnover frequency. The early methods used to purify *M. capsulatus* sMMO failed to increase the specific activity, meaning that activity was lost at about the same rate the protein was purified. This was corrected when a new procedure was developed for *M. trichosporium* OB3b (Fox et al). Before the new

procedure, the preparation of sMMO from *M. trichosporium* also gave the same low specific activity as that from *M. capsulatus* (0.2 s⁻¹ or lower). However, using the new prep, the *M. trichosporium* OB3b showed a steady increase in specific activity during the prep and a turnover frequency of 3.5 s⁻¹ for the homogeneous enzyme. It is likely that *M. capsulatus* sMMO would show the same high activity if similarly prepared because single turnover experiments show similar rate constants for the reaction cycle steps for sMMOs from the two sources. These rate constants were determined from the optical characteristics of MMOH and are measured under pseudo first order conditions in substrates, so they reflect the active fraction of the MMOH preparation. The more recent preparations of the sMMO from *M. capsulatus* show turnover frequency of 1 s⁻¹, which is trending toward the value observed for *M. trichosporium* sMMO. The bottom line is that the value seen for the mini-sMMO is not in the range of fully active sMMO from either source.

In the rebuttal letter, the authors show a superposition of the metal centers for *M. capsulatus* sMMO and *M. trichosporium* OB3b sMMO in the MMOH-MMOB complex. It is argued that they are somewhat different, so it is reasonable that the turnover frequencies would also be different. However, the resolution of the structure from *M. capsulatus* is much worse (2.9 Å) than that from *M. trichosporium* complex (1.9 Å). In protein crystallography, this is a major difference that would introduce a large errors in the distances from the *M. capsulatus* structure. Other spectroscopies and single turnover kinetic studies have failed to detect any significant differences between the complexes from the two sources. It is also true that the components from the two sources can be interchanged to give nearly full activity, so the important aspects of the interactions must be conserved. Consequently, all sources of native sMMO are likely to have a much higher turnover frequency than the mini-MMO. For the main point of the manuscript, this comparison does not matter and does not need to be made.

Revision:

In the revised manuscript, we removed the discussion about the comparison of catalytic activity (such as turnover frequency) between mini-sMMO and native sMMO. That is, the phrase, "... which is within a TF range of native sMMO (0.2 ~ 1.0 s⁻¹)¹⁹." (line 228229 of the second revised manuscript) was deleted.

[Comment 2]

Lines 260-269 – This is an improvement in the reporting of the EXAFS data, but it is still lacking. EXAFS data from others is always presented as a series of fits of potential

structures (i.e. 1 Fe vs 2 Fe, 4,5,6 ligands to the irons, 2 vs 3 nitrogen ligands etc.). The Debye-Waller values for these trials allows the reader to judge the best model. The values for the single fit seem to be reasonable based on the assumption that there is a diiron cluster present that is like that in the crystal structure. However, the Fe-Fe distance cannot be accurate to 3 decimal places as reported based on the quality of the fit. Also, the logic in the text that the Fe-Fe distance represents an average of diferric and an intermediate (Q) with a very short Fe-Fe distance cannot be supported by the data. Recent EXAFS studies have shown that Q has a much longer Fe-Fe distance than initially reported (in ref 47). The error in the earlier report is due to the geometry of the cell holder compartment of all traditional EXAFS instruments. When a weak (i.e. biological) sample is in the compartment, the traditional detector picks up scattering from the iron in the cell holder outside of the sample, leading to an erroneous distance. A new type of detector (HERFD) picks up scattering only from the sample and gives an Fe-Fe distance that is longer, not shorter, than that for diferric MMOH (ref 46). It is likely that the Fe-Fe distance reported in the current manuscript has a considerable error due to these factors. Also, Q has a very short lifetime in the presence of methane, so it will not be present in any concentration during turnover. One interpretation of the XANES data presented is that there is a mixture of Fe(III) and Fe(II) is present in the sample. This mixture should give a slightly longer average Fe-Fe distance than seen in the diferric enzyme. The bottom line is that the authors cannot claim that the short Fe-Fe distance observed is due to the mixture of diferric and a Q-like species in a freeze-dried sample after 16 h of reaction. These statements are not necessary for the main points of the manuscript and should be removed.

Revision:

In the revised manuscript, we removed the discussion about the comparison of EXAFS results (such as inter-iron distance within iron center) between mini-sMMO and native sMMO. That is, we deleted the sentences, "... which is between the distances of fully oxidized MMOH (MMOH_{ox}) (3.056 Å)⁴⁶ and MMOH at an intermediate state (MMOH_{int}, mostly comprised of Q state (44-61%)) (2.46, 2.47 Å)⁴⁷. Considering that sm1-afr was at methane-oxidizing state in the presence of NADH and dissolved oxygen, it seems that sm1-afr is at another intermediate state, closer to MMOH_{ox} than MMOH_{int} above." (line 263-267 of the second revised manuscript).

Reviewer #3:

[Comment 1]

Further to questions regarding the retro-B structure and modelling it is still not clear how the model for this protein was generated and prepared for MD simulations. In the methods section the authors state that models were taken from the PDB. In the case of retro-B there is not a suitable model for it in the PDB as it is my understanding that it is the inverted structure of the MMOB. One would assume that an alpha fold model would be used in the MD simulations. Could the authors clarify this please in the methods and comment on the rationale for the inversion of the sequence - which is really not clear at this point.

Revision:

First above all, we would like to emphasize again the following important characteristics of native sMMO, which should be considered in designing mini-sMMO: 1) the N-terminal region (~ 35 a.a.) of MMOB plays a critical role in the interaction with reduced MMOH α and 2) the N- and C-terminus of MMOH α and MMOB, respectively, are facing the same direction (Supplementary Figs. 1a,b). This suggests that the N-terminal region of MMOB needs to be freely interact with Δ H α within the construct of mini-sMMO. Accordingly, the C-terminus of MMOB (i.e. the N-terminus of reversed sequence of MMOB (retroB)) needs to be linked to the C-terminus of huHF.

As suggested by the reviewer, the model structure of retroB could be generated using AlphaFold or RosettaFold; however, the objective of this study is not to find the stable structure of retroB but to determine which one of B and retroB can function more efficiently as a regulatory protein for R_{FAD}- Δ H α , that is, can establish native binding contacts with Δ H α in R_{FAD}- Δ H α , which seems practically infeasible through the AI-based model prediction, given the formidably vast configurational space associated with the binding between R_{FAD}- Δ H α and regulator. Therefore, we chose the structure of native MMOH-MMOB complex (PDB 4GAM) as a defined binding template and investigated which one of B and retroB shows a better compatibility with the predefined template upon the binding with R_{FAD}- Δ H α . That is, we used the 4GAM conformation of MMOB as the initial structural template for the bound form of retroB and adopted the mirror conformation of MMOB of 4GAM structure as retroB - which retains the reversed sequence of MMOB - through assigning the N- and C-terminus of MMOB as C- and N-term of retroB, respectively. Then, we evaluated the stability of retroB in the binding with MMOH or R_{FAD}- Δ H α through MD simulation (RMSD and contact map analysis). Our design strategy prioritizes determining whether retroB can tolerate or even thrive within such a defined binding template, rather than finding the thermodynamically most stable

structure of retroB.

In Methods (p. 27), we newly added the following sentences: “Our computational approach aimed at designing a regulatory protein for $R_{FAD-\Delta H\alpha}$, so that the binding characteristics observed in the native MMOH-MMOB complex system can be reproduced. We selected the MMOB structure of native MMOB-MMOH complex (PDB 4GAM) as the initial framework for simulating retroB. That is, we constructed retroB structure model through adopting the mirror conformation of MMOB of 4GAM structure - assigning the N- and C-terminus of MMOB as C- and N-terminus of retroB, respectively - which retains the reversed sequence of MMOB. Then, we evaluated the stability of retroB upon the binding with MMOH or $R_{FAD-\Delta H\alpha}$ through MD simulation (RMSD and contact map analysis).”